# Winning the Pruning Gamble: A Unified Approach to Joint Sample and Token Pruning for Efficient Supervised Fine-Tuning

## Abstract

As supervised fine-tuning (SFT) evolves from a lightweight post-training step into a compute-intensive phase rivaling mid-training in scale, data efficiency has become critical for aligning large language models (LLMs) under tight budgets. Existing data pruning methods suffer from a fragmented design: they operate either at the sample level or the token level in isolation, failing to jointly optimize both dimensions. This disconnect leads to significant inefficiencies—high-value samples may still contain redundant tokens, while token-level pruning often discards crucial instructional or corrective signals embedded in individual examples. To address this bottleneck, we introduce the *Error–Uncertainty (EU) Plane*, a diagnostic framework that jointly characterizes the heterogeneous utility of training data across samples and tokens. Guided by this insight, we propose *Quadrant-based Tuning (Q-Tuning)*, a unified framework that strategically coordinates sample pruning and token pruning. Q-Tuning employs a two-stage strategy: first, it performs sample-level triage to retain examples rich in informative misconceptions or calibration signals; second, it applies an asymmetric token-pruning policy, using a context-aware scoring mechanism to trim less salient tokens exclusively from misconception samples while preserving calibration samples in their entirety. Our method sets a new state of the art across five diverse benchmarks. Remarkably, on SmolLM2-1.7B, Q-Tuning achieves a +38% average improvement over the full-data SFT baseline using only 12.5% of the original training data. As the first dynamic pruning approach to consistently outperform full-data training, Q-Tuning provides a practical and scalable blueprint for maximizing data utilization in budget-constrained LLM SFT. *The code is attached in the supplementary materials.*

## 1 Introduction

The explosive growth of alignment datasets—now routinely spanning billions of tokens—has fundamentally transformed Supervised Fine-Tuning (SFT) from a lightweight post-training step into a compute-intensive phase rivaling mid-training in scale (Ouyang et al., 2022; Dong et al., 2023; Yang et al., 2025; Achiam et al., 2023; Team et al., 2023). In this new regime, the primary challenge is no longer simply reducing data volume, but maximizing the utility of every retained token and sample—a task that demands accurate, on-the-fly estimation of data value. Yet despite the emphasis on data efficiency, recent work reveals a troubling paradox: even sophisticated dynamic pruning heuristics often underperform *simple random sampling* (Xia et al., 2024b). This starkly exposes a fundamental disconnect between current strategies and the true utility of alignment data.

At the heart of this challenge lies the fragmented design of existing pruning strategies. *Sample-level pruning* methods (Qin et al., 2024; Zhou et al., 2023a; Wang et al., 2025a; Yang et al., 2024) identify high-potential examples but treat all tokens within them as equally valuable—retaining redundant or even harmful content that dilutes alignment signals. Conversely, *token-level pruning* approaches (Lin et al., 2024; Xia et al., 2025; Chen et al., 2024b; Zhang et al., 2024c) apply context-agnostic heuristics uniformly across the dataset, blind to the semantic role of each sample. Such a one-size-fits-all policy fails to differentiate, for instance, between a sample containing a correctable misconception—where only specific tokens need refinement—and one serving as a calibration anchor,

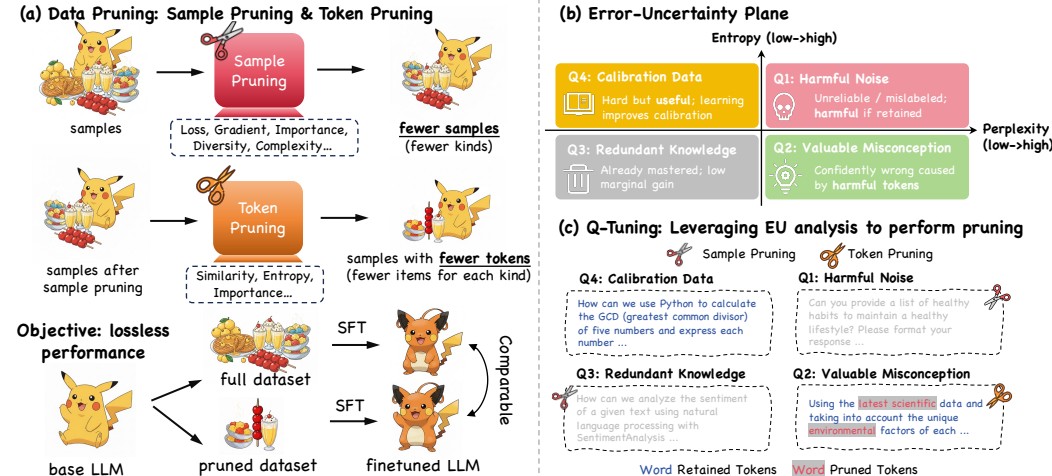

Figure 1: **(a) Data Pruning.** *Sample Pruning* reduces the training set size by selecting samples based on criteria such as loss, gradient magnitude, importance, diversity, and complexity; *Token Pruning* further refines individual samples by removing redundant or low-value tokens using metrics like similarity, entropy, and token importance. The pruned dataset is then used to fine-tune a base LLM, achieving the objective to match the performance finetuned with the full data. **(b) Error-Uncertainty (EU) Plane.** A two-dimensional analytical framework that categorizes training samples into four quadrants based on perplexity and entropy: Q1 (harmful noise — unreliable or mislabeled data), Q2 (valuable misconceptions — correctable errors that enhance learning), Q3 (redundant knowledge — already mastered content), and Q4 (calibration data — challenging yet informative samples that improve model calibration). **(c) Q-Tuning.** Unlike traditional approaches that apply sample and token pruning independently, our proposed Q-Tuning leverages the EU plane to jointly guide pruning at both levels. First, all harmful noise (Q1) and redundant knowledge (Q3) are pruned. Then, within valuable misconceptions (Q2), Q-Tuning selectively prunes detrimental tokens while preserving beneficial calibration samples (Q4) in full.

which must be preserved holistically to maintain model stability. By operating in isolation, neither paradigm captures the interdependent nature of sample and token utility. This raises a central question: *How can we dynamically coordinate sample selection and token pruning within a unified framework to maximize the true learning utility of limited data?*

To address this, we first formalize the problem as **Generalized Dynamic Data Pruning**, a bilevel optimization framework for jointly optimizing sample and token pruning. We then introduce a novel diagnostic lens, the **Error-Uncertainty (EU) Plane**, which categorizes training instances by mapping model error (perplexity) against model uncertainty (entropy). Specifically, as shown in Figure 1(b), training samples are categorized into four quadrants based on perplexity (low → high, indicating uncertainty) and entropy (low → high, indicating prediction confidence): (i) *Q1 (Harmful Noise)* — unreliable or mislabeled data that actively harms learning and should therefore be removed via sample-level pruning; (ii) *Q2 (Valuable Misconception)* — confidently wrong responses that, when pruned surgically at the token level, can be transformed into powerful teaching signals, making them ideal candidates for token-level pruning; (iii) *Q3 (Redundant Knowledge)* — mastered content offering diminishing returns, best eliminated through sample-level pruning to improve efficiency without sacrificing performance; and (iv) *Q4 (Calibration Data)* — hard but reliable samples essential for improving model confidence and robustness, which should be preserved in full — neither sample nor token pruning should be applied.

This insight directly motivates our solution: **Quadrant-based Tuning (Q-Tuning)**, the first principled, integrated method for dynamic data pruning. Guided by EU Plane analysis, Q-Tuning implements a two-stage, context-aware strategy, as shown in Figure 1(c). First, at the sample level, it acts as an intelligent triage system: retaining samples that offer clear signals for error correction or calibration (Q2 and Q4), while discarding those classified as harmful noise or redundant knowledge (Q1 and Q3). Crucially, Q-Tuning then applies a asymmetric token policy: for confidently wrong samples (valuable misconceptions), it performs token pruning to isolate the core misconception and amplify the learning signal; for uncertain but correct samples (calibration data), it preserves full token sequences to ensure robust uncertainty modeling. Our contributions are as follows:

1. We formalize the joint sample-token pruning problem through the **Generalized Dynamic Data Pruning** framework — a bilevel optimization objective for hybrid pruning strategies.

2. We introduce the **Error-Uncertainty (EU) Plane**, a tool that reveals and quantifies the heterogeneous value of data across error and uncertainty dimensions. Based on the EU analysis, we propose **Q-Tuning**, the first integrated, diagnosis-driven algorithm for dynamic pruning that coordinates sample and token decisions based on EU Plane insights.

3. Experiments demonstrate that Q-Tuning exceeds full-data training and all existing pruning baselines across 4 models, 5 benchmarks, and 6 different kinds of data budgets. Particularly, with LLaMA3-8B on GSM8K, Q-Tuning achieves 48.07 using only 35% of the data (surpassing full training by 6.0 and the best baseline by 9.9).

## 2 GENERALIZED DYNAMIC DATA PRUNING: A UNIFIED FRAMEWORK

We first propose *Generalized Dynamic Data Pruning*, a framework for accelerating model training where samples and their constituent tokens are selectively and adaptively omitted at each step.

Specifically, the framework first considers the coarse-grained *sample level pruning*, which involves identifying and discarding examples from a mini-batch deemed less informative for the model's state. Subsequently, *token-level pruning* operates on the retained samples, performing a finer-grained selection to keep a critical subset of tokens within each. This two-stage process is inherently *dynamic*: pruning criteria can be re-evaluated for each new mini-batch, allowing the data distribution used for gradient updates to evolve alongside the model's parameters $\theta$. The overarching objective is to focus computation on a "doubly-pruned" data subset to maximize training efficiency while preserving or enhancing the model's generalization.

To formalize this framework, consider a model $f_\theta$ with parameters $\theta$. At each training step $t$, a mini-batch $\mathcal{B}_t$ is drawn from the training distribution $\mathcal{D}$. The hierarchical pruning process can be modeled by two abstract operators: a sample-level pruner $\Psi$ and a token-level pruner $\Phi$.

**Stage 1: Sample-Level Pruning.** The operator $\Psi$ partitions mini-batch $\mathcal{B}_t$ into kept and discarded samples, governed by keep ratio $r_{\text{sample}} \in [0, 1]$. The retained set, $\mathcal{B}'_t$, is defined as:

$$\mathcal{B}'_t = \mathcal{B}_t \setminus \Psi(\mathcal{B}_t) \subseteq \mathcal{B}_t, \tag{1}$$

where $\Psi(\mathcal{B}_t)$ represents the set of discarded samples, and the size of the retained set is constrained such that $|\mathcal{B}'_t| = \lfloor r_{\text{sample}} \cdot |\mathcal{B}_t| \rfloor$.

**Stage 2: Token-Level Pruning.** For each sample in the retained set $\mathcal{B}'_t$, the operator $\Phi$ determines which tokens to keep, guided by a token keep ratio $r_{\text{token}} \in [0, 1]$. This is modeled by generating a binary mask $m(x) \in \{0, 1\}^{L(x)}$ for each sample $x \in \mathcal{B}'_t$, where $L(x)$ is its sequence length. The final, doubly-pruned mini-batch $\tilde{\mathcal{B}}_t$ is constructed by applying these masks:

$$\tilde{\mathcal{B}}_t = \Phi(\mathcal{B}'_t) = \{m(x) \odot x \mid x \in \mathcal{B}'_t\}, \tag{2}$$

where $\odot$ denotes element-wise product, and each mask satisfies $\|m(x)\|_1 = \lfloor r_{\text{token}} L(x) \rfloor$.

**Generalized Dynamic Data Pruning**. We now put all things together into a unified framework. The central problem of this framework is to identify the optimal dynamic pruning operators, $(\Phi, \Psi)$, that guide the training process to a final model $\theta^*$ with the best possible generalization performance. These operators are applied at each step $t$ to transform a mini-batch $\mathcal{B}_t$ into a computationally cheaper, pruned version $\tilde{\mathcal{B}}_t$, while adhering to predefined keep ratios $(r_{\text{sample}}, r_{\text{token}})$.

This problem is naturally captured as a *bi-level optimization problem*. The outer loop seeks optimal pruners, while the inner loop represents the iterative training procedure that produces the final model parameters under the guidance of these pruners. Formally, the objective is as follows:

$$\min_{\Phi, \Psi} \quad \mathbb{E}_{(x,y) \sim \mathcal{D}_{\text{test}}} \left[ \mathcal{L}_{\text{test}}(f_{\theta^*}(x), y) \right] \quad \text{s.t.} \quad \theta^* = \arg\min_\theta \sum_{t=1}^{T} \mathbb{E}_{\mathcal{B}_t \sim \mathcal{D}} \mathbb{E}_{(x,y) \sim \tilde{\mathcal{B}}_t} \left[ \mathcal{L}_{\text{train}}(f_{\theta_t}(x), y) \right]$$

$$\tag{3}$$

where at each step $t$: $\tilde{\mathcal{B}}_t = \Phi(\mathcal{B}_t \setminus \Psi(\mathcal{B}_t))$, with keep ratios $(r_{\text{sample}}, r_{\text{token}})$.

Figure 2: **(a) Constructing the Error-Uncertainty (EU) Plane via Bisect Search.** Given a dataset of training samples, we first perform inference using the base LLM to compute perplexity (PPL) and entropy for each sample. These metrics are then used to plot the samples on the EU-plane. A bisect search is applied to dynamically determine optimal thresholds ($\alpha_{\text{low}}^*$, $\alpha_{\text{high}}^*$, $\beta_{\text{low}}^*$, $\beta_{\text{high}}^*$) that partition the plane into four quadrants: Q1 (Harmful Noise), Q2 (Valuable Misconception), Q3 (Redundant Knowledge), and Q4 (Calibration Data). **(b) Sample Pruning for Q1 and Q3.** Samples in Q1 (harmful noise) and Q3 (redundant knowledge) are pruned at the sample level, while those in Q2 and Q4 are retained for further processing. **(c) Token-Level Pruning for Q2 Samples.** For retained Q2 samples, token-level pruning is applied based on both the token's own perplexity and the average perplexity of its neighboring tokens. Tokens with high local PPL are removed, preserving only the most informative ones, while all tokens in Q4 samples are kept intact to maintain calibration signals.

Specifically, the *outer objective* defines the quality metric for any pair of pruners $(\Phi, \Psi)$: the final test performance of the model they produce. The *inner objective* defines the training process itself, where the final parameters $\theta^*$ result from cumulatively minimizing the loss over a sequence of dynamically pruned mini-batches. All existing method that instantiates these operators $\Phi$ and $\Psi$ can be seen as a specific solution to this general problem.

## 3 WINNING THE PRUNING GAMBLE

Building on the previous analysis, we now introduce **Q-Tuning**, a dynamic pruning method guided by the Error–Uncertainty (EU) Plane. As shown in Figure 1, Q-Tuning proceeds in two coordinated stages: it first prunes harmful noise and redundant knowledge at the sample level, and then applies targeted token-level pruning to valuable misconceptions while preserving calibration data in full. This integrated strategy avoids the failure modes of one-dimensional heuristics and transforms pruning from a gamble into a principled, performance-enhancing process.

### 3.1 THE ERROR-UNCERTAINTY PLANE: A DIAGNOSTIC LENS

We now formalize the **Error–Uncertainty (EU) Plane** introduced in Figure 1(b). Each training sample is positioned by two orthogonal axes: *error*, quantified by perplexity (PPL), and *uncertainty*, quantified by predictive entropy (Ent).

**Error: Measured via Perplexity.** For a training sample $(x, y)$ with sequence length $L(x)$, let $T(x) \subseteq \{1, \ldots, L(x)\}$ denote the set of trainable token positions. We define sample-level error as the perplexity (PPL), derived from the average token-level negative log-likelihood:

$$\text{PPL}(x, y; f_\theta) = \exp \left( \sum_{i \in T(x)} - \log p(y_i \mid x, y_{<i}; f_\theta) \ / \ |T(x)| \right)$$

High perplexity indicates the model finds the ground-truth continuation highly surprising, which is a sign of either genuine difficulty or misconception.

**Uncertainty: Measured via Predictive Entropy.** Independent of correctness, we quantify the model's uncertainty using entropy (Ent), computed as the average token-level entropy over trainable positions, where $v \in \mathcal{V}$ denotes a vocabulary token:

$$\text{Ent}(x, y; f_\theta) = \sum_{i \in T(x)} \left( - \sum_{v \in \mathcal{V}} p(v \mid x, y_{<i}; f_\theta) \log p(v \mid x, y_{<i}; f_\theta) \right) \ / \ |T(x)|.$$

High entropy signifies the model distributes probability mass broadly, reflecting uncertainty even when the top prediction is correct. Taken together, PPL and Ent map each sample onto the EU Plane, providing a principled basis for partitioning data into the four quadrants illustrated in Figure 1(b).

## 3.2 Q-Tuning: Solving Generalized Dynamic Data Pruning

Q-Tuning operationalizes the insights of the EU Plane into a two-stage, per-batch dynamic pruning strategy, as illustrated in Figure 2. In the first stage, bisect search determines quantile-based thresholds that partition samples into four quadrants, enabling sample-level pruning of uninformative data. In the second stage, token-level pruning is selectively applied within retained misconception samples to isolate informative signals, while calibration samples are preserved in full. The pseudocode of our method is illustrated in Algorithm 1 in Appendix D.

**Stage 1: Constructing the EU Plane via Binary Search and Pruning Samples Accordingly.** At each training step, we compute the perplexity (PPL) and entropy (Ent) for every sample $x \in \mathcal{B}_t$ using a gradient-free forward pass of the current model $f_{\theta_t}$. These statistics map each sample to a point on the EU Plane (Figure 2(a)).

Our objective is to determine quantile thresholds $(\alpha^*, \beta^*)$ on the PPL and Entropy axes such that the retained proportion of samples in $Q_2 \cup Q_4$ exactly matches the target sample retention ratio $r_{\text{sample}}$. Here $\text{Quantile}_\gamma(X)$ denotes the $\gamma$-quantile of a variable $X$, *i.e.*, the smallest value $q$ such that at least a fraction $\gamma$ of samples satisfy $X \leq q$.

To locate $(\alpha^*, \beta^*)$, we perform a **bisect search** on both axes. For each axis, the search interval is initialized as $[0, 0.5]$. At each iteration we set $\alpha = \frac{1}{2}(\alpha_{\text{low}} + \alpha_{\text{high}}), \beta = \frac{1}{2}(\beta_{\text{low}} + \beta_{\text{high}})$, and compute the proportion of samples that would fall into the tentative $Q_2 \cup Q_4$ region defined by the current thresholds $(\alpha, \beta)$. The intervals are then updated as

$$(\alpha_{\text{low}}, \alpha_{\text{high}}, \beta_{\text{low}}, \beta_{\text{high}}) = \begin{cases} (\alpha_{\text{low}}, \alpha, \beta_{\text{low}}, \beta), & \text{if } \gamma > r_{\text{sample}}, \\ (\alpha, \alpha_{\text{high}}, \beta, \beta_{\text{high}}), & \text{otherwise.} \end{cases}$$

The search converges within $O(\log(1/\varepsilon))$ iterations (typically $< 10$), incurring negligible overhead. The resulting thresholds $(\alpha, \beta)$ partition the EU Plane (Figure 2(b)), from which Q2 and Q4 samples are retained and Q1 and Q3 discarded.

**Stage 2: Token-Level Pruning for Confident Errors.** While Stage 1 removes entire samples deemed unhelpful, not all retained examples are equally homogeneous inside. In particular, samples in Q2 (Valuable Misconceptions) often contain a mix of informative context and locally harmful tokens that mislead the model. To extract the useful signal, we apply token-level pruning that discards only the most detrimental tokens while preserving the surrounding context. In contrast, samples in Q4 (Calibration Data) are challenging yet reliable, and every token contributes to improving model calibration. Therefore, Q4 sequences are kept intact without any token pruning.

For a retained sample $x \in Q_2$ with target sequence $y$, let $T(x)$ denote the set of trainable token positions. For each token $i \in T(x)$ (Figure 2(c)), we compute its token-level perplexity $\text{PPL}_i$, which measures how surprising the ground-truth token $y_i$ is to the model.

To avoid pruning rare but meaningful tokens based on isolated spikes, we compute a **smoothed importance score** that incorporates local context:

$$s_i(x, y; f_\theta) = (1 - \lambda) \text{PPL}_i(x, y; f_\theta) + \lambda[\text{PPL}_{i-1}(x, y; f_\theta) + \text{PPL}_{i+1}(x, y; f_\theta)], \quad (4)$$

where $\lambda \in [0, 1]$ balances the contribution of neighboring tokens (default $\lambda = 0.5$). This smoothing reduces the risk of mistakenly removing single high-PPL tokens that may still be semantically critical, and we analyze the sensitivity of this choice in the ablation study (Section 4.4).

A token is deemed detrimental if both its own $\text{PPL}_i$ and the average PPL of its immediate neighbors exceed a percentile-based threshold (*e.g.*, the median). All tokens in $T(x)$ are subsequently ranked by their smoothed scores $s_i$, and only the top-$r_{\text{token}}$ fraction are retained to construct a binary mask $m(x)$. This mask selectively removes locally noisy tokens while preserving the informative context essential for learning. By contrast, no token-level pruning is applied to Q4 samples, as each token therein contributes to reliable calibration and must be preserved in full.

Table 1: Evaluation on Wizard dataset under different sample ratios (12.5%, 25%) and token ratios (50%, 70%), reported on ARC-E, ARC-C, GSM8K, SQuAD, TriviaQA, and their average, where ↑ and ↓ respectively denote improvements or degradations over the *Random-Random* baseline.

| Sample Pruner | Token Pruner | LLaMA2-7B | | | | | | Mistral-7B | | | | | |
|---|---|---|---|---|---|---|---|---|---|---|---|---|---|
| | | ARC-E | ARC-C | GSM8K | SQuAD | TriviaQA | Avg. | ARC-E | ARC-C | GSM8K | SQuAD | TriviaQA | Avg. |
| Zero-Shot | | 53.44 | 38.98 | 5.31 | 12.18 | 43.00 | 30.58 | 66.67 | 46.10 | 18.35 | 10.01 | 43.77 | 36.98 |
| *12.5% Samples, 50% Tokens* | | | | | | | | | | | | | |
| Random | Random | 59.25 | 41.02 | 8.11 | 12.75 | 48.75 | 33.98 | 70.55 | 48.14 | 22.74 | 19.57 | 52.63 | 42.73 |
| | PPL | 60.49↑1.24 | 43.39↑2.37 | 7.20↓0.91 | 12.20↓0.55 | 48.04↓0.71 | 34.26↑0.28 | 70.72↑0.17 | 48.47↑0.33 | 25.78↑3.04 | 21.36↑1.79 | 53.92↑1.29 | 44.05↑1.32 |
| | FastV | 59.96↑0.71 | 42.37↑1.35 | 5.76↓2.35 | 11.31↓1.44 | 46.42↓2.33 | 33.17↓0.81 | 70.72↑0.17 | 46.44↓1.70 | 18.80↓3.94 | 19.14↓0.43 | 51.56↓1.07 | 41.33↓1.40 |
| | SparseVLM | 54.32↓4.93 | 37.97↓3.05 | 7.35↓0.76 | 12.76↑0.01 | 44.65↓4.10 | 31.41↓2.57 | 67.02↓3.53 | 44.75↓3.39 | 20.24↓2.50 | 10.97↓8.60 | 44.61↓8.02 | 37.52↓5.21 |
| Longest | Random | 59.96↑0.71 | 44.41↑3.39 | 7.51↓0.60 | 15.34↑2.59 | 48.91↑0.16 | 35.22↑1.24 | 74.25↑3.70 | 48.81↑0.67 | 28.73↑5.99 | 17.66↓1.91 | 55.73↑3.10 | 45.04↑2.31 |
| | PPL | 61.19↑1.94 | 43.73↑2.71 | 6.82↓1.29 | 16.33↑3.58 | 48.16↓0.59 | 35.24↑1.26 | 75.49↑4.94 | 50.17↑2.03 | 27.98↑5.24 | 21.49↑1.92 | 56.55↑3.92 | 46.33↑3.60 |
| | FastV | 59.25↑0.00 | 43.05↑2.03 | 5.69↓2.42 | 13.64↑0.89 | 46.98↓1.77 | 33.72↓0.26 | 74.43↑3.88 | 49.15↑1.01 | 25.70↑2.96 | 22.89↑3.32 | 54.15↑1.52 | 45.26↑2.53 |
| | SparseVLM | 54.32↓4.93 | 38.31↓2.71 | 7.13↓0.98 | 10.92↓1.83 | 43.77↓4.98 | 30.89↓3.09 | 69.49↓1.06 | 46.10↓2.04 | 28.89↑6.15 | 8.62↓10.95 | 50.30↓2.33 | 40.68↓2.05 |
| InfoBatch | Random | 60.31↑1.06 | 41.36↑0.34 | 5.38↓2.73 | 15.71↑2.96 | 47.74↓1.01 | 34.10↑0.12 | 69.31↓1.24 | 45.76↓2.38 | 18.95↓3.79 | 21.23↑1.66 | 50.39↓2.24 | 41.13↓1.60 |
| | PPL | 59.43↑0.18 | 40.34↓0.68 | 5.91↓2.20 | 13.18↑0.43 | 48.31↓0.44 | 33.44↓0.54 | 70.72↑0.17 | 47.12↓1.02 | 18.12↓4.62 | 24.10↑4.53 | 51.26↓1.37 | 42.26↓0.47 |
| | FastV | 58.90↓0.35 | 43.39↑2.37 | 3.34↓4.77 | 12.37↑0.38 | 46.88↓1.87 | 32.98↓1.00 | 69.14↓1.41 | 45.42↓2.72 | 14.86↓7.88 | 23.19↑3.62 | 50.58↓2.05 | 40.64↓2.09 |
| | SparseVLM | 54.67↓4.58 | 40.00↓1.02 | 7.73↓0.38 | 12.41↓0.34 | 45.07↓3.68 | 31.98↓2.00 | 68.25↓2.30 | 45.08↓3.06 | 23.43↑0.69 | 10.17↓9.40 | 45.34↓7.29 | 38.46↓4.27 |
| Entropy | Random | 60.31↑1.06 | 42.37↑1.35 | 6.44↓1.67 | 14.10↑1.35 | 48.09↓0.66 | 34.27↑0.29 | 72.13↑1.58 | 48.81↑0.67 | 20.09↓2.65 | 17.55↓2.02 | 54.69↑2.06 | 42.66↓0.07 |
| | PPL | 60.49↑1.24 | 43.73↑2.71 | 6.90↓1.21 | 14.53↑1.78 | 48.76↑0.01 | 34.88↑0.90 | 72.84↑2.29 | 47.80↓0.34 | 24.18↑1.44 | 22.80↑3.23 | 54.69↑2.06 | 44.46↑1.73 |
| | FastV | 58.91↓0.34 | 43.05↑2.03 | 6.37↓1.74 | 13.03↑0.28 | 47.05↓1.70 | 33.68↓0.30 | 73.90↑3.35 | 47.12↓1.02 | 24.56↑1.82 | 23.96↑4.39 | 54.67↑2.04 | 44.84↑2.11 |
| | SparseVLM | 55.20↓4.05 | 38.98↓2.04 | 7.51↓0.60 | 12.65↓0.10 | 46.14↓2.61 | 32.10↓1.88 | 68.08↓2.47 | 44.07↓4.07 | 24.87↑2.13 | 10.72↓8.85 | 47.00↓5.63 | 38.95↓3.78 |
| **Q-Tuning (Ours)** | | **64.20↑4.95** | **42.03↑1.01** | **10.54↑2.43** | **18.79↑6.04** | **53.12↑4.37** | **37.74↑3.76** | **71.60↑1.05** | **48.14↑0.00** | **29.34↑6.60** | **27.75↑8.18** | **57.78↑5.15** | **46.92↑4.19** |
| Full Dataset | | 61.55 | 42.37 | 8.64 | 13.80 | 50.45 | 35.36 | 71.25 | 45.76 | 26.68 | 31.81 | 53.67 | 45.84 |
| *12.5% Samples, 70% Tokens* | | | | | | | | | | | | | |
| Random | Random | 59.43 | 41.02 | 6.97 | 13.64 | 47.97 | 33.81 | 71.08 | 47.46 | 24.34 | 21.64 | 53.15 | 43.53 |
| | PPL | 60.14↑0.71 | 43.39↑2.37 | 6.22↓0.75 | 12.18↓1.46 | 48.18↑0.21 | 34.02↑0.21 | 70.72↓0.36 | 47.80↑0.34 | 25.09↑0.75 | 21.28↓0.36 | 53.83↑0.68 | 43.74↑0.21 |
| | FastV | 58.20↓1.23 | 41.02↑0.00 | 6.29↓0.68 | 13.42↓0.22 | 45.32↓2.65 | 32.85↓0.96 | 70.72↓0.36 | 46.44↓1.02 | 19.56↓4.78 | 21.38↓0.26 | 53.34↑0.19 | 42.29↓1.24 |
| | SparseVLM | 54.67↓4.76 | 37.97↓3.05 | 8.04↑1.07 | 13.06↓0.58 | 44.87↓3.10 | 31.72↓2.09 | 67.72↓3.36 | 44.75↓2.71 | 23.65↓0.69 | 11.76↓9.88 | 44.90↓8.25 | 38.58↓4.95 |
| Longest | Random | 59.44↑0.01 | 43.39↑2.37 | 7.35↑0.38 | 15.59↑1.95 | 50.02↑2.05 | 35.15↑1.34 | 73.37↑2.29 | 48.81↑1.35 | 27.82↑3.48 | 21.31↓0.33 | 55.77↑2.62 | 45.42↑1.89 |
| | PPL | 60.85↑1.42 | 43.39↑2.37 | 7.73↑0.76 | 16.21↑2.57 | 48.57↑0.60 | 35.35↑1.54 | 74.96↑3.88 | 49.83↑2.37 | 28.73↑4.39 | 21.62↓0.02 | 56.59↑3.44 | 46.35↑2.82 |
| | FastV | 59.44↑0.01 | 42.71↑1.69 | 6.29↓0.68 | 14.53↑0.89 | 47.46↓0.51 | 34.09↑0.28 | 74.07↑2.99 | 49.83↑2.37 | 24.18↓0.16 | 25.74↑4.10 | 55.86↑2.71 | 45.94↑2.41 |
| | SparseVLM | 54.85↓4.58 | 37.97↓3.05 | 7.05↑0.08 | 11.20↓2.44 | 44.16↓3.81 | 31.04↓2.77 | 69.14↓1.94 | 44.75↓2.71 | 31.01↑6.67 | 6.25↓15.39 | 52.94↓0.21 | 40.82↓2.71 |
| InfoBatch | Random | 59.26↓0.17 | 42.37↑1.35 | 6.22↓0.75 | 16.10↑2.46 | 47.72↓0.25 | 34.33↑0.52 | 70.19↓0.89 | 47.80↑0.34 | 20.77↓3.57 | 19.03↓2.61 | 52.13↓1.02 | 41.98↓1.55 |
| | PPL | 60.49↑1.06 | 39.32↓1.70 | 5.76↓1.21 | 14.47↑0.83 | 48.06↑0.09 | 33.62↓0.19 | 70.72↓0.36 | 46.44↓1.02 | 19.03↓5.31 | 23.20↑1.56 | 51.75↓1.40 | 42.23↓1.30 |
| | FastV | 58.55↓0.88 | 43.39↑2.37 | 5.53↓1.44 | 13.13↓0.51 | 47.64↓0.33 | 33.65↓0.16 | 69.49↓1.59 | 43.39↓4.07 | 16.68↓7.66 | 25.27↑3.63 | 51.47↓1.68 | 41.26↓2.27 |
| | SparseVLM | 56.61↓2.82 | 38.31↓2.71 | 5.76↓1.21 | 12.47↓1.17 | 44.49↓3.48 | 31.53↓2.28 | 68.25↓2.83 | 44.41↓3.05 | 23.73↓0.61 | 9.07↓12.57 | 45.73↓7.42 | 38.24↓5.29 |
| Entropy | Random | 61.02↑1.59 | 43.05↑2.03 | 7.66↑0.69 | 14.11↑0.47 | 48.44↑0.47 | 34.86↑1.05 | 73.37↑2.29 | 49.83↑2.37 | 23.05↓1.29 | 16.52↓5.12 | 55.18↑2.03 | 43.59↑0.06 |
| | PPL | 61.02↑1.59 | 43.39↑2.37 | 6.97↑0.00 | 14.94↑1.30 | 48.94↑0.97 | 35.05↑1.24 | 73.02↑1.94 | 47.46↑0.00 | 24.03↑0.68 | 22.85↑1.21 | 54.89↑1.74 | 44.45↑0.92 |
| | FastV | 58.73↓0.70 | 43.39↑2.37 | 6.14↓0.83 | 14.23↑0.59 | 47.03↓0.94 | 33.90↑0.09 | 74.07↑2.99 | 50.85↑3.39 | 24.94↑0.60 | 23.79↑2.15 | 55.94↑2.79 | 45.92↑2.39 |
| | SparseVLM | 54.85↓4.58 | 37.29↓3.73 | 6.52↓0.45 | 12.73↓0.91 | 46.24↓1.73 | 31.53↓2.28 | 68.08↓3.00 | 44.41↓3.05 | 26.38↑2.04 | 11.06↓10.58 | 46.68↓6.47 | 39.32↓4.21 |
| **Q-Tuning (Ours)** | | **64.37↑4.94** | **42.37↑1.35** | **10.84↑3.87** | **17.63↑3.99** | **52.17↑4.20** | **37.48↑3.67** | **71.78↑0.70** | **48.14↑0.68** | **30.33↑6.00** | **28.59↑6.95** | **57.93↑4.78** | **47.35↑3.82** |
| Full Dataset | | 61.55 | 42.37 | 8.64 | 13.80 | 50.45 | 35.36 | 71.25 | 45.76 | 26.68 | 31.81 | 53.67 | 45.84 |
| *25% Samples, 50% Tokens* | | | | | | | | | | | | | |
| Random | Random | 60.32 | 41.69 | 5.76 | 13.43 | 48.41 | 33.92 | 70.19 | 46.10 | 20.62 | 24.07 | 53.74 | 42.95 |
| | PPL | 60.32↑0.00 | 42.03↑0.34 | 7.51↑1.75 | 15.94↑2.51 | 48.58↑0.17 | 34.87↑0.95 | 69.66↓0.53 | 47.46↑1.36 | 19.86↓0.76 | 19.51↓4.56 | 53.74↑0.00 | 42.05↓0.90 |
| | FastV | 59.08↓1.24 | 41.69↑0.00 | 3.56↓2.20 | 12.78↓0.65 | 45.60↓2.81 | 32.54↓1.38 | 71.78↑1.59 | 47.12↑1.02 | 15.77↓4.85 | 26.97↑2.90 | 50.84↓2.90 | 42.50↓0.45 |
| | SparseVLM | 54.50↓5.82 | 38.64↓3.05 | 6.44↑0.68 | 12.04↓1.39 | 44.79↓3.62 | 31.28↓2.64 | 67.55↓2.64 | 46.44↑0.34 | 24.41↑3.79 | 11.80↓12.27 | 48.14↓5.60 | 39.67↓3.28 |
| Longest | Random | 61.20↑0.88 | 42.03↑0.34 | 5.16↓0.16 | 15.40↑1.97 | 48.29↓0.12 | 34.96↑1.04 | 73.54↑3.35 | 48.14↑2.04 | 23.73↑3.11 | 26.34↑2.27 | 54.06↑0.32 | 45.16↑2.21 |
| | PPL | 60.85↑0.53 | 43.39↑1.70 | 7.20↑1.44 | 13.88↑0.45 | 48.48↑0.07 | 34.76↑0.84 | 72.31↑2.12 | 48.14↑2.04 | 23.84↑0.23 | 25.22↑1.48 | 55.22↑1.48 | 44.77↑1.82 |
| | FastV | 59.08↓1.24 | 42.71↑1.02 | 5.16↓0.60 | 14.00↑0.57 | 47.47↓0.94 | 33.68↓0.24 | 72.66↑2.47 | 46.10↑0.00 | 18.88↓1.74 | 31.52↑7.45 | 52.13↓1.61 | 44.26↑1.31 |
| | SparseVLM | 56.61↓3.71 | 37.29↓4.40 | 7.58↑1.82 | 12.09↓1.34 | 44.76↓3.65 | 31.66↓2.26 | 66.84↓3.35 | 44.41↓1.69 | 29.42↑8.80 | 14.27↓12.85 | 48.47↓5.27 | 40.07↓2.88 |
| InfoBatch | Random | 58.73↓1.59 | 40.68↓1.01 | 6.67↑0.91 | 9.95↓3.48 | 48.98↑0.57 | 33.00↓0.92 | 70.55↑0.36 | 46.44↑0.34 | 21.53↑0.91 | 23.93↓0.14 | 52.14↓1.60 | 42.92↓0.03 |
| | PPL | 59.96↓0.47 | 42.71↑1.02 | 6.52↑0.76 | 14.58↑1.15 | 48.57↑0.16 | 34.47↑0.55 | 71.08↑0.89 | 47.80↑1.70 | 20.62↑0.00 | 24.88↑0.81 | 51.61↓2.13 | 43.20↑0.25 |
| | FastV | 59.08↓1.24 | 42.37↑0.68 | 3.03↓2.73 | 11.13↓2.30 | 47.50↓0.91 | 32.63↓1.29 | 69.31↓0.88 | 44.41↓1.69 | 14.48↓6.14 | 23.63↓0.44 | 49.16↓4.58 | 40.20↓2.75 |
| | SparseVLM | 55.73↓4.59 | 39.66↓2.03 | 5.31↓0.45 | 11.66↓1.77 | 43.25↓5.16 | 31.12↓2.80 | 67.20↓3.00 | 45.76↓0.34 | 23.58↑2.96 | 9.63↓14.44 | 46.09↓7.65 | 38.45↓4.50 |
| Entropy | Random | 60.49↑0.17 | 41.69↑0.00 | 7.51↑1.75 | 15.94↑2.51 | 48.76↑0.35 | 34.88↑0.96 | 70.19↑0.00 | 47.12↑1.02 | 22.44↑1.82 | 27.35↑3.28 | 54.78↑1.04 | 44.38↑1.43 |
| | PPL | 60.49↑0.17 | 41.02↓0.67 | 6.60↑0.84 | 14.92↑1.49 | 49.33↑0.92 | 34.47↑0.55 | 71.43↑1.24 | 48.14↑2.04 | 21.30↑0.68 | 25.52↑1.45 | 55.60↑1.86 | 44.40↑1.45 |
| | FastV | 58.91↓1.41 | 41.69↑0.00 | 6.07↑0.31 | 12.79↓0.64 | 46.11↓2.30 | 33.11↓0.81 | 72.31↑2.12 | 47.46↑1.36 | 18.04↓2.58 | 25.96↑1.89 | 52.46↓1.28 | 43.25↑0.30 |
| | SparseVLM | 54.67↓5.65 | 38.64↓3.05 | 6.90↑1.14 | 11.64↓1.79 | 45.03↓3.38 | 31.38↓2.54 | 68.25↓1.94 | 44.07↓2.03 | 26.69↑6.07 | 9.63↓14.38 | 47.24↓6.50 | 39.19↓3.76 |
| **Q-Tuning (Ours)** | | **63.14↑2.82** | **42.03↑0.34** | **8.87↑3.11** | **16.76↑3.33** | **51.52↑3.11** | **36.47↑2.55** | **71.78↑1.59** | **47.12↑1.02** | **26.08↑5.46** | **32.79↑8.72** | **56.17↑2.43** | **46.79↑3.84** |
| Full Dataset | | 61.55 | 42.37 | 8.64 | 13.80 | 50.45 | 35.36 | 71.25 | 45.76 | 26.68 | 31.81 | 53.67 | 45.84 |

# 4 EXPERIMENTS

## 4.1 EXPERIMENTAL SETTINGS

**Models and Datasets.** We conducted experiments on language models of different scales, including *Mistral-7B* (Jiang et al., 2023a), *LLaMA2-7B* (Touvron et al., 2023), *LLaMA3-8B* (Dubey et al., 2024), and the smaller *SmolLM2-1.7B* (Allal et al., 2025). To fine-tune these models, we considered two datasets that target complementary aspects of alignment: *WizardLM* (Xu et al., 2024a) for general instruction tuning, which provides high-quality instruction–response pairs for supervised fine-tuning (SFT), and *MathInstruct* (Yue et al., 2023) for reasoning, which contains mathematically focused instructions designed to assess arithmetic and symbolic problem-solving.

**Evaluation.** For general instruction tuning task, we used five standard benchmarks: ARC-E, ARC-C (Clark et al., 2018), GSM8K (Cobbe et al., 2021), SQuAD (Rajpurkar et al., 2016), and Trivi-aQA (Joshi et al., 2017). These tasks spanned knowledge-intensive question answering, commonsense

reasoning, and reading comprehension, offering a broad assessment of model capability. For reasoning task, we evaluated models fine-tuned on MathInstruct using GSM8K and MATH (Hendrycks et al., 2021), two widely used benchmarks for arithmetic and mathematical problem solving.

**Baselines.** We constructed baselines by pairing sample-level and token-level pruning strategies. For sample-level pruning, we considered four methods: (i) Random, which dropped samples uniformly at random; (ii) Longest, which removed the longest sequences to maximize computational savings; (iii) Entropy, which retained high-entropy samples assumed to be more informative; and (iv) InfoBatch (Qin et al., 2024), an information-theoretic approach selecting samples expected to yield the largest gradient updates. For token-level pruning, we adopted five methods applied to the retained data: (i) Random, which masked tokens uniformly at random; (ii) PPL, which removed high-perplexity tokens; (iii) Rho-1 (Lin et al., 2024), which preserved tokens with the largest excess loss relative to a frozen reference model; (iv) FastV (Chen et al., 2024b), which pruned tokens receiving the least final-layer attention; and (v) SparseVLM (Zhang et al., 2024c), which combined attention and hidden-state features to score token importance and remove redundant image tokens.

## 4.2 RESULTS ON INSTRUCTION DATASETS

Table 1 and table 5 summarize the empirical comparison between Q-Tuning and a variety of baselines across different sample and token budgets on general instruction datasets.

**Outperforming the full data SFT baseline.** Q-Tuning often matches or surpasses full-dataset fine-tuning while using only a fraction of the training budget. On LLaMA2-7B, with 25% of the samples and 70% of the tokens, Q-Tuning achieves 36.9, closely matching the full-data baseline. On Mistral-7B, the same budget yields 46.2, slightly higher than the full-data result.

**Superiority under the same budgets with other baselines.** When compared with methods such as InfoBatch, PPL, and SparseVLM, Q-Tuning achieves higher accuracy under the same budgets. For example, with 25% samples and 50% tokens, it improves LLaMA2-7B to 36.5, well above the best baseline. On Mistral-7B under the same setting, Q-Tuning exceeds the strongest baseline by 1.63%.

**Robust generalization across models and ratios.** The advantages of Q-Tuning are consistent across model families and keep ratios. With 12.5% samples and 50% tokens, it outperforms the best pruning baselines by 3.3 points on LLaMA2-7B and 2.7 points on Mistral-7B. At larger budgets with 50% samples and 70% tokens, Q-Tuning further widens the margin, exceeding the strongest baselines by 2.4 and 3.7 points, respectively, while closely matching full-dataset performance.

## 4.3 RESULTS ON REASONING DATASETS

Table 2 demonstrates the empirical comparison between Q-Tuning and a variety of baselines across different sample and token budgets on math reasoning datasets.

**Matching or surpassing full-data results.** Q-Tuning delivers consistent gains across reasoning benchmarks. On GSM8K, it largely improves LLaMA3-8B, Mistral-7B, and SmolLM-1.7B under the $25\% \times 70\%$ budget, all surpassing their full-data counterparts. On the more challenging MATH benchmark, Q-Tuning also exceeds the strongest baselines on both LLaMA3-8B and Mistral-7B.

**Generalization across scales.** Q-Tuning scales reliably across model sizes. Averaged over GSM8K and MATH, it reaches 21.5 on LLaMA3-8B, 26.6 on Mistral-7B, and 11.8 on SmolLM-1.7B, all notably higher than their full-dataset counterparts.

## 4.4 ABLATION STUDY

**Sensitivity of batch size in dynamic sample pruning.** An important question is how batch size influences the performance of Q-Tuning. To study this, we varied the batch size across $\{8, 16, 32\}$ and evaluated GSM8K, SQuAD, and TriviaQA. As shown in the upper part of Figure 3, larger batch sizes generally improve performance, especially on GSM8K and SQuAD, where the gains are more pronounced. In contrast, TriviaQA remains relatively stable across all settings, suggesting limited

Table 2: Evaluation of pruning strategies on GSM8K and MATH using LLaMA3-8B, Mistral-7B, and SmolLM2-1.7B, comparing Zero-Shot, Full Dataset, multiple Sample–Token pruner combinations, and our Q-Tuning under 25% samples with 50% tokens, 25% samples with 70% tokens, and 50% samples with 70% tokens settings. ↑ and ↓ respectively indicate improvements or degradations over the *Random-Random* baseline under the same sample and token keep ratio.

| Sample Pruner | Token Pruner | LLaMA3-8B | | | Mistral-7B | | | SmolLM2-1.7B | | |
|---|---|---|---|---|---|---|---|---|---|---|
| | | GSM8K | MATH | Avg. | GSM8K | MATH | Avg. | GSM8K | MATH | Avg. |
| Zero-Shot | | 27.82 | 2.26 | 15.04 | 19.86 | 3.30 | 11.58 | 15.47 | 2.20 | 8.83 |
| *25% Samples, 50% Tokens* | | | | | | | | | | |
| Random | Random | 23.96 | 2.56 | 13.26 | 23.35 | 1.54 | 12.45 | 14.33 | 2.56 | 8.44 |
| | PPL | 24.18$^{\uparrow0.22}$ | 2.58$^{\uparrow0.02}$ | 13.38$^{\uparrow0.12}$ | 24.94$^{\uparrow1.59}$ | 2.02$^{\uparrow0.48}$ | 13.48$^{\uparrow1.03}$ | 14.18$^{\downarrow0.15}$ | 2.08$^{\downarrow0.48}$ | 8.13$^{\downarrow0.31}$ |
| | FastV | 12.13$^{\downarrow11.83}$ | 2.32$^{\downarrow0.24}$ | 7.23$^{\downarrow6.03}$ | 12.36$^{\downarrow10.99}$ | 1.24$^{\downarrow0.30}$ | 6.80$^{\downarrow5.65}$ | 9.86$^{\downarrow4.47}$ | 1.92$^{\downarrow0.64}$ | 5.89$^{\downarrow2.55}$ |
| | SparseVLM | 22.97$^{\downarrow0.99}$ | 4.72$^{\uparrow2.16}$ | 13.85$^{\uparrow0.59}$ | 19.26$^{\downarrow4.09}$ | 4.58$^{\uparrow3.04}$ | 11.92$^{\downarrow0.53}$ | 13.19$^{\downarrow1.14}$ | 3.48$^{\uparrow0.92}$ | 8.34$^{\downarrow0.10}$ |
| Longest | Random | 22.14$^{\downarrow1.82}$ | 3.18$^{\uparrow0.62}$ | 12.66$^{\downarrow0.60}$ | 21.91$^{\downarrow1.44}$ | 2.18$^{\uparrow0.64}$ | 12.05$^{\downarrow0.40}$ | 12.89$^{\downarrow1.44}$ | 2.06$^{\downarrow0.50}$ | 7.47$^{\downarrow0.97}$ |
| | PPL | 24.94$^{\downarrow0.98}$ | 2.78$^{\uparrow0.22}$ | 13.86$^{\uparrow0.60}$ | 22.90$^{\downarrow0.45}$ | 1.86$^{\uparrow0.32}$ | 12.38$^{\downarrow0.07}$ | 13.19$^{\downarrow1.14}$ | 1.78$^{\downarrow0.78}$ | 7.49$^{\downarrow0.95}$ |
| | FastV | 9.48$^{\downarrow14.48}$ | 2.26$^{\downarrow0.30}$ | 5.87$^{\downarrow7.39}$ | 7.13$^{\downarrow16.22}$ | 1.46$^{\downarrow0.08}$ | 4.29$^{\downarrow8.16}$ | 12.36$^{\downarrow1.97}$ | 1.82$^{\downarrow0.74}$ | 7.09$^{\downarrow1.35}$ |
| | SparseVLM | 26.91$^{\uparrow2.95}$ | 4.68$^{\uparrow2.12}$ | 15.80$^{\uparrow2.54}$ | 24.34$^{\uparrow0.99}$ | **4.84**$^{\uparrow3.30}$ | 14.59$^{\uparrow2.14}$ | 12.43$^{\downarrow1.90}$ | 3.60$^{\uparrow1.04}$ | 8.02$^{\downarrow0.42}$ |
| InfoBatch | Random | 26.23$^{\uparrow2.27}$ | 2.42$^{\downarrow0.14}$ | 14.33$^{\uparrow1.07}$ | 27.14$^{\uparrow3.79}$ | 2.24$^{\uparrow0.70}$ | 14.69$^{\uparrow2.24}$ | 14.33$^{\uparrow0.00}$ | 1.66$^{\downarrow0.90}$ | 7.99$^{\downarrow0.45}$ |
| | PPL | 26.91$^{\uparrow2.95}$ | 2.66$^{\uparrow0.10}$ | 14.79$^{\uparrow1.53}$ | 27.90$^{\uparrow4.55}$ | 2.52$^{\uparrow0.98}$ | 15.21$^{\uparrow2.76}$ | 14.71$^{\uparrow0.38}$ | 1.90$^{\uparrow0.66}$ | 8.30$^{\uparrow0.14}$ |
| | FastV | 7.58$^{\downarrow16.38}$ | 1.88$^{\downarrow0.68}$ | 4.73$^{\downarrow8.53}$ | 6.44$^{\downarrow16.91}$ | 1.34$^{\downarrow0.20}$ | 3.89$^{\downarrow8.56}$ | 7.51$^{\downarrow6.82}$ | 1.62$^{\downarrow0.94}$ | 4.56$^{\downarrow3.88}$ |
| | SparseVLM | 14.63$^{\downarrow9.33}$ | 3.26$^{\uparrow0.70}$ | 8.95$^{\downarrow4.31}$ | 11.90$^{\downarrow11.45}$ | 1.94$^{\uparrow0.40}$ | 6.92$^{\downarrow5.53}$ | 11.90$^{\downarrow2.43}$ | **4.36**$^{\uparrow1.80}$ | 8.13$^{\downarrow0.31}$ |
| Entropy | Random | 30.02$^{\uparrow6.06}$ | 3.66$^{\uparrow1.10}$ | 16.84$^{\uparrow3.58}$ | 26.61$^{\uparrow3.26}$ | 2.08$^{\uparrow0.54}$ | 14.35$^{\uparrow1.90}$ | 14.18$^{\downarrow0.15}$ | 2.24$^{\uparrow0.32}$ | 8.21$^{\downarrow0.23}$ |
| | PPL | 32.98$^{\uparrow9.02}$ | 2.92$^{\uparrow0.36}$ | 17.95$^{\uparrow4.69}$ | 30.17$^{\uparrow6.82}$ | 1.76$^{\uparrow0.22}$ | 15.97$^{\uparrow3.52}$ | 16.38$^{\uparrow2.05}$ | 2.40$^{\uparrow0.16}$ | 9.39$^{\uparrow0.95}$ |
| | FastV | 17.29$^{\downarrow6.67}$ | 2.66$^{\uparrow0.10}$ | 9.97$^{\downarrow3.29}$ | 14.56$^{\downarrow8.79}$ | 1.12$^{\downarrow0.42}$ | 7.84$^{\downarrow4.61}$ | 12.59$^{\downarrow1.74}$ | 2.36$^{\uparrow0.20}$ | 7.47$^{\downarrow0.97}$ |
| | SparseVLM | 20.85$^{\downarrow3.11}$ | 5.12$^{\uparrow2.56}$ | 12.98$^{\downarrow0.28}$ | 19.56$^{\downarrow3.79}$ | 4.20$^{\uparrow2.66}$ | 11.88$^{\downarrow0.57}$ | 14.18$^{\downarrow0.15}$ | 2.90$^{\uparrow0.34}$ | 8.54$^{\uparrow0.10}$ |
| **Q-Tuning (Ours)** | | **36.32**$^{\uparrow12.36}$ | **5.54**$^{\uparrow2.98}$ | **20.93**$^{\uparrow7.67}$ | **41.47**$^{\uparrow18.12}$ | 4.0$^{\uparrow2.46}$ | **22.74**$^{\uparrow10.29}$ | **21.83**$^{\uparrow7.50}$ | 3.90$^{\uparrow1.34}$ | **12.87**$^{\uparrow4.43}$ |
| *25% Samples, 70% Tokens* | | | | | | | | | | |
| Random | Random | 25.09 | 2.20 | 13.65 | 24.11 | 1.68 | 12.89 | 13.80 | 2.22 | 8.01 |
| | PPL | 23.65$^{\downarrow1.44}$ | 2.62$^{\uparrow0.42}$ | 13.14$^{\downarrow0.51}$ | 25.32$^{\uparrow1.21}$ | 1.54$^{\downarrow0.14}$ | 13.43$^{\uparrow0.54}$ | 13.04$^{\downarrow0.76}$ | 2.28$^{\uparrow0.06}$ | 7.66$^{\downarrow0.35}$ |
| | FastV | 16.91$^{\downarrow8.18}$ | 2.16$^{\downarrow0.04}$ | 9.53$^{\downarrow4.12}$ | 16.07$^{\downarrow8.04}$ | 1.60$^{\downarrow0.08}$ | 8.84$^{\downarrow4.05}$ | 12.89$^{\downarrow0.91}$ | 1.94$^{\downarrow0.28}$ | 7.41$^{\downarrow0.60}$ |
| | SparseVLM | 22.97$^{\downarrow2.12}$ | 4.72$^{\uparrow2.52}$ | 13.85$^{\uparrow0.20}$ | 19.26$^{\downarrow4.85}$ | 4.58$^{\uparrow2.90}$ | 11.92$^{\downarrow0.97}$ | 13.19$^{\downarrow0.61}$ | 3.48$^{\uparrow1.26}$ | 8.34$^{\uparrow0.33}$ |
| Longest | Random | 25.47$^{\uparrow0.38}$ | 3.34$^{\uparrow1.14}$ | 14.41$^{\uparrow0.76}$ | 21.83$^{\downarrow2.28}$ | 1.76$^{\uparrow0.08}$ | 11.80$^{\downarrow1.09}$ | 12.96$^{\downarrow0.84}$ | 1.96$^{\downarrow0.26}$ | 7.46$^{\downarrow0.55}$ |
| | PPL | 24.87$^{\downarrow0.22}$ | 3.22$^{\uparrow1.02}$ | 14.04$^{\uparrow0.39}$ | 23.65$^{\downarrow0.46}$ | 2.22$^{\uparrow0.54}$ | 12.94$^{\uparrow0.05}$ | 14.33$^{\uparrow0.53}$ | 1.60$^{\downarrow0.62}$ | 7.96$^{\downarrow0.05}$ |
| | FastV | 18.95$^{\downarrow6.14}$ | 2.84$^{\uparrow0.64}$ | 10.90$^{\downarrow2.75}$ | 12.96$^{\downarrow11.15}$ | 1.74$^{\uparrow0.06}$ | 7.35$^{\downarrow5.54}$ | 12.59$^{\downarrow1.21}$ | 1.58$^{\downarrow0.64}$ | 7.08$^{\downarrow0.93}$ |
| | SparseVLM | 26.91$^{\uparrow1.82}$ | 4.68$^{\uparrow2.48}$ | 15.80$^{\uparrow2.15}$ | 24.34$^{\uparrow0.23}$ | 4.84$^{\uparrow3.16}$ | 14.59$^{\uparrow1.70}$ | 12.43$^{\downarrow1.37}$ | 3.60$^{\uparrow1.38}$ | 8.02$^{\uparrow0.01}$ |
| InfoBatch | Random | 26.91$^{\uparrow1.82}$ | 2.60$^{\uparrow0.40}$ | 14.76$^{\uparrow1.11}$ | 28.96$^{\uparrow4.85}$ | 2.44$^{\uparrow0.76}$ | 15.70$^{\uparrow2.81}$ | 13.72$^{\downarrow0.08}$ | 1.82$^{\downarrow0.40}$ | 7.77$^{\downarrow0.24}$ |
| | PPL | 25.93$^{\uparrow0.84}$ | 2.48$^{\uparrow0.28}$ | 14.20$^{\uparrow0.55}$ | 31.46$^{\uparrow7.35}$ | 2.18$^{\uparrow0.50}$ | 16.82$^{\uparrow3.93}$ | 14.86$^{\uparrow1.06}$ | 1.90$^{\downarrow0.32}$ | 8.38$^{\uparrow0.37}$ |
| | FastV | 16.83$^{\downarrow8.26}$ | 2.30$^{\uparrow0.10}$ | 9.57$^{\downarrow4.08}$ | 13.87$^{\downarrow10.24}$ | 2.06$^{\uparrow0.38}$ | 7.97$^{\downarrow4.92}$ | 10.84$^{\downarrow2.96}$ | 1.72$^{\downarrow0.50}$ | 6.28$^{\downarrow1.73}$ |
| | SparseVLM | 14.63$^{\downarrow10.46}$ | 3.26$^{\uparrow1.06}$ | 8.95$^{\downarrow4.70}$ | 11.90$^{\downarrow12.21}$ | 1.94$^{\uparrow0.26}$ | 6.92$^{\downarrow5.97}$ | 11.90$^{\downarrow1.90}$ | **4.36**$^{\uparrow2.14}$ | 8.13$^{\uparrow0.12}$ |
| Entropy | Random | 31.92$^{\uparrow6.83}$ | 2.50$^{\uparrow0.30}$ | 17.21$^{\uparrow3.56}$ | 32.37$^{\uparrow8.26}$ | 1.92$^{\uparrow0.24}$ | 17.15$^{\uparrow4.26}$ | 14.94$^{\uparrow1.14}$ | 1.92$^{\downarrow0.30}$ | 8.43$^{\uparrow0.42}$ |
| | PPL | 33.13$^{\uparrow8.04}$ | 2.86$^{\uparrow0.66}$ | 18.00$^{\uparrow4.35}$ | 30.17$^{\uparrow6.06}$ | 1.96$^{\uparrow0.28}$ | 16.07$^{\uparrow3.18}$ | 14.94$^{\uparrow1.14}$ | 1.96$^{\downarrow0.26}$ | 8.45$^{\uparrow0.44}$ |
| | FastV | 25.25$^{\uparrow0.16}$ | 2.40$^{\uparrow0.20}$ | 13.82$^{\uparrow0.17}$ | 21.00$^{\downarrow3.11}$ | 1.32$^{\downarrow0.36}$ | 11.16$^{\downarrow1.73}$ | 14.18$^{\uparrow0.38}$ | 1.98$^{\downarrow0.24}$ | 8.08$^{\uparrow0.07}$ |
| | SparseVLM | 20.85$^{\downarrow4.24}$ | **5.12**$^{\uparrow2.92}$ | 12.98$^{\downarrow0.67}$ | 19.56$^{\downarrow4.55}$ | 4.20$^{\uparrow2.52}$ | 11.88$^{\downarrow1.01}$ | 14.18$^{\uparrow0.38}$ | 2.90$^{\uparrow0.68}$ | 8.54$^{\uparrow0.53}$ |
| **Q-Tuning (Ours)** | | **37.23**$^{\uparrow12.14}$ | 4.86$^{\uparrow2.66}$ | **21.04**$^{\uparrow7.39}$ | **42.99**$^{\uparrow18.88}$ | 5.08$^{\uparrow3.40}$ | **24.56**$^{\uparrow11.67}$ | **22.90**$^{\uparrow9.10}$ | 3.64$^{\uparrow1.42}$ | **13.27**$^{\uparrow5.26}$ |
| *50% Samples, 70% Tokens* | | | | | | | | | | |
| Random | Random | 27.90 | 2.50 | 15.20 | 32.30 | 2.46 | 17.38 | 14.94 | 1.76 | 8.35 |
| | PPL | 27.45$^{\downarrow0.45}$ | 2.50$^{\uparrow0.00}$ | 14.97$^{\downarrow0.23}$ | 31.99$^{\downarrow0.31}$ | 2.04$^{\downarrow0.42}$ | 17.02$^{\downarrow0.36}$ | 16.15$^{\uparrow1.21}$ | 1.80$^{\uparrow0.04}$ | 8.97$^{\uparrow0.62}$ |
| | FastV | 18.20$^{\downarrow9.70}$ | 1.92$^{\downarrow0.58}$ | 10.06$^{\downarrow5.14}$ | 17.44$^{\downarrow14.86}$ | 1.72$^{\downarrow0.74}$ | 9.58$^{\downarrow7.80}$ | 12.13$^{\downarrow2.81}$ | 1.60$^{\downarrow0.16}$ | 6.87$^{\downarrow1.48}$ |
| | SparseVLM | 10.31$^{\downarrow17.59}$ | 3.58$^{\uparrow1.08}$ | 6.95$^{\downarrow8.25}$ | 12.21$^{\downarrow20.09}$ | 2.86$^{\uparrow0.40}$ | 7.53$^{\downarrow9.85}$ | 11.90$^{\downarrow3.04}$ | 3.34$^{\uparrow1.58}$ | 7.62$^{\downarrow0.73}$ |
| Longest | Random | 26.23$^{\downarrow1.67}$ | 2.76$^{\uparrow0.26}$ | 14.50$^{\downarrow0.70}$ | 27.82$^{\downarrow4.48}$ | 2.20$^{\downarrow0.26}$ | 15.01$^{\downarrow2.37}$ | 15.69$^{\uparrow0.75}$ | 1.92$^{\uparrow0.16}$ | 8.81$^{\uparrow0.46}$ |
| | PPL | 30.25$^{\uparrow2.35}$ | 2.64$^{\uparrow0.14}$ | 16.45$^{\uparrow1.25}$ | 31.46$^{\downarrow0.84}$ | 2.00$^{\downarrow0.46}$ | 16.73$^{\downarrow0.65}$ | 16.45$^{\uparrow1.51}$ | 1.94$^{\uparrow0.18}$ | 9.20$^{\uparrow0.85}$ |
| | FastV | 18.42$^{\downarrow9.48}$ | 2.28$^{\downarrow0.22}$ | 10.35$^{\downarrow4.85}$ | 18.65$^{\downarrow13.65}$ | 1.82$^{\downarrow0.64}$ | 10.24$^{\downarrow7.14}$ | 12.43$^{\downarrow2.51}$ | 1.94$^{\uparrow0.18}$ | 7.19$^{\downarrow1.16}$ |
| | SparseVLM | 18.95$^{\downarrow8.95}$ | **4.84**$^{\uparrow2.34}$ | 11.90$^{\downarrow3.30}$ | 19.48$^{\downarrow12.82}$ | 4.74$^{\uparrow2.28}$ | 12.11$^{\downarrow5.27}$ | 11.07$^{\downarrow3.87}$ | 3.90$^{\uparrow2.14}$ | 7.48$^{\downarrow0.87}$ |
| InfoBatch | Random | 29.42$^{\uparrow1.52}$ | 2.92$^{\uparrow0.42}$ | 16.17$^{\uparrow0.97}$ | 35.03$^{\uparrow2.73}$ | 2.50$^{\uparrow0.04}$ | 18.76$^{\uparrow1.38}$ | 16.00$^{\uparrow1.06}$ | 1.78$^{\uparrow0.02}$ | 8.89$^{\uparrow0.54}$ |
| | PPL | 28.73$^{\uparrow0.83}$ | 2.78$^{\uparrow0.28}$ | 15.76$^{\uparrow0.56}$ | 38.82$^{\uparrow6.52}$ | 2.98$^{\uparrow0.52}$ | 20.90$^{\uparrow3.52}$ | 15.39$^{\uparrow0.45}$ | 2.20$^{\uparrow0.44}$ | 8.80$^{\uparrow0.45}$ |
| | FastV | 19.33$^{\downarrow8.57}$ | 1.76$^{\downarrow0.74}$ | 10.55$^{\downarrow4.65}$ | 18.04$^{\downarrow14.26}$ | 1.94$^{\downarrow0.52}$ | 9.99$^{\downarrow7.39}$ | 12.05$^{\downarrow2.89}$ | 1.86$^{\uparrow0.10}$ | 6.96$^{\downarrow1.39}$ |
| | SparseVLM | 9.10$^{\downarrow18.80}$ | 3.12$^{\uparrow0.62}$ | 6.11$^{\downarrow9.09}$ | 12.05$^{\downarrow20.25}$ | 2.32$^{\downarrow0.14}$ | 7.19$^{\downarrow10.19}$ | 10.84$^{\downarrow4.10}$ | **3.96**$^{\uparrow2.20}$ | 7.40$^{\downarrow0.95}$ |
| Entropy | Random | 32.60$^{\uparrow4.70}$ | 2.14$^{\downarrow0.36}$ | 17.37$^{\uparrow2.17}$ | 38.67$^{\uparrow6.37}$ | 2.10$^{\downarrow0.36}$ | 20.38$^{\uparrow3.00}$ | 18.88$^{\uparrow3.94}$ | 2.18$^{\uparrow0.42}$ | 10.53$^{\uparrow2.18}$ |
| | PPL | 33.97$^{\uparrow6.07}$ | 3.10$^{\uparrow0.60}$ | 18.53$^{\uparrow3.33}$ | 40.18$^{\uparrow7.88}$ | 2.30$^{\downarrow0.16}$ | 21.24$^{\uparrow3.86}$ | 17.21$^{\uparrow2.27}$ | 2.06$^{\uparrow0.30}$ | 9.64$^{\uparrow1.29}$ |
| | FastV | 24.94$^{\downarrow2.96}$ | 2.26$^{\downarrow0.24}$ | 13.60$^{\downarrow1.60}$ | 21.53$^{\downarrow10.77}$ | 1.56$^{\downarrow0.90}$ | 11.55$^{\downarrow5.83}$ | 16.22$^{\uparrow1.28}$ | 1.78$^{\uparrow0.02}$ | 9.00$^{\uparrow0.65}$ |
| | SparseVLM | 10.31$^{\downarrow17.59}$ | 4.56$^{\uparrow2.06}$ | 7.44$^{\downarrow7.76}$ | 9.40$^{\downarrow22.90}$ | 4.40$^{\uparrow1.94}$ | 6.90$^{\downarrow10.48}$ | 13.95$^{\downarrow0.99}$ | 3.68$^{\uparrow1.92}$ | 8.82$^{\uparrow0.47}$ |
| **Q-Tuning (Ours)** | | **38.21**$^{\uparrow10.31}$ | 4.30$^{\uparrow1.80}$ | **21.26**$^{\uparrow6.06}$ | **48.07**$^{\uparrow15.77}$ | 6.14$^{\uparrow3.68}$ | **26.57**$^{\uparrow9.19}$ | **20.47**$^{\uparrow5.53}$ | 3.20$^{\uparrow1.44}$ | **11.84**$^{\uparrow3.49}$ |
| Full Dataset | | 32.90 | 3.02 | 17.96 | 42.08 | 3.08 | 22.58 | 16.53 | 2.10 | 9.31 |

sensitivity to batch size. Overall, these results indicate that Q-Tuning is robust to the choice of batch size, with moderate to larger batches offering additional benefits in certain tasks.

**Effectiveness of context awareness $\lambda$.** We also examined the impact of neighbor awareness, controlled by the coefficient $\lambda$. As shown in the lower part of Figure 3, moderate values of $\lambda$ improve performance on GSM8K and SQuAD, whereas extreme values yield diminishing or unstable gains. In contrast, TriviaQA shows little sensitivity to $\lambda$, with performance remain-

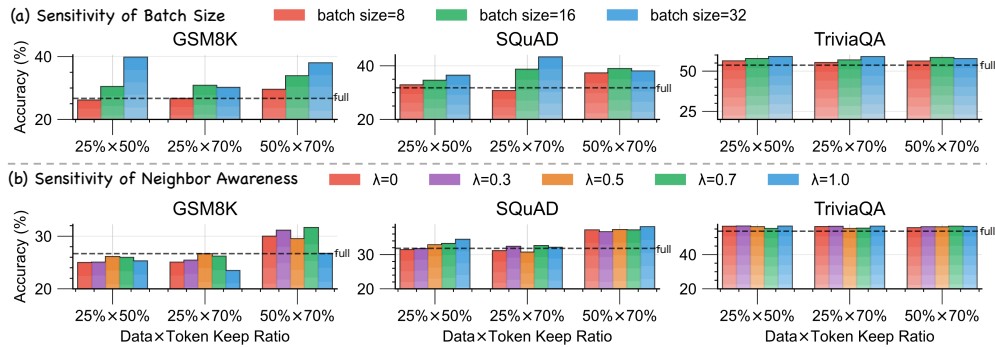

Figure 3: Effect of varying (a) batch size (8, 16, 32) and (b) neighbor awareness $\lambda$ (0, 0.3, 0.5, 0.7, 1.0) for Mistral-7B under three data–token keep ratio configurations ($25\% \times 50\%$, $25\% \times 70\%$, $50\% \times 70\%$). Results are reported on GSM8K, SQuAD, and TriviaQA. Dashed lines marked "full" denote models trained on the full data without pruning. Additional benchmark results (Avg. of five benchmarks, ARC-E, ARC-C) are provided in the Figure 5 in Appendix C.2.

ing stable across all settings. These results suggest that Q-Tuning benefits from incorporating neighbor awareness, but only up to a moderate level, beyond which gains are marginal.

**Ablation study on Token Pruning**

**Method in Q-Tuning.** Table 3 compares token pruning methods under a fixed 25% sample and 50% token keep ratio on Mistral-7B. Rho-1 uses a reference model obtained by train-

Table 3: Ablation study on token pruner in Q-tuning.

| Method | Avg. | ARC-E | ARC-C | GSM8K | SQuAD | TriviaQA |
|---|---|---|---|---|---|---|
| Rho-1 | 45.03 | 69.66 | 46.78 | 24.03 | 29.23 | 55.43 |
| PPL ($\lambda = 0$) | 45.92 | 71.60 | 46.44 | 25.32 | 29.71 | **56.54** |
| Reversed PPL ($\lambda = 0.5$) | 44.86 | **73.02** | 47.12 | 16.68 | 32.01 | 55.47 |
| PPL ($\lambda = 0.5$) | **46.79** | 71.78 | **47.12** | **26.08** | **32.79** | 56.17 |

ing Mistral-7B on the full dataset for 3 epochs. While baseline methods such as Rho-1 and PPL perform competitively on certain benchmarks, they exhibit inconsistent or imbalanced performance across tasks. In contrast, our Neighbor-aware PPL achieves the best overall balance and attains the highest average score. Notably, reversing the token scoring logic (*i.e.*, pruning low-PPL tokens instead of high-PPL ones) significantly degrades performance, underscoring the importance of our context-aware pruning strategy.

**Can Q-Tuning Outperform Other Methods in Independent Sample and Token Pruning?** The core innovation of Q-Tuning is its rejection of the naive, independent application of sample and token pruning. To evaluate whether each pruning strategy is effective on its own, we conducted ablation studies. Figure 4 provides a direct comparison under controlled conditions: panel (a) illustrates dynamic sample pruning with all tokens retained, while panel (b) shows dynamic token pruning with all samples retained. Across both scenarios, Q-Tuning consistently surpasses all baseline methods.

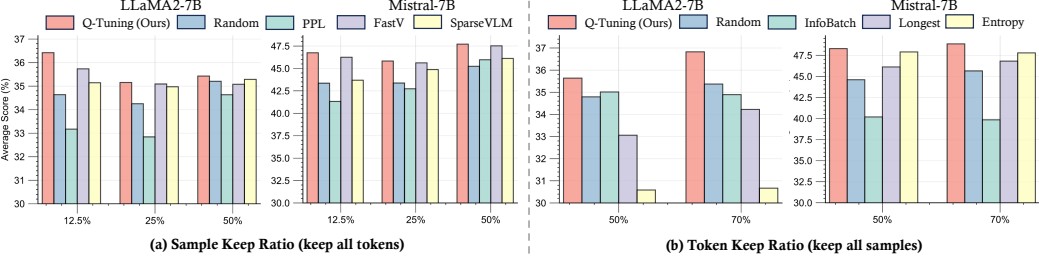

Figure 4: Comparison of independent (a) dynamic sample pruning and (b) dynamic token pruning across different keep ratios for LLaMA2-7B and Mistral-7B. Q-Tuning still outperforms all baselines.

**How samples distribution evolve during training?** We conducted experiments on different sample pruners and visualized the average entropy and PPL for samples. Tracking 100 randomly sampled instances, we found that Q-Tuning reduces both perplexity and entropy more rapidly than other strategies, which in turn facilitates better model performance (see Figure 6 in Appendix C.2).

## 5 CONCLUSION

This work turns risky dynamic data pruning for LLM fine-tuning into a reliable, high-utility strategy. By analyzing sample modes via the novel Error-Uncertainty Plane, we expose data's heterogeneous

value and the need for nuance. Our Quadrant-based Tuning (Q-Tuning) uses a two-stage framework to smartly coordinate sample- and token-level pruning, preserving valuable signals while removing noise—boosting efficiency without sacrificing (and often improving) performance.

## REPRODUCIBILITY STATEMENT

The supplementary material provides the runnable code implementation of Q-Tuning together with training scripts, detailed algorithmic descriptions, and hyperparameter settings (Sections 3, Appendix B.2), which are sufficient to reproduce the main results reported in the paper.

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

# A   RELATED WORK

## A.1   SAMPLE PRUNING

A growing body of work explores sample pruning as a means to reduce training cost by selecting smaller yet higher-quality subsets of data without compromising performance (Ivison et al., 2025; Yang et al., 2024; Xia et al., 2024a; Zhou et al., 2024). For example, LIMA (Zhou et al., 2023a) shows that aligning LLMs requires only a small collection of high-quality instruction–response pairs rather than large-scale corpora. Broadly, existing approaches fall into static and dynamic pruning. In the static setting, data subsets are determined in advance using fixed criteria: SVP (Coleman et al., 2020) leverages proxy models to estimate sample importance, D3 (Jin et al., 2025) combines diversity, difficulty, and dependability into weighted coresets, and Less is More (Deng et al., 2025) shows that carefully curated preference subsets can outperform full datasets by filtering noisy or redundant examples. Other static approaches target in-context examples: LIMO (Ye et al., 2025) removes redundant demonstrations while distilling essential reasoning patterns, and methods such as DEFT (Ivison et al., 2022), Alpagasus (Chen et al., 2023), and Data Whisperer (Wang et al., 2025a) automatically select or reweight demonstrations based on influence estimation, contribution to performance, or few-shot evaluation.

In contrast, dynamic pruning adapts sample usage throughout training. Early work by (Raju et al., 2021) proposed two methods, $\epsilon$-greedy and UCB, which retain uncertain examples while discarding easier ones. Subsequent approaches explore alternative criteria for adaptive pruning: RHO-LOSS (Mindermann et al., 2022) assigns importance using high-quality reference models, Info-Batch (Qin et al., 2024) removes low-loss examples on the fly, and more recent methods such as MATES (Huang et al., 2024) and DWM (Li et al., 2025) reweight or select samples according to the evolving model state.Beyond single-example pruning, several methods consider structured or representation-based criteria: ActivePrune (Azeemi et al., 2024) selects demonstrations per input via similarity metrics, SCAN (Guo & Kankanhalli, 2024) iteratively removes ill-matched and redundant data during contrastive pretraining, GDeR (Zhang et al., 2024a) models data as a prototypical graph to prune noisy or redundant samples, and DQ (Zhou et al., 2023b) clusters data in representation space and replaces each cluster with representative subsets.

## A.2   TOKEN PRUNING

Parallel to sample pruning, recent research (Wang et al., 2025b; Pan et al., 2024; Keith et al., 2024) has increasingly focused on token-level pruning to retain only the most informative portions of input sequences. At the pretraining stage, methods such as Rho-1 (Lin et al., 2024) leverage reference models to estimate token importance, discarding uninformative tokens. In fine-tuning and inference, pruning is often performed dynamically: Token Cleaning (Pang et al., 2025) identifies harmful tokens as noisy labels and prunes them based on their estimated influence on parameter updates, while approaches like LazyLLM (Fu et al., 2024) and SlimInfer (Long et al., 2025) accelerate long-context inference by selectively dropping tokens or hidden-state blocks with the help of attention signals and cache mechanisms. Along similar lines, TokenSkip (Xia et al., 2025) removes redundant reasoning steps by skipping semantically less important tokens, and FTP (Xu et al., 2024b) reduces prefilling cost through grouped token processing and optimized attention computation. Another family of approaches instead performs input compression before inference, exemplified by LLMLingua (Jiang et al., 2023b) and Selective Context (LI et al., 2023), which prune low-information tokens, phrases, or sentences through coarse-to-fine filtering with budget control. Beyond text-only models, multimodal pruning exploits cross-modal attention patterns: FastV (Chen et al., 2024a) and SparseVLM (Zhang et al., 2024b) drop redundant visual tokens guided by attention, while LMTL (Huang et al., 2023) dynamically adjusts computation by pruning unnecessary visual features. More recent advances further refine pruning with task-specific objectives, such as GAP (Chien et al., 2025), which preserves spatial grounding by correcting position misalignments during token removal, and DART (Wen et al., 2025), which reduces token-level computation through confidence-based early stopping.

Despite their progress, prior work typically treats sample pruning and token pruning as independent lines of research. Sample pruning methods focus on reducing the number of training examples but leave token-level redundancy unaddressed, while token pruning techniques emphasize sequence-level efficiency without considering redundancy across training samples. Such separation limits the potential for jointly optimizing efficiency and effectiveness. In this work, we bridge these directions

by proposing a unified framework that integrates sample-level and token-level pruning, enabling models to simultaneously filter uninformative data and compress redundant inputs.

# B   ADDITIONAL EXPERIMENTAL SETTINGS

## B.1   IMPLEMENTATION DETAILS

All experiments were conducted using the *LLaMA-Factory* framework (Zheng et al., 2024), which provided utilities for training and evaluation across diverse large language models. For benchmark evaluation, we used the *OpenCompass* framework (Contributors, 2023), which offered a standardized interface to a wide range of tasks. Unless otherwise specified, models were trained for 3 epochs with a learning rate of $1 \times 10^{-4}$ and a batch size of 8 per device. In all pruning experiments, we applied pruning exclusively to the question part of each input, leaving answers intact. All experiments were performed on NVIDIA A100 GPUs. For the *Random* baseline, we repeated each experiment five times and reported the averaged results.

## B.2   HYPERPARAMETERS

Table 4: Complete hyperparameter configurations employed in our experiments.

| Hyperparameter | Value |
|---|---|
| $\lambda$ | 0.5 |
| Train batch size | 8 |
| Epoch | 3 |
| Learning rate | $1 \times 10^{-4}$ |
| Lora rank | 8 |
| Cutoff length | 2048 |
| Gradient accumulation step | 4 |
| Learning rate scheduler type | cosine |

We provide the full hyperparameter settings used in our experiments in Table 4. Unless otherwise noted, all other parameters follow the default settings of the *LLaMA-Factory* framework.

# C    ADDITIONAL EXPERIMENTAL RESULTS

## C.1    RESULTS ON INSTRUCTION DATASETS

Table 5: Evaluation on OpenHermes dataset under different sample ratios (25%, 50%) and token ratios (50%, 70%), reported on ARC-E, ARC-C, GSM8K, SQuAD, TriviaQA, and their average, where ↑ and ↓ respectively denote improvements or degradations over the *Random-Random* baseline.

| Sample Pruner | Token Pruner | LLaMA2-7B | | | | | | Mistral-7B | | | | | |
|---|---|---|---|---|---|---|---|---|---|---|---|---|---|
| | | ARC-E | ARC-C | GSM8K | SQuAD | TriviaQA | Avg. | ARC-E | ARC-C | GSM8K | SQuAD | TriviaQA | Avg. |
| Zero-Shot | | 53.44 | 38.98 | 5.31 | 12.18 | 43.00 | 30.58 | 66.67 | 46.10 | 18.35 | 10.01 | 43.77 | 36.98 |
| *25% Samples, 70% Tokens* | | | | | | | | | | | | | |
| Random | Random | 60.67 | 41.69 | 6.22 | 13.91 | 48.77 | 34.25 | 70.02 | 46.78 | 19.71 | 23.50 | 52.93 | 42.59 |
| | PPL | 60.32↓0.35 | 42.03↑0.34 | 5.91↓0.31 | 15.88↑1.97 | 48.56↓0.21 | 34.54↑0.29 | 69.49↓0.53 | 48.14↑1.36 | 21.08↑1.37 | 20.63↓2.87 | 53.73↑0.80 | 42.61↑0.02 |
| | FastV | 60.32↓0.35 | 42.71↑1.02 | 6.07↓0.15 | 14.92↑1.01 | 47.20↓1.57 | 34.24↓0.01 | 71.25↑1.23 | 47.80↑1.02 | 18.95↓0.76 | 24.29↑0.79 | 52.07↓0.86 | 42.87↑0.28 |
| | SparseVLM | 55.73↓4.94 | 37.97↓3.72 | 7.20↑0.98 | 11.83↓2.08 | 44.96↓3.81 | 31.54↓2.71 | 67.72↓2.30 | 44.75↓2.03 | 25.32↑5.61 | 12.69↓10.81 | 47.46↓5.47 | 39.59↓3.00 |
| Longest | Random | 61.02↑0.35 | 41.69↑0.00 | 7.20↑0.98 | 15.61↑1.70 | 49.53↑0.76 | 35.01↑0.76 | **74.43**↑4.41 | 47.80↑1.02 | 24.56↑4.85 | 26.79↑3.29 | 55.24↑2.31 | 45.76↑3.17 |
| | PPL | 61.20↑0.53 | 43.05↑1.36 | 7.28↑1.06 | 14.06↑0.15 | 48.87↑0.10 | 34.89↑0.64 | 72.49↑2.47 | 47.46↑0.68 | 25.55↑5.84 | 23.92↑0.42 | 55.26↑2.33 | 44.93↑2.34 |
| | FastV | 59.26↓1.41 | 43.05↑1.36 | 5.53↓0.69 | 13.98↑0.07 | 48.03↓0.74 | 33.97↓0.28 | 72.84↑2.82 | 47.80↑1.02 | 19.79↑0.08 | 31.54↑8.04 | 53.24↑0.31 | 45.04↑2.45 |
| | SparseVLM | 55.73↓4.94 | 36.27↓5.42 | 8.04↑1.82 | 11.39↓2.52 | 44.65↓4.12 | 31.22↓3.03 | 68.08↓1.94 | 43.73↓3.05 | 25.78↑6.07 | 13.47↓10.03 | 48.00↓4.93 | 39.81↓2.78 |
| InfoBatch | Random | 59.79↓0.88 | 42.71↑1.02 | 5.84↓0.38 | 10.42↓3.49 | 48.99↑0.22 | 33.55↓0.70 | 70.37↑0.35 | 46.44↓0.34 | 21.00↑1.29 | 24.09↑0.59 | 53.75↑0.82 | 43.13↑0.54 |
| | PPL | 60.32↓0.35 | **43.39**↑1.70 | 6.67↑0.45 | 13.88↓0.55 | 49.03↑0.26 | 34.66↑0.41 | 71.08↑1.06 | 46.10↓0.68 | 22.44↑2.73 | 22.07↓1.43 | 52.17↓0.76 | 42.77↑0.18 |
| | FastV | 59.61↓1.06 | 42.71↑1.02 | 3.34↓2.88 | 12.18↓1.73 | 47.98↓0.79 | 33.16↓1.09 | 69.66↓0.36 | 45.76↓1.02 | 15.39↓4.21 | 24.75↑1.25 | 49.51↓3.42 | 41.01↓1.58 |
| | SparseVLM | 55.56↓5.11 | 38.31↓3.38 | 5.38↓0.84 | 10.81↓3.10 | 43.08↓5.69 | 30.63↓3.62 | 65.78↓4.24 | 45.08↓1.70 | 20.70↑0.99 | 10.97↓12.53 | 48.66↓4.27 | 38.24↓4.35 |
| Entropy | Random | 60.67↑0.00 | 41.02↓0.67 | 5.99↓0.23 | 14.92↑1.01 | 49.34↑0.57 | 34.39↑0.14 | 70.37↑0.35 | **48.47**↑1.69 | 22.21↑2.50 | **56.05**↑3.12 | 44.93↑2.34 | |
| | PPL | 61.55↑0.88 | 41.02↓0.67 | 5.38↓0.84 | 14.90↑0.99 | 49.72↑0.95 | 34.51↑0.26 | 70.90↑0.88 | 48.14↑1.36 | 21.15↑1.44 | 26.22↑2.72 | 55.83↑2.90 | 44.45↑1.86 |
| | FastV | 59.79↓0.88 | 42.03↑0.34 | 6.44↑0.22 | 13.21↓0.70 | 47.58↓1.19 | 33.81↓0.44 | 71.43↑1.41 | 47.12↑0.34 | 20.24↑0.53 | 26.06↑2.56 | 54.40↑1.47 | 43.85↑1.26 |
| | SparseVLM | 54.50↓6.17 | 37.97↓3.72 | 8.87↑2.65 | 12.25↓1.66 | 45.05↓3.72 | 31.73↓2.52 | 67.20↓2.82 | 45.76↓1.02 | 25.17↑5.46 | 9.64↓13.86 | 48.38↓4.55 | 39.23↓3.36 |
| **Q-Tuning (Ours)** | | **62.43**↑1.76 | 42.37↑0.68 | **9.25**↑3.03 | **19.53**↑5.62 | **50.78**↑2.01 | **36.87**↑2.62 | 71.60↑1.58 | 47.12↑0.34 | **26.61**↑6.90 | 30.64↑7.14 | 55.13↑2.20 | **46.22**↑3.63 |
| Full Dataset | | 61.55 | 42.37 | 8.64 | 13.80 | 50.45 | 35.36 | 71.25 | 45.76 | 26.68 | 31.81 | 53.67 | 45.84 |
| *50% Samples, 50% Tokens* | | | | | | | | | | | | | |
| Random | Random | 62.08 | 41.36 | 6.75 | 12.14 | 48.86 | 34.24 | 71.25 | 46.44 | 21.53 | 24.91 | 54.16 | 43.66 |
| | PPL | 61.02↓1.06 | **43.05**↑1.69 | 6.82↑0.07 | 15.08↑2.94 | 49.10↑0.24 | 35.01↑0.77 | 72.49↑1.24 | 46.78↑0.34 | 21.70↑0.68 | 33.28↑8.37 | 53.83↓0.33 | 45.72↑2.06 |
| | FastV | 60.32↓1.76 | 42.03↑0.67 | 4.40↓2.35 | 11.80↓0.34 | 48.35↓0.51 | 33.38↓0.86 | 72.13↑0.88 | 45.42↓1.02 | 16.22↓5.31 | 27.69↑2.78 | 50.58↓3.58 | 42.41↓1.25 |
| | SparseVLM | 55.73↓6.35 | 38.31↓3.05 | 7.20↑0.45 | 11.33↓0.81 | 44.80↓4.06 | 31.47↓2.77 | 67.90↓3.35 | 45.42↓1.02 | 25.78↑4.25 | 11.85↓13.06 | 47.64↓6.52 | 39.72↓3.94 |
| Longest | Random | 60.85↓1.23 | 42.03↑0.67 | 7.05↑0.30 | 14.90↑2.76 | 49.46↑0.60 | 34.86↑0.62 | 72.49↑1.24 | 46.78↑0.34 | 21.91↑0.38 | 29.41↑4.50 | 55.99↑1.83 | 45.45↑1.79 |
| | PPL | 62.08↑0.00 | 41.69↑0.33 | 7.51↑0.76 | 14.89↑2.75 | 48.99↑0.13 | 35.03↑0.79 | 71.60↑0.35 | 46.78↑0.34 | 24.11↑2.58 | 29.17↑4.26 | 55.83↑1.67 | 45.50↑1.84 |
| | FastV | 60.32↓1.76 | 40.68↓0.68 | 4.17↓2.58 | 13.26↑1.12 | 48.24↓0.62 | 33.33↓0.91 | 72.31↑1.06 | 46.78↑0.34 | 16.83↓4.70 | 30.37↑5.46 | 53.99↓0.17 | 43.85↑0.19 |
| | SparseVLM | 56.97↓5.11 | 39.66↓1.70 | 6.60↓0.15 | 11.93↓0.21 | 44.52↓4.34 | 31.93↓2.31 | 69.31↓1.94 | 42.71↓3.73 | 23.73↑2.20 | 12.72↓12.19 | 49.90↓4.26 | 39.68↓3.98 |
| InfoBatch | Random | 58.38↓3.70 | 42.71↑1.35 | 5.76↓0.99 | 13.52↑1.38 | 48.95↑0.09 | 33.86↓0.38 | 71.08↓0.17 | 46.44↑0.00 | 21.38↓0.15 | 26.73↑1.82 | 53.67↓0.49 | 43.86↑0.20 |
| | PPL | 60.67↓1.41 | 42.71↑1.35 | 5.76↓0.99 | 14.29↑2.15 | 49.18↑0.32 | 34.52↑0.28 | 71.43↑0.18 | 47.80↑1.36 | 24.11↑2.58 | 30.22↑5.31 | 53.97↓0.19 | 45.50↑1.84 |
| | FastV | 59.44↓2.64 | 41.36↑0.00 | 3.87↓2.88 | 12.39↑0.25 | 48.25↓0.61 | 33.06↓1.18 | 70.72↓0.53 | 45.42↓1.02 | 17.97↓3.56 | 27.83↑2.92 | 51.53↓2.63 | 41.88↓1.78 |
| | SparseVLM | 56.08↓6.00 | 37.97↓3.39 | 6.52↓0.23 | 10.33↓1.81 | 43.84↓5.02 | 30.95↓3.29 | 68.25↓3.00 | 41.69↓4.75 | **29.72**↑8.19 | 9.95↓14.96 | 49.22↓4.94 | 39.77↓3.89 |
| Entropy | Random | 59.96↓2.12 | 42.71↑1.35 | 7.66↑0.91 | **15.47**↑3.33 | 49.07↑0.21 | 34.97↑0.73 | 72.13↑0.88 | 47.80↑1.36 | 22.67↑1.14 | 25.19↑0.28 | 55.70↑1.54 | 44.70↑1.04 |
| | PPL | 60.49↓1.59 | 41.02↓0.34 | 5.67↓1.08 | 13.51↑1.37 | 49.75↑0.89 | 34.29↑0.05 | 72.13↑0.88 | 47.46↑1.02 | 25.17↑3.64 | 30.25↑5.34 | **56.58**↑2.42 | 46.39↑2.73 |
| | FastV | 59.26↓2.82 | 42.37↑1.01 | 4.02↓2.73 | 13.81↑1.67 | 48.93↑0.07 | 33.68↓0.56 | 71.60↑0.35 | 47.46↑1.02 | 17.97↓3.56 | 29.82↑4.91 | 54.40↑0.24 | 44.25↑0.59 |
| | SparseVLM | 56.97↓5.11 | 38.64↓2.72 | 5.53↓1.22 | 11.28↓0.86 | 44.72↓4.14 | 31.43↓2.81 | 67.55↓3.70 | 43.73↓2.71 | 25.85↑4.32 | 7.09↓17.82 | 48.24↓5.92 | 38.49↓5.17 |
| **Q-Tuning (Ours)** | | **62.79**↑0.71 | 42.03↑0.67 | **10.46**↑3.71 | 14.53↑2.39 | **51.05**↑2.19 | **36.17**↑1.93 | **73.37**↑2.12 | **48.14**↑1.70 | 28.81↑7.28 | **36.35**↑11.44 | 56.30↑2.14 | **48.59**↑4.93 |
| Full Dataset | | 61.55 | 42.37 | 8.64 | 13.80 | 50.45 | 35.36 | 71.25 | 45.76 | 26.68 | 31.81 | 53.67 | 45.84 |
| *50% Samples, 70% Tokens* | | | | | | | | | | | | | |
| Random | Random | 61.02 | 41.36 | 7.43 | 15.56 | 48.91 | 34.85 | 71.60 | 47.12 | 22.59 | 27.86 | 53.68 | 44.57 |
| | PPL | 60.67↓0.35 | 42.71↑1.35 | 6.75↓0.68 | 14.93↓0.63 | 49.13↑0.22 | 34.84↓0.01 | 72.31↑0.71 | 47.12↑0.00 | 20.70↓1.89 | 32.02↑4.16 | 53.56↓0.12 | 45.14↑0.57 |
| | FastV | 59.61↓1.41 | 42.37↑1.01 | 5.53↓1.90 | 13.36↓2.20 | 48.48↓0.43 | 33.87↓0.98 | 72.31↑0.71 | 45.08↓2.04 | 18.95↓3.64 | 28.60↑0.74 | 51.40↓2.28 | 43.27↓1.30 |
| | SparseVLM | 54.67↓6.35 | 38.98↓2.38 | 7.28↓0.15 | 10.84↓4.72 | 44.68↓4.23 | 31.29↓3.56 | 68.78↓2.82 | 46.10↓1.02 | 23.65↑1.06 | 11.32↓16.54 | 49.04↓4.64 | 39.78↓4.79 |
| Longest | Random | 61.20↑0.18 | 41.02↓0.34 | 6.37↓1.06 | 14.90↓0.66 | 49.79↑0.88 | 34.65↓0.20 | 72.13↑0.53 | **48.47**↑1.35 | 23.05↑0.46 | 29.60↑1.74 | 55.94↑2.26 | 45.84↑1.27 |
| | PPL | 62.26↑1.24 | 41.36↑0.00 | 6.82↓0.61 | 14.83↓0.73 | 49.04↑0.13 | 34.86↑0.01 | 72.31↑0.71 | 47.80↑0.68 | 24.87↑2.28 | 27.33↓0.53 | 55.62↑1.94 | 45.30↑0.73 |
| | FastV | 59.61↓1.41 | **42.71**↑1.35 | 5.53↓1.90 | 14.12↓1.44 | 48.77↓0.14 | 34.15↓0.70 | 72.66↑1.06 | 45.08↓2.04 | 19.94↓2.65 | 31.53↑3.67 | 54.66↑0.98 | 44.77↑0.20 |
| | SparseVLM | 55.38↓5.64 | 39.66↓1.70 | 6.14↓1.29 | 11.51↓4.05 | 44.56↓4.35 | 31.45↓3.40 | 69.14↓2.46 | 45.42↓1.70 | 23.43↑0.84 | 13.42↓14.44 | 47.86↓5.82 | 39.85↓4.72 |
| InfoBatch | Random | **61.55**↑0.53 | 40.34↓1.02 | 5.84↓1.59 | 14.99↓0.57 | 49.45↑0.54 | 34.43↓0.42 | 70.90↓0.70 | 45.76↓1.36 | 22.97↑0.38 | 28.85↑0.99 | 54.11↑0.43 | 44.52↓0.05 |
| | PPL | 60.85↓0.17 | 42.03↑0.67 | 6.75↓0.68 | 14.22↓1.34 | 48.41↓0.50 | 34.45↓0.40 | 71.08↓0.52 | 45.76↓1.36 | 22.82↑0.23 | 32.80↑4.94 | 54.63↑0.95 | 45.42↑0.85 |
| | FastV | 58.91↓2.11 | 41.36↑0.00 | 3.42↓3.41 | 14.98↓0.58 | 48.76↓0.15 | 33.60↓1.25 | 70.19↓1.41 | 45.76↓1.36 | 19.86↓2.73 | 28.95↑1.09 | 50.92↓2.76 | 43.14↓1.43 |
| | SparseVLM | 55.91↓5.11 | 37.29↓4.07 | 6.22↓1.21 | 10.40↓5.16 | 43.75↓5.16 | 30.71↓4.14 | 68.96↓2.64 | 44.07↓3.05 | 25.63↑3.04 | 9.82↓18.04 | 49.56↓4.12 | 39.61↓4.96 |
| Entropy | Random | 60.49↓0.53 | 41.69↑0.33 | 7.13↓0.30 | 15.41↓0.15 | 48.33↓0.58 | 34.61↓0.24 | 72.13↑0.53 | 47.46↑0.34 | 23.43↑0.84 | 28.99↑1.13 | 55.44↑1.76 | 45.49↑0.92 |
| | PPL | 60.85↓0.17 | 40.68↓0.68 | 6.29↓1.14 | 13.38↓2.18 | 49.94↑1.03 | 34.23↓0.62 | 72.31↑0.71 | 46.44↓0.68 | 23.20↑0.61 | 31.44↑3.58 | **56.67**↑2.99 | 46.01↑1.44 |
| | FastV | 58.91↓2.11 | 42.37↑1.01 | 4.83↓2.58 | 14.27↓1.29 | 48.84↓0.07 | 33.85↓1.00 | 71.96↑0.36 | 48.14↑1.02 | 20.39↓2.20 | 30.30↑2.44 | 55.05↑0.48 | 45.05↑0.48 |
| | SparseVLM | 56.79↓4.23 | 37.63↓3.73 | 5.84↓1.59 | 11.85↓3.71 | 43.82↓5.09 | 31.18↓3.67 | 67.55↓4.05 | 46.44↓0.68 | 25.17↑2.58 | 10.36↓17.50 | 48.60↓5.08 | 39.62↓4.95 |
| **Q-Tuning (Ours)** | | 61.38↑0.36 | 42.03↑0.67 | **9.55**↑2.12 | **17.69**↑2.13 | **51.04**↑2.13 | **36.34**↑1.49 | **73.90**↑2.30 | 47.46↑0.34 | **29.49**↑6.90 | **37.24**↑9.38 | 56.07↑2.39 | **48.83**↑4.26 |
| Full Dataset | | 61.55 | 42.37 | 8.64 | 13.80 | 50.45 | 35.36 | 71.25 | 45.76 | 26.68 | 31.81 | 53.67 | 45.84 |

The detailed results of additional experiments on general instruction tuning are presented in Table 5, providing a comprehensive comparison across different pruning strategies.

## C.2 ABLATION STUDY

**Sensitivity of hyperparameters** We further examined the robustness of our framework with respect to two critical hyperparameters: batch size and neighbor awareness $\lambda$. As shown in Figure 5, varying the batch size (8, 16, 32) has only a marginal effect across benchmarks and keep-ratio configurations, suggesting that our pruning strategy remains stable even under smaller mini-batches. Meanwhile, the performance under different values of $\lambda$ (0.0–1.0) remains consistently close to or above the full-data baseline, indicating that the method is not overly sensitive to the strength of neighbor-aware token scoring. Taken together, these results highlight the robustness of our approach with respect to hyperparameter choices, further underscoring its practicality in diverse training settings.

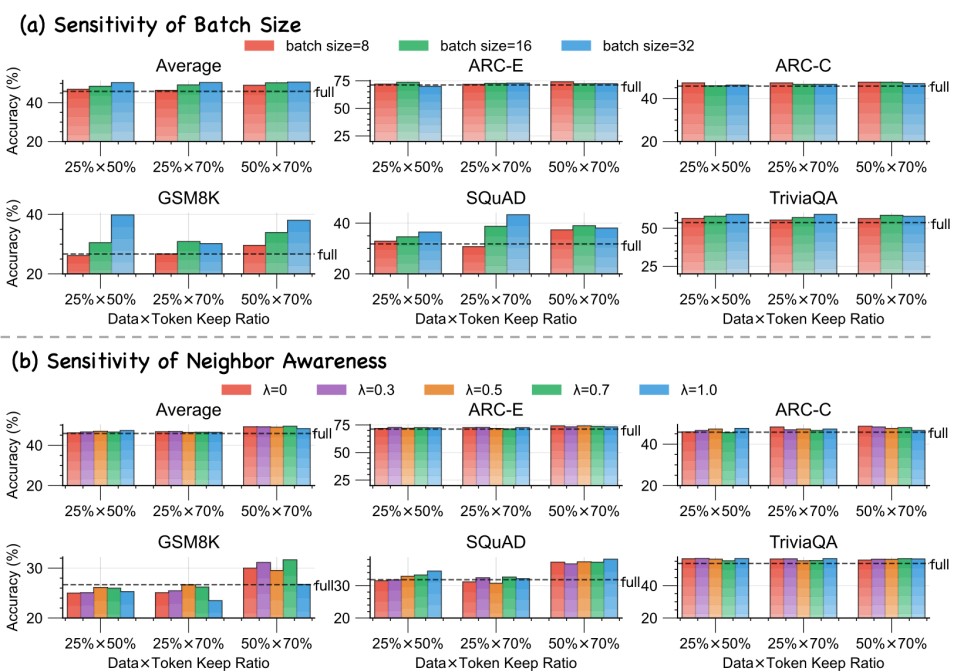

Figure 5: Effect of varying (a) batch size (8, 16, 32) and (b) neighbor awareness $\lambda$ (0, 0.3, 0.5, 0.7, 1.0) for Mistral-7B under three data–token keep ratio configurations ($25\% \times 50\%$, $25\% \times 70\%$, $50\% \times 70\%$). Results are reported on GSM8K, SQuAD, and TriviaQA. Dashed lines marked "full" denote models trained on the full data without pruning.

**The evolution of samples distribution during training** In Figure 6, we further show the training dynamics by tracking the average perplexity and entropy of 100 randomly sampled instances. These results provide supporting evidence that Q-Tuning reduces both metrics more rapidly than alternative pruners.

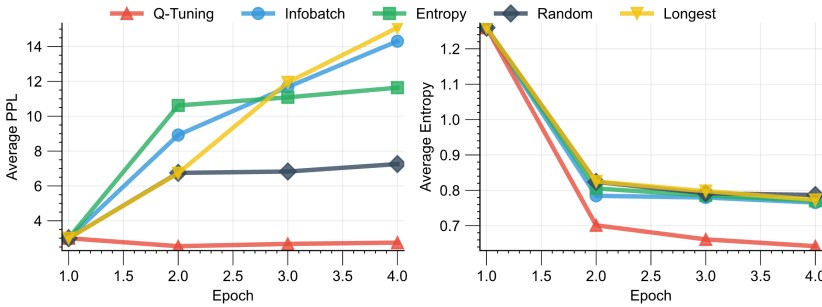

Figure 6: **Training dynamics of different sample pruners.** We tracked the average perplexity (PPL) and entropy on 100 randomly sampled instances across training epochs. Compared to baseline strategies, Q-Tuning consistently reduces both metrics at a faster rate, indicating more efficient learning dynamics.

# D  PSEUDOCODE OF THE PROPOSED METHOD

To facilitate clarity and reproducibility, we summarize the proposed Q-Tuning algorithm in Algorithm 1. The procedure unfolds within a single training iteration and consists of two tightly coupled stages: (i) *sample pruning*, where instances are dynamically selected based on their position in the error–uncertainty (EU) plane through an efficient bisection-based search of quantile thresholds, and (ii) *token pruning*, where retained samples undergo finer-grained filtering to preserve only the most informative subset of tokens.

---

**Algorithm 1** Q-Tuning: dynamic data pruning in one iteration

---

1: **Input:** Mini-batch $\mathcal{B}_t$, model $f_{\theta_t}$, retention ratios $r_{\text{sample}}, r_{\text{token}}$, smoothing $\lambda$.
2: **Output:** Pruned mini-batch $\tilde{\mathcal{B}}_t$.
3: *// Stage 1: Sample Pruning via EU Plane*
4: Compute $(\text{PPL}(x, y; f_\theta), \text{Ent}(x, y; f_\theta))$ for each $x \in \mathcal{B}_t$.
5: Initialize ranges $\alpha_{\text{low}} = 0$, $\alpha_{\text{high}} = 0.49$, $\beta_{\text{low}} = 0$, $\beta_{\text{high}} = 0.49$.
6: **for** $k = 1$ to $K_{\max}$ **do**          ▷ Bisection iterations on both axes
7:      $\alpha \leftarrow (\alpha_{\text{low}} + \alpha_{\text{high}})/2$, $\beta \leftarrow (\beta_{\text{low}} + \beta_{\text{high}})/2$.
8:      Derive thresholds $ppl_{\text{hi}} = Q_{1-\alpha}(\text{PPL})$, $ppl_{\text{lo}} = Q_\alpha(\text{PPL})$, $ent_{\text{lo}} = Q_\beta(\text{Ent})$, $ent_{\text{hi}} = Q_{1-\beta}(\text{Ent})$.
9:      Partition samples into quadrants $Q_1$–$Q_4$.
10:      $r \leftarrow \frac{|Q_2| + |Q_4|}{|\mathcal{B}_t|}$.
11:      **if** $r < r_{\text{sample}}$ **then**
12:          $\alpha_{\text{low}} \leftarrow \alpha$, $\beta_{\text{low}} \leftarrow \beta$          ▷ Too few kept, relax thresholds
13:      **else**
14:          $\alpha_{\text{high}} \leftarrow \alpha$, $\beta_{\text{high}} \leftarrow \beta$          ▷ Too many kept, tighten thresholds
15:      **end if**
16: **end for**
17: $\mathcal{B}'_t \leftarrow Q_2 \cup Q_4$.
18: **if** $|\mathcal{B}'_t| < r_{\text{sample}} \cdot |\mathcal{B}_t|$ **then**          ▷ Ensure target sample ratio
19:      Compute **supp-score** for each $x \in Q_1 \cup Q_3$: $\text{supp}(x) = \max\{\hat{\text{PPL}}(x, y; f_\theta) - \hat{\text{Ent}}(x, y; f_\theta), \hat{\text{Ent}}(x, y; f_\theta) - \hat{\text{PPL}}(x, y; f_\theta)\}$, where $\hat{\cdot}$ denotes min–max normalization.
20:      Select top-scoring samples to augment $\mathcal{B}'_t$ until target size is met.
21: **end if**
22: *// Stage 2: Token Pruning*
23: $\tilde{\mathcal{B}}_t \leftarrow \emptyset$.
24: **for** each $x \in \mathcal{B}'_t$ **do**
25:      **if** $x \in Q_4$ **then**
26:          Keep all tokens of $x$.          ▷ Calibration samples intact
27:      **else if** $x \in Q_2$ **then**
28:          For each token $i \in T(x)$, compute

$$s_i(x, y; f_\theta) = (1 - \lambda)\,\text{PPL}_i(x, y; f_\theta) + \lambda\,[\text{PPL}_{i-1}(x, y; f_\theta) + \text{PPL}_{i+1}(x, y; f_\theta)].$$

29:          Keep top-$r_{\text{token}}$ fraction of tokens ranked by $s_i$.
30:      **end if**
31:      $\tilde{\mathcal{B}}_t \leftarrow \tilde{\mathcal{B}}_t \cup \{m(x) \odot x\}$.
32: **end for**
33: **return** $\tilde{\mathcal{B}}_t$.

---

## E EXAMPLE SAMPLES FROM EACH QUADRANT

We report representative samples from each quadrant (Q1–Q4) at the beginning of training, including their perplexity (PPL), entropy, and the corresponding instruction–output pairs.

| PPL | Entropy | Example (Instruction / Output) |
|---|---|---|
| **Q1 Examples** | | |
| 8.33 | 2.12 | Instruction: What are the three primary colors? Output: The three primary colors are red, blue, and yellow. |
| 8.04 | 1.73 | Instruction: How can we reduce air pollution? Output: There are a number of ways to reduce air pollution, such as shifting to renewable energy sources... |
| 91.24 | 1.91 | Instruction: Explain why the fraction 4/16 is equivalent to 1/4. Output: The fraction 4/16 is equivalent to 1/4 because both numerator and denominator are divisible by 4. |
| **Q2 Examples** | | |
| 8.65 | 1.54 | Instruction: Write a short story in third person narration about a protagonist making an important career decision. Output: John was at a crossroads in his life. He had just graduate... |
| 11.02 | 1.48 | Instruction: Write a short story in third person narration about a protagonist making an important career decision. Output: John was at a crossroads in his life. He had just graduate... |
| 15.74 | 1.50 | Instruction: Identify the odd one out.\nTwitter, Instagram, Telegram Output: Telegram |
| **Q3 Examples** | | |
| 4.80 | 1.45 | Instruction: Describe the structure of an atom. Output: An atom is made up of a nucleus containing protons and neutrons, surrounded by electrons... |
| 2.41 | 1.09 | Instruction: How can we reduce air pollution? Output: There are a number of ways to reduce air pollution, such as shifting to renewable energy sources... |
| 4.73 | 1.49 | Instruction: Identify the odd one out (Twitter, Instagram, Telegram). Output: Telegram |
| **Q4 Examples** | | |
| 6.03 | 1.83 | Instruction: Give three tips for staying healthy. Output: 1. Eat a balanced diet... 2. Exercise regularly... |
| 8.04 | 1.71 | Instruction: What are the three primary colors? Output: The three primary colors are red, blue, and yellow. |
| 5.37 | 1.79 | Instruction: Write a short story in third person narration about a protagonist making an important career decision. Output: John was at a crossroads in his life. He had just graduate... |

## F    USE OF LLMs

This work did not involve the use of large language models (LLMs) at any stage. The design of experiments, data analysis, and manuscript preparation were conducted entirely by the authors through conventional computational methods and human expertise, without reliance on automated text generation or model-driven reasoning.

# G    DIVERSE EXPERIMENTAL RESULTS

## G.1    EXPERIMENTAL SETTINGS

**Models and Datasets.** Beyond the main experiments on Mistral-7B, LLaMA2-7B, LLaMA3-8B, and SmolLM2-1.7B, we further evaluate Q-Tuning on the *Qwen3-8B* model (Yang et al., 2025) and larger Qwen models (*Qwen-14B*, *Qwen-32B*). For Qwen family models, we fine-tune on the *OpenHermes* dataset (Teknium, 2023), which contains diverse instruction–response pairs for general-purpose alignment. In addition, the extended baselines on LLaMA2-7B and Mistral-7B and the extreme data-retention experiments on Mistral-7B are all conducted on WizardLM (Xu et al., 2024a).

**Evaluation.** Unless otherwise specified, all extended experiments are evaluated on the same five standard benchmarks used in the main results: ARC-E, ARC-C (Clark et al., 2018), GSM8K (Cobbe et al., 2021), SQuAD (Rajpurkar et al., 2016), and TriviaQA (Joshi et al., 2017). These benchmarks jointly cover knowledge-intensive QA, commonsense reasoning, and reading comprehension, and we report both task-wise scores and their average.

**Additional Baselines.** To construct a more competitive comparison pool, we augment the main pruning baselines with several stronger sample- and token-level methods. For sample-level selection, we evaluate: (i) **Random**, which subsamples training instances uniformly; (ii) **Alpagasus** (Chen et al., 2023); (iii) **Deita** (Liu et al., 2023); (iv) **DS2** (Pang et al., 2024); and (v) **Less** (Xia et al., 2024a). Among these, Alpagasus, Deita, and DS2 are LLM-based ranking approaches that score and select high-value instructions, while Less is a gradient-driven saliency method that identifies important samples based on estimated gradient contributions. For token-level pruning (Table 10), we additionally consider: (i) **Rho-1** (Lin et al., 2024), which scores tokens by their excess loss relative to a reference model; and (ii) **TokenCleaning** (Pang et al., 2025), which removes tokens deemed noisy or uninformative based on token-level statistics.

## G.2 RESULTS ON QWEN3-8B

Table 7: Evaluation on OpenHermes dataset on Qwen3-8B under different sample ratios (12.5%, 25%) and token ratios (50%, 70%), reported on ARC-E, ARC-C, GSM8K, SQuAD, TriviaQA, and their average, where ↑ and ↓ respectively denote improvements or degradations over the *Random-Random* baseline.

| Sample Pruner | Token Pruner | Qwen3-8B | | | | | |
|---|---|---|---|---|---|---|---|
| | | ARC-E | ARC-C | GSM8K | SQuAD | TriviaQA | Avg. |
| Zero-Shot | | 69.31 | 44.41 | 74.45 | 27.63 | 51.76 | 53.51 |
| *12.5% Samples, 50% Tokens* | | | | | | | |
| Random | Random | 76.54 | 51.53 | 60.88 | 35.74 | 40.13 | 52.96 |
| | PPL | $77.60^{\uparrow 1.06}$ | $52.20^{\uparrow 0.67}$ | $58.00^{\downarrow 2.88}$ | $35.55^{\downarrow 0.19}$ | $40.73^{\uparrow 0.60}$ | $52.82^{\downarrow 0.14}$ |
| | FastV | $76.19^{\downarrow 0.35}$ | $52.54^{\uparrow 1.01}$ | $56.48^{\downarrow 4.40}$ | $35.04^{\downarrow 0.70}$ | $41.07^{\uparrow 0.94}$ | $52.26^{\downarrow 0.70}$ |
| | SparseVLM | $70.90^{\downarrow 5.64}$ | $45.42^{\downarrow 6.11}$ | $73.69^{\uparrow 12.81}$ | $24.39^{\downarrow 11.35}$ | **$50.58^{\uparrow 10.45}$** | $53.00^{\uparrow 0.04}$ |
| Longest | Random | $78.48^{\uparrow 1.94}$ | $53.22^{\uparrow 1.69}$ | $62.70^{\uparrow 1.82}$ | $33.97^{\downarrow 1.77}$ | $42.13^{\uparrow 2.00}$ | $54.10^{\uparrow 1.14}$ |
| | PPL | $78.48^{\uparrow 1.94}$ | $53.22^{\uparrow 1.69}$ | $56.18^{\downarrow 4.70}$ | $32.05^{\downarrow 3.69}$ | $41.91^{\uparrow 1.78}$ | $52.37^{\downarrow 0.59}$ |
| | FastV | $77.60^{\uparrow 1.06}$ | $51.86^{\uparrow 0.33}$ | $60.12^{\downarrow 0.76}$ | $32.57^{\downarrow 3.17}$ | $43.01^{\uparrow 2.88}$ | $53.03^{\uparrow 0.07}$ |
| | SparseVLM | $69.14^{\downarrow 7.40}$ | $44.75^{\downarrow 6.78}$ | $75.59^{\uparrow 14.71}$ | $22.24^{\downarrow 13.50}$ | $50.02^{\uparrow 9.89}$ | $52.35^{\downarrow 0.61}$ |
| InfoBatch | Random | $80.25^{\uparrow 3.71}$ | $53.56^{\uparrow 2.03}$ | $60.58^{\downarrow 0.30}$ | $38.51^{\uparrow 2.77}$ | $44.36^{\uparrow 4.23}$ | $55.45^{\uparrow 2.49}$ |
| | PPL | $79.01^{\uparrow 2.47}$ | $53.56^{\uparrow 2.03}$ | $62.47^{\uparrow 1.59}$ | $35.69^{\downarrow 0.05}$ | $42.06^{\uparrow 1.93}$ | $54.56^{\uparrow 1.60}$ |
| | FastV | $79.01^{\uparrow 2.47}$ | $51.19^{\downarrow 0.34}$ | $45.49^{\downarrow 15.39}$ | $38.71^{\uparrow 2.97}$ | $44.72^{\uparrow 4.59}$ | $51.82^{\downarrow 1.14}$ |
| | SparseVLM | $68.96^{\downarrow 7.58}$ | $44.41^{\downarrow 7.12}$ | **$77.71^{\uparrow 16.83}$** | $19.88^{\downarrow 15.86}$ | $49.35^{\uparrow 9.22}$ | $52.06^{\downarrow 0.90}$ |
| Entropy | Random | $76.37^{\downarrow 0.17}$ | $51.53^{\uparrow 0.00}$ | $62.77^{\uparrow 1.89}$ | $33.90^{\downarrow 1.84}$ | $43.07^{\uparrow 2.94}$ | $53.53^{\uparrow 0.57}$ |
| | PPL | $78.66^{\uparrow 2.12}$ | $52.54^{\uparrow 1.01}$ | $61.94^{\uparrow 1.06}$ | $35.62^{\downarrow 0.12}$ | $42.73^{\uparrow 2.60}$ | $54.30^{\uparrow 1.34}$ |
| | FastV | $76.90^{\uparrow 0.36}$ | $53.56^{\uparrow 2.03}$ | $65.05^{\uparrow 4.17}$ | $36.14^{\uparrow 0.40}$ | $43.99^{\uparrow 3.86}$ | $55.13^{\uparrow 2.17}$ |
| | SparseVLM | $71.08^{\downarrow 5.46}$ | $44.07^{\downarrow 7.46}$ | $75.89^{\uparrow 15.01}$ | $22.77^{\downarrow 13.00}$ | $50.47^{\uparrow 10.34}$ | $52.85^{\downarrow 0.11}$ |
| **Q-Tuning (Ours)** | | **$80.95^{\uparrow 4.41}$** | **$53.56^{\uparrow 2.03}$** | $70.51^{\uparrow 9.63}$ | **$45.76^{\uparrow 10.02}$** | $50.16^{\uparrow 10.03}$ | **$60.19^{\uparrow 7.23}$** |
| *12.5% Samples, 70% Tokens* | | | | | | | |
| Random | Random | 76.54 | 52.20 | 61.64 | 34.88 | 41.02 | 53.26 |
| | PPL | $77.07^{\uparrow 0.53}$ | $52.20^{\uparrow 0.00}$ | $59.21^{\downarrow 2.43}$ | $35.20^{\uparrow 0.32}$ | $40.87^{\downarrow 0.15}$ | $52.91^{\downarrow 0.35}$ |
| | FastV | $76.37^{\downarrow 0.17}$ | $50.51^{\downarrow 1.69}$ | $59.36^{\downarrow 2.28}$ | $35.25^{\uparrow 0.37}$ | $41.01^{\downarrow 0.01}$ | $52.50^{\downarrow 0.76}$ |
| | SparseVLM | $70.90^{\downarrow 5.64}$ | $45.42^{\downarrow 6.78}$ | $73.69^{\uparrow 12.05}$ | $24.39^{\downarrow 10.49}$ | $50.58^{\uparrow 9.56}$ | $53.00^{\downarrow 0.26}$ |
| Longest | Random | $79.19^{\uparrow 2.65}$ | **$53.90^{\uparrow 1.70}$** | $61.64^{\uparrow 0.00}$ | $34.14^{\downarrow 0.74}$ | $42.23^{\uparrow 1.21}$ | $54.22^{\uparrow 0.96}$ |
| | PPL | $78.48^{\uparrow 1.94}$ | $51.53^{\downarrow 0.67}$ | $55.65^{\downarrow 5.99}$ | $32.95^{\downarrow 1.93}$ | $42.45^{\uparrow 1.43}$ | $52.21^{\downarrow 1.05}$ |
| | FastV | $78.13^{\uparrow 1.59}$ | $51.53^{\downarrow 0.67}$ | $59.21^{\downarrow 2.43}$ | $33.16^{\downarrow 1.72}$ | $42.19^{\uparrow 1.17}$ | $52.84^{\downarrow 0.42}$ |
| | SparseVLM | $69.14^{\downarrow 7.40}$ | $44.75^{\downarrow 7.45}$ | $75.59^{\uparrow 13.95}$ | $22.24^{\downarrow 12.64}$ | $50.02^{\uparrow 9.00}$ | $52.35^{\downarrow 0.91}$ |
| InfoBatch | Random | $78.84^{\uparrow 2.30}$ | $51.86^{\downarrow 0.34}$ | $58.38^{\downarrow 3.26}$ | $38.12^{\uparrow 3.24}$ | $39.98^{\downarrow 1.04}$ | $53.44^{\uparrow 0.18}$ |
| | PPL | $78.48^{\uparrow 1.94}$ | $52.88^{\uparrow 0.68}$ | $62.17^{\uparrow 0.53}$ | $36.28^{\uparrow 1.40}$ | $42.96^{\uparrow 1.94}$ | $54.55^{\uparrow 1.29}$ |
| | FastV | $79.89^{\uparrow 3.35}$ | $53.22^{\uparrow 1.02}$ | $55.19^{\downarrow 6.45}$ | $38.63^{\uparrow 3.75}$ | $44.23^{\uparrow 3.21}$ | $54.23^{\uparrow 0.97}$ |
| | SparseVLM | $68.96^{\downarrow 7.58}$ | $44.41^{\downarrow 7.79}$ | **$77.71^{\uparrow 16.07}$** | $19.88^{\downarrow 15.00}$ | $49.35^{\uparrow 8.33}$ | $52.06^{\downarrow 1.20}$ |
| Entropy | Random | $77.07^{\downarrow 0.53}$ | $51.86^{\downarrow 0.34}$ | $60.73^{\downarrow 0.91}$ | $34.30^{\downarrow 0.58}$ | $42.34^{\uparrow 1.32}$ | $53.26^{\uparrow 0.00}$ |
| | PPL | $78.84^{\uparrow 2.30}$ | $52.88^{\uparrow 0.68}$ | $61.26^{\downarrow 0.38}$ | $36.44^{\uparrow 1.56}$ | $43.53^{\uparrow 2.51}$ | $54.59^{\uparrow 1.33}$ |
| | FastV | $77.07^{\downarrow 0.53}$ | $51.86^{\downarrow 0.34}$ | $64.22^{\uparrow 2.58}$ | $36.63^{\uparrow 1.75}$ | $43.94^{\uparrow 2.92}$ | $54.74^{\uparrow 1.48}$ |
| | SparseVLM | $71.08^{\downarrow 5.46}$ | $44.07^{\downarrow 8.13}$ | $75.89^{\uparrow 14.25}$ | $22.77^{\downarrow 12.11}$ | $50.47^{\uparrow 9.45}$ | $52.85^{\downarrow 0.41}$ |
| **Q-Tuning (Ours)** | | **$80.95^{\uparrow 4.41}$** | $52.20^{\uparrow 0.00}$ | $70.13^{\uparrow 8.49}$ | **$47.60^{\uparrow 12.72}$** | $50.58^{\uparrow 9.56}$ | **$60.29^{\uparrow 7.03}$** |
| *25% Samples, 50% Tokens* | | | | | | | |
| Random | Random | 77.07 | 51.86 | 61.03 | 35.11 | 40.55 | 53.13 |
| | PPL | $77.78^{\uparrow 0.71}$ | $54.58^{\uparrow 2.72}$ | $58.53^{\downarrow 2.50}$ | $35.43^{\uparrow 0.32}$ | $40.68^{\uparrow 0.13}$ | $53.40^{\uparrow 0.27}$ |
| | FastV | $79.54^{\uparrow 2.47}$ | $52.54^{\uparrow 0.68}$ | $53.45^{\downarrow 7.58}$ | $36.89^{\uparrow 1.78}$ | $43.13^{\uparrow 2.58}$ | $53.11^{\downarrow 0.02}$ |
| | SparseVLM | $69.14^{\downarrow 7.93}$ | $45.76^{\downarrow 6.10}$ | **$77.71^{\uparrow 16.68}$** | $20.80^{\downarrow 14.31}$ | $50.20^{\uparrow 9.65}$ | $52.72^{\downarrow 0.41}$ |
| Longest | Random | $79.01^{\uparrow 1.94}$ | $54.58^{\uparrow 2.72}$ | $51.10^{\downarrow 9.93}$ | $36.30^{\uparrow 1.19}$ | $41.93^{\uparrow 1.38}$ | $52.58^{\downarrow 0.55}$ |
| | PPL | $79.01^{\uparrow 1.94}$ | **$54.92^{\uparrow 3.06}$** | $55.42^{\downarrow 5.61}$ | $35.10^{\downarrow 0.01}$ | $41.09^{\uparrow 0.54}$ | $53.11^{\downarrow 0.02}$ |
| | FastV | **$80.95^{\uparrow 3.88}$** | $51.53^{\downarrow 0.33}$ | $53.30^{\downarrow 7.73}$ | $39.73^{\uparrow 4.62}$ | $44.34^{\uparrow 3.79}$ | $53.97^{\uparrow 0.84}$ |
| | SparseVLM | $72.49^{\downarrow 4.58}$ | $44.41^{\downarrow 7.45}$ | $74.83^{\uparrow 13.80}$ | $22.08^{\downarrow 13.03}$ | $48.93^{\uparrow 8.38}$ | $52.55^{\downarrow 0.58}$ |
| InfoBatch | Random | $77.25^{\uparrow 0.18}$ | $51.53^{\downarrow 0.33}$ | $63.61^{\uparrow 2.58}$ | $36.99^{\uparrow 1.88}$ | $45.15^{\uparrow 4.60}$ | $54.91^{\uparrow 1.78}$ |
| | PPL | $78.84^{\uparrow 1.77}$ | $54.58^{\uparrow 2.72}$ | $60.20^{\downarrow 0.83}$ | $35.81^{\uparrow 0.70}$ | $43.53^{\uparrow 3.00}$ | $54.59^{\uparrow 1.46}$ |
| | FastV | $80.25^{\uparrow 3.18}$ | $52.54^{\uparrow 0.68}$ | $48.52^{\downarrow 12.51}$ | $39.62^{\uparrow 4.51}$ | $44.87^{\uparrow 4.32}$ | $53.16^{\uparrow 0.03}$ |
| | SparseVLM | $69.49^{\downarrow 7.58}$ | $44.07^{\downarrow 7.79}$ | $76.88^{\uparrow 15.85}$ | $18.29^{\downarrow 16.82}$ | $50.16^{\uparrow 9.61}$ | $51.78^{\downarrow 1.35}$ |
| Entropy | Random | $77.07^{\uparrow 0.00}$ | $53.22^{\uparrow 1.36}$ | $62.02^{\uparrow 1.00}$ | $35.40^{\uparrow 0.29}$ | $42.91^{\uparrow 2.36}$ | $54.12^{\uparrow 0.99}$ |
| | PPL | $79.37^{\uparrow 2.30}$ | $54.24^{\uparrow 2.38}$ | $52.24^{\downarrow 8.79}$ | $33.13^{\downarrow 1.98}$ | $40.82^{\uparrow 0.27}$ | $51.96^{\downarrow 1.17}$ |
| | FastV | $79.54^{\uparrow 2.47}$ | $51.86^{\uparrow 0.00}$ | $52.92^{\downarrow 8.11}$ | $35.68^{\uparrow 0.57}$ | $43.79^{\uparrow 3.24}$ | $52.76^{\downarrow 0.37}$ |
| | SparseVLM | $70.19^{\downarrow 6.88}$ | $43.39^{\downarrow 8.47}$ | $75.21^{\uparrow 14.18}$ | $19.97^{\downarrow 15.14}$ | $50.46^{\uparrow 9.91}$ | $51.84^{\downarrow 1.29}$ |
| **Q-Tuning (Ours)** | | $79.89^{\uparrow 2.82}$ | $52.88^{\uparrow 1.02}$ | $69.37^{\uparrow 8.34}$ | **$43.54^{\uparrow 8.43}$** | $48.33^{\uparrow 7.78}$ | **$58.80^{\uparrow 5.67}$** |
| Full Dataset | | 77.78 | 52.20 | 65.13 | 38.05 | 44.08 | 55.45 |

Table 8: Evaluation on OpenHermes dataset on Qwen3-8B under different sample ratios (25%, 50%) and token ratios (50%, 70%), reported on ARC-E, ARC-C, GSM8K, SQuAD, TriviaQA, and their average, where ↑ and ↓ respectively denote improvements or degradations over the *Random-Random* baseline.

| Sample Pruner | Token Pruner | Qwen3-8B | | | | | |
|---|---|---|---|---|---|---|---|
| | | ARC-E | ARC-C | GSM8K | SQuAD | TriviaQA | Avg. |
| Zero-Shot | | 69.31 | 44.41 | 74.45 | 27.63 | 51.76 | 53.51 |
| *25% Samples, 70% Tokens* | | | | | | | |
| Random | Random | 77.43 | 52.20 | 57.85 | 35.55 | 41.47 | 52.90 |
| | PPL | $76.90^{\downarrow0.53}$ | $53.90^{\uparrow1.70}$ | $60.27^{\uparrow2.42}$ | $35.70^{\downarrow0.15}$ | $40.73^{\downarrow0.74}$ | $53.50^{\uparrow0.60}$ |
| | FastV | $79.19^{\uparrow1.76}$ | $51.86^{\downarrow0.34}$ | $56.03^{\downarrow1.82}$ | $36.01^{\uparrow0.46}$ | $43.86^{\uparrow2.39}$ | $53.39^{\uparrow0.49}$ |
| | SparseVLM | $69.14^{\downarrow8.29}$ | $45.76^{\downarrow6.44}$ | $77.71^{\uparrow19.86}$ | $20.80^{\downarrow14.75}$ | $50.20^{\uparrow8.73}$ | $52.72^{\downarrow0.18}$ |
| Longest | Random | $79.01^{\uparrow1.58}$ | $54.58^{\uparrow2.38}$ | $53.83^{\downarrow4.02}$ | $37.56^{\uparrow2.01}$ | $42.40^{\uparrow0.93}$ | $53.48^{\uparrow0.58}$ |
| | PPL | $78.48^{\uparrow1.05}$ | $\mathbf{55.59}^{\uparrow3.39}$ | $55.72^{\uparrow2.13}$ | $35.16^{\downarrow0.39}$ | $41.31^{\downarrow0.16}$ | $53.25^{\uparrow0.35}$ |
| | FastV | $79.89^{\uparrow2.46}$ | $52.54^{\uparrow0.34}$ | $53.22^{\downarrow4.63}$ | $37.90^{\uparrow2.35}$ | $43.79^{\uparrow2.32}$ | $53.47^{\uparrow0.57}$ |
| | SparseVLM | $72.49^{\downarrow4.94}$ | $44.41^{\downarrow7.79}$ | $74.83^{\uparrow16.98}$ | $22.08^{\downarrow13.47}$ | $48.93^{\uparrow7.46}$ | $52.55^{\downarrow0.35}$ |
| InfoBatch | Random | $78.66^{\uparrow1.23}$ | $52.54^{\uparrow0.34}$ | $63.68^{\uparrow5.83}$ | $37.56^{\uparrow2.01}$ | $44.51^{\uparrow3.04}$ | $55.39^{\uparrow2.49}$ |
| | PPL | $78.84^{\uparrow1.41}$ | $53.22^{\uparrow1.02}$ | $62.40^{\uparrow4.55}$ | $36.39^{\uparrow0.84}$ | $44.87^{\uparrow3.40}$ | $55.14^{\uparrow2.24}$ |
| | FastV | $\mathbf{79.89}^{\uparrow2.46}$ | $53.90^{\uparrow1.70}$ | $55.04^{\downarrow2.82}$ | $38.79^{\uparrow3.24}$ | $44.28^{\uparrow2.81}$ | $54.38^{\uparrow1.48}$ |
| | SparseVLM | $69.49^{\downarrow7.94}$ | $44.07^{\downarrow8.13}$ | $76.88^{\uparrow19.03}$ | $18.29^{\downarrow17.26}$ | $50.16^{\uparrow8.69}$ | $51.78^{\downarrow1.12}$ |
| Entropy | Random | $76.90^{\downarrow0.53}$ | $52.54^{\uparrow0.34}$ | $57.47^{\downarrow0.38}$ | $34.58^{\downarrow0.97}$ | $42.50^{\uparrow1.03}$ | $52.80^{\downarrow0.10}$ |
| | PPL | $78.66^{\uparrow1.23}$ | $53.90^{\uparrow1.70}$ | $52.46^{\downarrow5.39}$ | $33.72^{\downarrow1.83}$ | $41.16^{\downarrow0.31}$ | $51.98^{\downarrow0.92}$ |
| | FastV | $79.01^{\uparrow1.58}$ | $52.20^{\uparrow0.00}$ | $57.54^{\uparrow0.31}$ | $35.72^{\uparrow0.17}$ | $42.94^{\uparrow1.47}$ | $53.48^{\uparrow0.58}$ |
| | SparseVLM | $70.19^{\downarrow7.24}$ | $43.39^{\downarrow8.81}$ | $75.21^{\uparrow17.36}$ | $19.97^{\downarrow15.58}$ | $\mathbf{50.46}^{\uparrow9.00}$ | $51.84^{\downarrow1.06}$ |
| **Q-Tuning (Ours)** | | $79.72^{\uparrow2.29}$ | $52.88^{\uparrow0.68}$ | $71.11^{\uparrow13.26}$ | $\mathbf{43.70}^{\uparrow8.15}$ | $48.58^{\uparrow7.11}$ | $\mathbf{59.20}^{\uparrow6.30}$ |
| *50% Samples, 50% Tokens* | | | | | | | |
| Random | Random | 77.95 | 51.86 | 63.61 | 36.64 | 42.22 | 54.46 |
| | PPL | $78.48^{\uparrow0.53}$ | $53.22^{\uparrow1.36}$ | $62.02^{\downarrow1.59}$ | $35.96^{\downarrow0.68}$ | $42.56^{\uparrow0.34}$ | $54.45^{\downarrow0.01}$ |
| | FastV | $79.19^{\uparrow1.24}$ | $51.53^{\downarrow0.33}$ | $49.73^{\downarrow13.88}$ | $40.64^{\uparrow4.00}$ | $44.45^{\uparrow2.23}$ | $53.11^{\downarrow1.35}$ |
| | SparseVLM | $72.13^{\downarrow5.82}$ | $46.10^{\downarrow5.76}$ | $73.69^{\uparrow10.08}$ | $18.74^{\downarrow17.90}$ | $\mathbf{50.39}^{\uparrow8.17}$ | $52.21^{\downarrow2.25}$ |
| Longest | Random | $78.84^{\uparrow0.89}$ | $54.24^{\uparrow2.38}$ | $59.67^{\downarrow3.94}$ | $36.66^{\uparrow0.02}$ | $43.05^{\uparrow0.83}$ | $54.49^{\uparrow0.03}$ |
| | PPL | $79.72^{\uparrow1.77}$ | $\mathbf{54.24}^{\uparrow2.38}$ | $58.15^{\downarrow5.46}$ | $39.88^{\uparrow3.24}$ | $44.01^{\uparrow1.79}$ | $55.20^{\uparrow0.74}$ |
| | FastV | $79.72^{\uparrow1.77}$ | $51.86^{\uparrow0.00}$ | $47.99^{\downarrow15.62}$ | $40.57^{\uparrow3.93}$ | $44.89^{\uparrow2.67}$ | $53.01^{\downarrow1.45}$ |
| | SparseVLM | $70.19^{\downarrow7.76}$ | $43.39^{\downarrow8.47}$ | $74.22^{\uparrow10.61}$ | $17.90^{\downarrow18.74}$ | $49.34^{\uparrow7.12}$ | $51.01^{\downarrow3.45}$ |
| InfoBatch | Random | $78.84^{\uparrow0.89}$ | $52.88^{\uparrow1.02}$ | $64.37^{\uparrow0.76}$ | $36.98^{\uparrow0.34}$ | $45.49^{\uparrow3.27}$ | $55.71^{\uparrow1.25}$ |
| | PPL | $77.95^{\uparrow0.00}$ | $53.56^{\uparrow1.70}$ | $60.73^{\downarrow2.88}$ | $38.10^{\uparrow1.47}$ | $44.39^{\uparrow2.17}$ | $54.95^{\uparrow0.49}$ |
| | FastV | $\mathbf{80.42}^{\uparrow2.47}$ | $52.88^{\uparrow1.02}$ | $50.34^{\downarrow13.27}$ | $\mathbf{40.79}^{\uparrow4.15}$ | $45.14^{\uparrow2.92}$ | $53.92^{\downarrow0.54}$ |
| | SparseVLM | $66.49^{\downarrow11.46}$ | $43.39^{\downarrow8.42}$ | $74.30^{\uparrow10.69}$ | $15.16^{\downarrow21.48}$ | $49.64^{\uparrow7.42}$ | $49.80^{\downarrow4.66}$ |
| Entropy | Random | $78.13^{\uparrow0.18}$ | $51.53^{\downarrow0.33}$ | $61.94^{\downarrow1.67}$ | $34.59^{\downarrow2.05}$ | $42.23^{\uparrow0.01}$ | $53.68^{\downarrow0.78}$ |
| | PPL | $78.66^{\uparrow0.71}$ | $52.88^{\uparrow1.02}$ | $54.97^{\downarrow8.61}$ | $36.11^{\downarrow0.53}$ | $42.06^{\downarrow0.16}$ | $52.94^{\downarrow1.52}$ |
| | FastV | $79.54^{\uparrow1.59}$ | $53.22^{\uparrow1.36}$ | $47.08^{\downarrow16.53}$ | $40.24^{\uparrow3.58}$ | $43.33^{\uparrow1.11}$ | $52.68^{\downarrow1.78}$ |
| | SparseVLM | $70.90^{\downarrow7.05}$ | $43.39^{\downarrow8.47}$ | $78.70^{\uparrow15.09}$ | $16.25^{\downarrow20.39}$ | $50.10^{\uparrow7.87}$ | $51.87^{\downarrow2.59}$ |
| **Q-Tuning (Ours)** | | $80.25^{\uparrow2.30}$ | $53.90^{\uparrow2.04}$ | $70.05^{\uparrow6.44}$ | $39.03^{\uparrow2.39}$ | $48.70^{\uparrow6.48}$ | $58.39^{\uparrow3.93}$ |
| *50% Samples, 70% Tokens* | | | | | | | |
| Random | Random | 78.13 | 52.20 | 61.49 | 36.91 | 42.75 | 54.30 |
| | PPL | $77.95^{\downarrow0.18}$ | $52.20^{\uparrow0.00}$ | $59.59^{\downarrow1.90}$ | $36.14^{\downarrow0.77}$ | $42.51^{\downarrow0.24}$ | $53.68^{\downarrow0.62}$ |
| | FastV | $79.01^{\uparrow0.88}$ | $53.90^{\uparrow1.70}$ | $58.30^{\downarrow3.19}$ | $40.33^{\uparrow3.42}$ | $45.20^{\uparrow2.45}$ | $55.35^{\uparrow1.05}$ |
| | SparseVLM | $72.13^{\downarrow6.00}$ | $46.10^{\downarrow6.10}$ | $73.69^{\uparrow12.20}$ | $18.74^{\downarrow18.17}$ | $\mathbf{50.39}^{\uparrow7.64}$ | $52.21^{\downarrow2.09}$ |
| Longest | Random | $78.48^{\uparrow0.35}$ | $52.54^{\uparrow0.34}$ | $59.14^{\downarrow2.35}$ | $37.27^{\uparrow0.36}$ | $42.38^{\downarrow0.37}$ | $53.96^{\downarrow0.34}$ |
| | PPL | $79.19^{\uparrow1.06}$ | $53.56^{\uparrow1.36}$ | $60.96^{\downarrow0.53}$ | $39.78^{\uparrow2.87}$ | $44.17^{\uparrow1.42}$ | $55.53^{\uparrow1.23}$ |
| | FastV | $\mathbf{80.25}^{\uparrow2.12}$ | $52.20^{\uparrow0.00}$ | $56.79^{\downarrow4.70}$ | $\mathbf{40.53}^{\uparrow3.62}$ | $45.62^{\uparrow2.87}$ | $55.08^{\uparrow0.78}$ |
| | SparseVLM | $70.19^{\downarrow7.94}$ | $43.39^{\downarrow8.81}$ | $74.22^{\uparrow12.73}$ | $17.90^{\downarrow19.01}$ | $49.34^{\uparrow6.59}$ | $51.01^{\downarrow3.29}$ |
| InfoBatch | Random | $78.31^{\uparrow0.18}$ | $52.88^{\uparrow0.68}$ | $62.32^{\uparrow0.83}$ | $35.55^{\downarrow1.36}$ | $45.15^{\uparrow2.40}$ | $54.84^{\uparrow0.54}$ |
| | PPL | $78.31^{\uparrow0.18}$ | $54.58^{\uparrow2.38}$ | $63.23^{\uparrow1.74}$ | $38.89^{\uparrow1.98}$ | $44.16^{\uparrow1.41}$ | $55.83^{\uparrow1.53}$ |
| | FastV | $79.01^{\uparrow0.88}$ | $52.88^{\uparrow0.68}$ | $58.30^{\downarrow3.19}$ | $39.05^{\uparrow2.14}$ | $46.05^{\uparrow3.30}$ | $55.06^{\uparrow0.76}$ |
| | SparseVLM | $66.49^{\downarrow11.64}$ | $43.39^{\downarrow8.81}$ | $74.30^{\uparrow12.81}$ | $15.16^{\downarrow21.75}$ | $49.64^{\uparrow6.89}$ | $49.80^{\downarrow4.50}$ |
| Entropy | Random | $78.48^{\uparrow0.35}$ | $53.22^{\uparrow1.02}$ | $60.65^{\downarrow0.84}$ | $35.64^{\downarrow1.27}$ | $42.31^{\downarrow0.44}$ | $54.06^{\downarrow0.24}$ |
| | PPL | $78.13^{\uparrow0.00}$ | $53.22^{\uparrow1.02}$ | $57.85^{\downarrow3.64}$ | $36.85^{\downarrow0.06}$ | $41.70^{\downarrow1.05}$ | $53.55^{\downarrow0.75}$ |
| | FastV | $79.37^{\uparrow1.24}$ | $52.88^{\uparrow0.68}$ | $52.62^{\downarrow8.87}$ | $39.76^{\uparrow2.85}$ | $44.18^{\uparrow1.43}$ | $53.76^{\downarrow0.54}$ |
| | SparseVLM | $70.90^{\downarrow7.23}$ | $43.39^{\downarrow8.81}$ | $78.70^{\uparrow17.21}$ | $16.25^{\downarrow20.66}$ | $50.10^{\uparrow7.79}$ | $51.87^{\downarrow2.43}$ |
| **Q-Tuning (Ours)** | | $78.84^{\uparrow0.71}$ | $52.88^{\uparrow0.68}$ | $69.90^{\uparrow8.41}$ | $39.54^{\uparrow2.63}$ | $48.15^{\uparrow5.40}$ | $57.86^{\uparrow3.56}$ |
| Full Dataset | | 77.78 | 52.20 | 65.13 | 38.05 | 44.08 | 55.45 |

The detailed results on the OpenHermes dataset using Qwen3-8B (Yang et al., 2025) are summarized in Tables 7 and 8, which systematically evaluate different sample ratios and token ratios.

**Strong performance with limited data budgets.** Across all evaluated budgets, Q-Tuning consistently outperforms traditional pruning-based fine-tuning methods and surpasses the full-data baseline. For instance, under the extremely low-budget setting of 12.5% samples and 50% tokens, Q-Tuning reaches an average of 60.19, outperforming the full-dataset result (55.45) by more than +4.7 points. Even at 12.5% samples, 70% tokens, Q-Tuning maintains the same strong average (60.29), showing remarkable robustness under aggressive pruning.

**Consistent superiority under larger budgets.** When the available data increases, Q-Tuning continues to dominate other baselines. At 25% samples, 50% tokens, Q-Tuning attains an average score of 58.80, exceeding the strongest baseline (InfoBatch–Random) by nearly +3.9 points. Similarly, at 25% samples, 70% tokens, it reaches 59.20, still outperforming all competing methods such as InfoBatch, Entropy, and SparseVLM by a clear margin, and exceeding the full-data fine-tuning by +3.75 points.

**Performance saturation and stable generalization.** At even larger budgets, such as 50% samples with 50% or 70% tokens, Q-Tuning continues to deliver strong and stable improvements. In the 50% / 50% setting, it achieves an average of 58.39, improving over all pruning-based baselines by 3–6 points. Under 50% / 70%, it reaches 57.86, maintaining competitive accuracy and demonstrating consistent generalization despite heavy token-level sparsity.

## G.3 RESULTS ON ADDITIONAL BASELINES

Table 9: Evaluation on Wizard dataset under different sample ratios (12.5%, 25%) and token ratios (50%, 100%) with the same token pruner (*Random*), conducted on additional baselines. Reported benchmarks include ARC-E, ARC-C, GSM8K, SQuAD, TriviaQA, and their average, where ↑ and ↓ respectively denote improvements or degradations over the *Random* baseline.

| Sample Pruner | Token Ratio | LLaMA2-7B | | | | | | Mistral-7B | | | | | |
|---|---|---|---|---|---|---|---|---|---|---|---|---|---|
| | | ARC-E | ARC-C | GSM8K | SQuAD | TriviaQA | Avg. | ARC-E | ARC-C | GSM8K | SQuAD | TriviaQA | Avg. |
| Zero-Shot | | 53.44 | 38.98 | 5.31 | 12.18 | 43.00 | 30.58 | 66.67 | 46.10 | 18.35 | 10.01 | 43.77 | 36.98 |
| *12.5% Samples* | | | | | | | | | | | | | |
| Random | 50% | 59.25 | 41.02 | 8.11 | 12.75 | 48.75 | 33.98 | 70.55 | 48.14 | 22.74 | 19.57 | 52.63 | 42.73 |
| Alpagasus | 50% | 58.91$^{\downarrow0.34}$ | 41.69$^{\uparrow0.67}$ | 6.67$^{\downarrow1.44}$ | 12.81$^{\uparrow0.06}$ | 46.28$^{\downarrow2.47}$ | 33.27$^{\downarrow0.71}$ | 72.84$^{\uparrow2.29}$ | 48.47$^{\uparrow0.33}$ | 20.55$^{\downarrow2.19}$ | 20.41$^{\uparrow0.84}$ | 52.21$^{\downarrow0.42}$ | 42.90$^{\uparrow0.17}$ |
| | 100% | 58.91$^{\downarrow0.34}$ | 41.36$^{\uparrow0.34}$ | 7.58$^{\downarrow0.53}$ | 13.45$^{\uparrow0.70}$ | 46.96$^{\downarrow1.79}$ | 33.65$^{\downarrow0.33}$ | 70.72$^{\uparrow0.17}$ | 48.81$^{\uparrow0.67}$ | 23.35$^{\uparrow0.61}$ | 24.30$^{\uparrow4.73}$ | 52.71$^{\uparrow0.08}$ | 43.98$^{\uparrow1.25}$ |
| Deita | 50% | 58.20$^{\downarrow1.05}$ | 42.37$^{\uparrow1.35}$ | 7.43$^{\downarrow0.68}$ | 13.85$^{\uparrow1.10}$ | 45.59$^{\downarrow3.16}$ | 33.49$^{\downarrow0.49}$ | 72.66$^{\uparrow2.11}$ | 50.51$^{\uparrow2.37}$ | 22.97$^{\downarrow0.23}$ | 20.91$^{\uparrow1.34}$ | 54.17$^{\uparrow1.54}$ | 44.25$^{\uparrow1.52}$ |
| | 100% | 58.38$^{\downarrow0.87}$ | 42.37$^{\uparrow1.35}$ | 5.46$^{\downarrow2.65}$ | 13.99$^{\uparrow1.24}$ | 45.63$^{\downarrow3.12}$ | 33.17$^{\downarrow0.81}$ | 72.13$^{\uparrow1.58}$ | 50.17$^{\uparrow2.03}$ | 22.51$^{\downarrow0.23}$ | 24.46$^{\uparrow4.89}$ | 55.28$^{\uparrow2.65}$ | 44.92$^{\uparrow2.19}$ |
| DS2 | 50% | 58.73$^{\downarrow0.52}$ | 42.37$^{\uparrow1.35}$ | 5.99$^{\downarrow2.12}$ | 15.05$^{\uparrow2.30}$ | 45.51$^{\downarrow3.24}$ | 33.53$^{\downarrow0.45}$ | 71.25$^{\uparrow0.70}$ | 47.46$^{\downarrow0.68}$ | 21.23$^{\downarrow1.51}$ | 21.57$^{\uparrow2.00}$ | 51.23$^{\downarrow1.40}$ | 42.55$^{\downarrow0.18}$ |
| | 100% | 59.08$^{\downarrow0.17}$ | 41.36$^{\uparrow0.34}$ | 6.67$^{\downarrow1.44}$ | 15.14$^{\uparrow2.39}$ | 45.69$^{\downarrow3.06}$ | 33.59$^{\downarrow0.39}$ | 72.13$^{\uparrow1.58}$ | 49.83$^{\uparrow1.69}$ | 23.35$^{\uparrow0.61}$ | 23.99$^{\uparrow4.42}$ | 52.85$^{\uparrow0.22}$ | 44.43$^{\uparrow1.70}$ |
| Less | 50% | 58.20$^{\downarrow1.05}$ | 42.37$^{\uparrow1.35}$ | 6.67$^{\downarrow1.44}$ | 14.56$^{\uparrow1.81}$ | 46.62$^{\downarrow2.13}$ | 33.69$^{\downarrow0.29}$ | 73.19$^{\uparrow2.64}$ | 49.49$^{\uparrow1.35}$ | 22.90$^{\downarrow0.16}$ | 19.61$^{\uparrow0.04}$ | 53.14$^{\uparrow0.51}$ | 43.67$^{\uparrow0.94}$ |
| | 100 | 58.38$^{\downarrow0.87}$ | 43.39$^{\uparrow2.37}$ | 6.82$^{\downarrow1.29}$ | 16.16$^{\uparrow3.41}$ | 46.72$^{\downarrow2.03}$ | 34.30$^{\uparrow0.32}$ | 73.90$^{\uparrow3.35}$ | 48.47$^{\uparrow0.33}$ | 23.12$^{\uparrow0.38}$ | 21.67$^{\uparrow2.10}$ | 56.00$^{\uparrow3.37}$ | 44.63$^{\uparrow1.90}$ |
| **Q-Tuning (Ours)** | 50% | **64.20**$^{\uparrow4.95}$ | 42.03$^{\uparrow1.01}$ | **10.54**$^{\uparrow2.43}$ | **18.79**$^{\uparrow6.04}$ | **53.12**$^{\uparrow4.37}$ | **37.74**$^{\uparrow3.76}$ | 71.60$^{\uparrow1.05}$ | 48.14$^{\uparrow0.00}$ | **29.34**$^{\uparrow6.60}$ | **27.75**$^{\uparrow8.18}$ | **57.78**$^{\uparrow5.15}$ | **46.92**$^{\uparrow4.19}$ |
| *25% Samples* | | | | | | | | | | | | | |
| Random | 50% | 60.32 | 41.69 | 5.76 | 13.43 | 48.41 | 33.92 | 70.19 | 46.10 | 20.62 | 24.07 | 53.74 | 42.95 |
| Alpagasus | 50% | 59.61$^{\downarrow0.71}$ | 42.71$^{\uparrow1.02}$ | 5.91$^{\uparrow0.15}$ | 14.12$^{\uparrow0.69}$ | 47.30$^{\downarrow1.11}$ | 33.93$^{\uparrow0.01}$ | 70.37$^{\uparrow0.18}$ | 47.80$^{\uparrow1.70}$ | 20.85$^{\uparrow0.23}$ | 22.38$^{\downarrow1.69}$ | 52.87$^{\downarrow0.87}$ | 42.78$^{\downarrow0.17}$ |
| | 100% | 60.49$^{\uparrow0.17}$ | 43.39$^{\uparrow1.70}$ | 7.51$^{\uparrow1.75}$ | 13.16$^{\downarrow0.27}$ | 48.41$^{\uparrow0.00}$ | 34.59$^{\uparrow0.67}$ | 71.60$^{\uparrow1.41}$ | 50.17$^{\uparrow4.07}$ | 23.35$^{\uparrow2.73}$ | 22.38$^{\downarrow1.69}$ | 55.73$^{\uparrow1.99}$ | 44.65$^{\uparrow1.70}$ |
| Deita | 50% | 59.08$^{\downarrow1.24}$ | 41.69$^{\uparrow0.00}$ | 7.66$^{\uparrow1.90}$ | 13.23$^{\downarrow0.20}$ | 47.55$^{\downarrow0.86}$ | 33.84$^{\downarrow0.08}$ | 72.13$^{\uparrow1.94}$ | 47.80$^{\uparrow1.70}$ | 21.00$^{\uparrow0.38}$ | 23.16$^{\downarrow0.91}$ | 53.97$^{\uparrow0.23}$ | 43.61$^{\uparrow0.66}$ |
| | 100% | 58.73$^{\downarrow1.59}$ | 42.71$^{\uparrow1.02}$ | 4.92$^{\downarrow0.84}$ | 14.88$^{\uparrow1.45}$ | 48.24$^{\downarrow0.17}$ | 33.90$^{\downarrow0.02}$ | 72.66$^{\uparrow2.47}$ | 47.80$^{\uparrow1.70}$ | 22.29$^{\uparrow1.67}$ | 25.87$^{\uparrow1.80}$ | 54.40$^{\uparrow0.66}$ | 44.60$^{\uparrow1.65}$ |
| DS2 | 50% | 59.44$^{\downarrow0.88}$ | 42.03$^{\uparrow0.34}$ | 6.44$^{\uparrow0.68}$ | 15.59$^{\uparrow2.16}$ | 47.34$^{\downarrow1.07}$ | 34.17$^{\uparrow0.25}$ | 70.72$^{\uparrow0.53}$ | 48.47$^{\uparrow2.37}$ | 22.74$^{\uparrow2.12}$ | 21.15$^{\downarrow2.92}$ | 53.49$^{\downarrow0.25}$ | 43.32$^{\uparrow0.37}$ |
| | 100% | 59.26$^{\downarrow1.06}$ | 41.36$^{\downarrow0.33}$ | 7.05$^{\uparrow1.29}$ | 15.20$^{\uparrow1.77}$ | 48.76$^{\uparrow0.35}$ | 34.33$^{\uparrow0.41}$ | 71.43$^{\uparrow1.24}$ | 48.47$^{\uparrow2.37}$ | 22.44$^{\uparrow1.82}$ | 21.72$^{\downarrow2.35}$ | 53.76$^{\uparrow0.02}$ | 43.57$^{\uparrow0.62}$ |
| Less | 50% | 59.44$^{\downarrow0.88}$ | 43.39$^{\uparrow1.70}$ | 7.05$^{\uparrow1.29}$ | 15.89$^{\uparrow2.46}$ | 46.60$^{\downarrow1.81}$ | 34.47$^{\uparrow0.55}$ | 71.78$^{\uparrow1.59}$ | 47.80$^{\uparrow1.70}$ | 22.14$^{\uparrow1.52}$ | 24.51$^{\uparrow0.44}$ | 53.04$^{\downarrow0.70}$ | 43.85$^{\uparrow0.90}$ |
| | 100 | 59.79$^{\downarrow0.53}$ | 41.02$^{\downarrow0.67}$ | 6.37$^{\uparrow0.61}$ | 14.44$^{\uparrow1.01}$ | 48.44$^{\uparrow0.03}$ | 34.01$^{\uparrow0.09}$ | 71.60$^{\uparrow1.41}$ | 49.49$^{\uparrow3.39}$ | 21.68$^{\uparrow1.06}$ | 30.85$^{\uparrow6.78}$ | 53.40$^{\downarrow0.34}$ | 45.41$^{\uparrow2.46}$ |
| **Q-Tuning (Ours)** | 50% | **63.14**$^{\uparrow2.82}$ | 42.03$^{\uparrow0.34}$ | **8.87**$^{\uparrow3.11}$ | **16.76**$^{\uparrow3.33}$ | **51.52**$^{\uparrow3.11}$ | **36.47**$^{\uparrow2.55}$ | 71.78$^{\uparrow1.59}$ | 47.12$^{\uparrow1.02}$ | **26.08**$^{\uparrow5.46}$ | **32.79**$^{\uparrow8.72}$ | **56.17**$^{\uparrow2.43}$ | **46.79**$^{\uparrow3.84}$ |
| Full Dataset | | 61.55 | 42.37 | 8.64 | 13.80 | 50.45 | 35.36 | 71.25 | 45.76 | 26.68 | 31.81 | 53.67 | 45.84 |

Table 10: Evaluation on Wizard dataset under different sample ratios (12.5%, 25%, 100%) and fixed token ratio (50%) with the same sample pruner (*Random*), conducted on additional baselines. Reported benchmarks include ARC-E, ARC-C, GSM8K, SQuAD, TriviaQA, and their average, where ↑ and ↓ respectively denote improvements or degradations over the *Random* baseline.

| Token Pruner | Sample Ratio | LLaMA2-7B | | | | | | Mistral-7B | | | | | |
|---|---|---|---|---|---|---|---|---|---|---|---|---|---|
| | | ARC-E | ARC-C | GSM8K | SQuAD | TriviaQA | Avg. | ARC-E | ARC-C | GSM8K | SQuAD | TriviaQA | Avg. |
| Zero-Shot | | 53.44 | 38.98 | 5.31 | 12.18 | 43.00 | 30.58 | 66.67 | 46.10 | 18.35 | 10.01 | 43.77 | 36.98 |
| *50% Tokens* | | | | | | | | | | | | | |
| Random | 12.5% | 59.25 | 41.02 | 8.11 | 12.75 | 48.75 | 33.98 | 70.55 | 48.14 | 22.74 | 19.57 | 52.63 | 42.73 |
| | 25% | 60.32$^{\uparrow1.07}$ | 41.69$^{\uparrow0.67}$ | 5.76$^{\downarrow2.35}$ | 13.43$^{\uparrow0.68}$ | 48.41$^{\downarrow0.34}$ | 33.92$^{\downarrow0.06}$ | 70.19$^{\downarrow0.36}$ | 46.10$^{\downarrow2.04}$ | 20.62$^{\downarrow2.12}$ | 24.07$^{\uparrow4.50}$ | 53.74$^{\uparrow1.11}$ | 42.95$^{\uparrow0.22}$ |
| Rho-1 | 12.5% | 53.97$^{\downarrow5.28}$ | 40.34$^{\downarrow0.68}$ | 6.67$^{\downarrow1.44}$ | 13.27$^{\uparrow0.52}$ | 46.62$^{\downarrow2.13}$ | 32.17$^{\downarrow1.81}$ | 71.60$^{\uparrow1.05}$ | 47.46$^{\downarrow0.68}$ | 20.62$^{\downarrow2.12}$ | 23.74$^{\uparrow4.17}$ | 53.71$^{\uparrow1.08}$ | 43.43$^{\uparrow0.70}$ |
| | 25% | 59.08$^{\downarrow0.17}$ | 41.02$^{\uparrow0.00}$ | 6.67$^{\downarrow1.44}$ | 13.99$^{\uparrow1.24}$ | 47.19$^{\downarrow1.56}$ | 33.59$^{\downarrow0.39}$ | 70.37$^{\uparrow0.18}$ | 46.78$^{\downarrow1.36}$ | 21.76$^{\uparrow0.98}$ | 20.85$^{\uparrow1.28}$ | 52.22$^{\downarrow0.41}$ | 42.40$^{\downarrow0.33}$ |
| | 100% | 60.67$^{\uparrow1.42}$ | 41.36$^{\uparrow0.34}$ | 7.28$^{\downarrow0.83}$ | 14.97$^{\uparrow2.22}$ | 49.82$^{\uparrow1.07}$ | 34.82$^{\uparrow0.84}$ | 63.67$^{\downarrow6.88}$ | 43.39$^{\downarrow4.75}$ | 11.68$^{\downarrow11.06}$ | 15.46$^{\downarrow4.11}$ | 31.84$^{\downarrow20.79}$ | 33.21$^{\downarrow9.52}$ |
| TokenCleaning | 12.5% | 58.73$^{\downarrow0.52}$ | 43.73$^{\uparrow2.71}$ | 7.43$^{\downarrow0.68}$ | 12.84$^{\uparrow0.09}$ | 46.37$^{\downarrow2.38}$ | 33.82$^{\downarrow0.16}$ | 71.60$^{\uparrow1.05}$ | 49.49$^{\uparrow1.35}$ | 21.00$^{\downarrow1.74}$ | 22.73$^{\uparrow3.16}$ | 53.03$^{\uparrow0.40}$ | 43.57$^{\uparrow0.84}$ |
| | 25% | 59.96$^{\uparrow0.71}$ | 42.37$^{\uparrow1.35}$ | 5.99$^{\downarrow2.12}$ | 13.62$^{\uparrow0.87}$ | 46.14$^{\downarrow2.61}$ | 33.62$^{\downarrow0.36}$ | 70.19$^{\downarrow0.36}$ | 47.80$^{\downarrow0.34}$ | 19.94$^{\downarrow2.80}$ | 24.17$^{\uparrow4.60}$ | 52.02$^{\downarrow0.61}$ | 42.82$^{\uparrow0.09}$ |
| | 100% | 61.20$^{\uparrow1.95}$ | 41.02$^{\uparrow0.00}$ | 7.73$^{\downarrow0.38}$ | 15.41$^{\uparrow2.66}$ | 48.29$^{\downarrow0.46}$ | 34.73$^{\uparrow0.75}$ | 74.07$^{\uparrow3.52}$ | 46.44$^{\uparrow1.70}$ | 24.79$^{\uparrow2.05}$ | 33.26$^{\uparrow13.69}$ | 54.32$^{\uparrow1.69}$ | 46.58$^{\uparrow3.85}$ |
| **Q-Tuning (Ours)** | 12.5% | **64.20**$^{\uparrow4.95}$ | 42.03$^{\uparrow1.01}$ | **10.54**$^{\uparrow2.43}$ | **18.79**$^{\uparrow6.04}$ | **53.12**$^{\uparrow4.37}$ | **37.74**$^{\uparrow3.76}$ | 71.60$^{\uparrow1.05}$ | 48.14$^{\uparrow0.00}$ | **29.34**$^{\uparrow6.60}$ | **27.75**$^{\uparrow8.18}$ | **57.78**$^{\uparrow5.15}$ | **46.92**$^{\uparrow4.19}$ |
| | 25% | **63.14**$^{\uparrow3.89}$ | 42.03$^{\uparrow1.01}$ | **8.87**$^{\uparrow0.76}$ | **16.76**$^{\uparrow4.01}$ | **51.52**$^{\uparrow2.77}$ | **36.47**$^{\uparrow2.49}$ | 71.78$^{\uparrow1.23}$ | 47.12$^{\uparrow1.02}$ | **26.08**$^{\uparrow3.34}$ | **32.79**$^{\uparrow13.22}$ | **56.17**$^{\uparrow3.54}$ | **46.79**$^{\uparrow4.06}$ |
| Full Dataset | | 61.55 | 42.37 | 8.64 | 13.80 | 50.45 | 35.36 | 71.25 | 45.76 | 26.68 | 31.81 | 53.67 | 45.84 |

Tables 9 and 10 extend our evaluation to a broader set of pruning strategies under matched sample–token budgets. Table 9 varies sample ratios (12.5%, 25%) and token ratios (50%, 100%) under a fixed random token pruner, while Table 10 fixes the token ratio at 50% and varies the sample ratio (12.5%, 25%, 100%) under a random sample pruner.

**Advantages over additional sample-selection baselines.** Even against these competitive sample-selection approaches in Table 9, Q-Tuning consistently delivers the largest gains over the Random pruner under the same budgets. With 12.5% samples and 50% tokens, Q-Tuning improves LLaMA2-7B from 33.98 to 37.74 and Mistral-7B from 42.73 to 46.92, with gains observed on all five benchmarks rather than trading off one task for another. At the 25% sample ratio, Q-Tuning remains the only method that improves upon Random on every benchmark and achieves the strongest average performance among all evaluated baselines. These results show that, even relative to strong LLM-based ranking and gradient-based selection schemes, Q-Tuning extracts more value from the same subset of training data.

**Outperforming full data performance on a low budget.** Under both 12.5% and 25% sample ratios, Q-Tuning not only surpasses the additional baselines but also closely approaches or exceeds the full-dataset SFT performance. On LLaMA2-7B, Q-Tuning attains averages of 37.74 and 36.47, surpassing the full-data score of 35.36. On Mistral-7B, it reaches 46.92 and 46.79, again outperforming the full-data baseline of 45.84. This demonstrates that, even in the presence of strong sample-selection competitors, Q-Tuning can match or exceed full-data training while using fewer samples and tokens.

**Benefits over token-level baselines at fixed token ratio.** Table 10 further contrasts Q-Tuning with token-level pruning and cleaning approaches under a fixed 50% token ratio. While Rho-1 and TokenCleaning can yield moderate gains over Random in some configurations, they exhibit notable instability across models and tasks: for instance, Rho-1 severely degrades Mistral-7B performance at 100% samples, and TokenCleaning often needs the full sample budget to remain competitive. In contrast, Q-Tuning consistently achieves the highest average performance at both 12.5% and 25% sample ratios for LLaMA2-7B and Mistral-7B, with particularly large improvements on reasoning-heavy benchmarks such as GSM8K and SQuAD. Notably, for Mistral-7B, Q-Tuning with only 12.5% samples already matches or surpasses TokenCleaning's best results obtained with the full sample set, highlighting the robustness and data efficiency of our method under token-budget constraints.

### G.4 RESULTS ON LESS SAMPLES

Table 11: Q-Tuning under extreme data-retention settings on the Wizard dataset using Mistral-7B. We evaluated sample ratios of 6.25% and 12.5% combined with token ratios of 50% and 70%, retaining only 3–9% of the original data. Performance is reported on ARC-E, ARC-C, GSM8K, SQuAD, TriviaQA, and their average, and compared against both zero-shot and full-dataset baselines.

| Sample Ratio | Token Ratio | Data Retention | Mistral-7B | | | | | |
|---|---|---|---|---|---|---|---|---|
| | | | ARC-E | ARC-C | GSM8K | SQuAD | TriviaQA | Avg. |
| Zero-Shot | | - | 66.67 | 46.10 | 18.35 | 10.01 | 43.77 | 36.98 |
| 6.25% | 50% | 3.125% | 70.72 | 46.10 | **40.41** | 17.25 | 56.68 | 46.23 |
| 6.25% | 70% | 4.375% | 71.60 | **48.47** | 39.73 | 16.13 | 57.16 | 46.62 |
| 12.50% | 50% | 6.250% | 71.60 | 48.14 | 29.34 | 27.75 | **57.78** | **46.92** |
| 12.50% | 70% | 8.750% | **71.78** | 47.12 | 26.08 | **32.79** | 56.17 | 46.79 |
| Full Dataset | | 100% | 71.25 | 45.76 | 26.68 | 31.81 | 53.67 | 45.84 |

**Data efficiency under extreme retention.** Table 11 shows that Q-Tuning remains highly data-efficient even when retaining only 3–9% of the original training tokens on Mistral-7B. Across all four extreme settings, the average score ranges from 46.23 to 46.92, consistently outperforming both the zero-shot baseline (36.98) and the full-dataset SFT baseline (45.84), despite using at most 8.75% of the data. Notably, with just 6.25% samples and 50% tokens (*i.e.*, 3.125% data retention), Q-Tuning already achieves an average of 46.23, exceeding full-data fine-tuning. Increasing the token ratio to 70% at the same sample budget further improves the average to 46.62, while still operating in a regime where less than 5% of the original data is used.

**Strong performance with less than 10% data.** When the sample ratio is increased to 12.5% (still retaining only 6.25–8.75% of the data), Q-Tuning performance further improves. At 12.5% samples and 50% tokens, it attains the highest average score of 46.92, and at 12.5% samples and 70% tokens it achieves a comparable 46.79. These settings consistently outperform the full-dataset baseline not only on the average metric, but also on several individual benchmarks: for example, GSM8K improves from 26.68 (full data) to 40.41 in the most extreme 3.125% retention setting, SQuAD is boosted up to 32.79 (vs. 31.81 full data), and TriviaQA increases from 53.67 to 57.78.

## G.5 RESULTS ON MODELS OF DIFFERENT SCALES

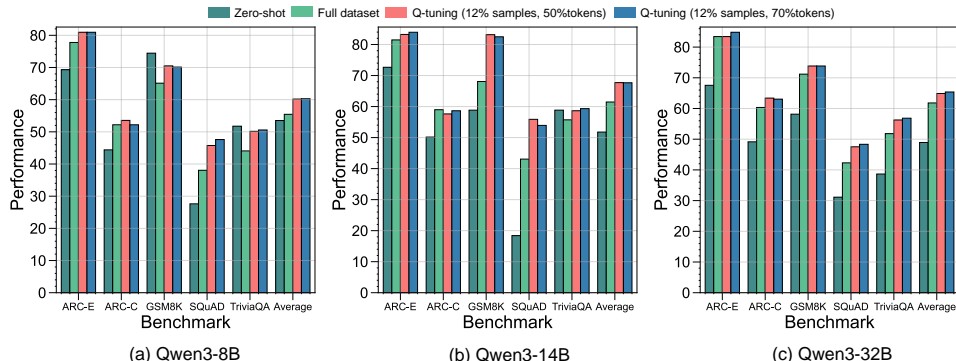

Figure 7: Comparison of Qwen models of varying scales (8B, 14B, 32B) across multiple benchmarks (ARC-E, ARC-C, GSM8K, SQuAD, TriviaQA, and their average). We report performance of Zero-shot, full dataset, and Q-Tuning with 12.5% samples under two token ratios (50% and 70%).

**Scalability across model sizes.** Figure 7 examines Q-Tuning on Qwen models of three different scales (8B, 14B, 32B), revealing that its benefits persist and even strengthen as model capacity increases. Across all benchmarks, the two Q-Tuning settings (12.5% samples with 50% or 70% tokens) consistently match or surpass the full-dataset SFT baseline, despite using only a fraction of the training data.

**Improved reasoning and QA performance at larger scales.** The largest performance margins emerge on reasoning-intensive benchmarks such as GSM8K and ARC-C. For the 32B model, Q-Tuning produces substantial boosts over both zero-shot and full-data fine-tuning, especially under the 70% token budget. For example, Q-Tuning improves GSM8K, SQuAD, and TriviaQA simultaneously, yielding the highest average score across all configurations.

## H ABLATION ACROSS DIFFERENT QUADRANTS

Table 12: Ablation on four quadrants on different datasets and models under sample ratios (12.5%, 25%) and token ratio (50%). We report performance under four quadrants (Q1–Q4) for both sample pruning and token pruning, where ✓ indicates that the corresponding quadrant is pruned. Experiments were conducted on Wizard using LLaMA2-7B. Metrics include ARC-E, ARC-C, GSM8K, SQuAD, TriviaQA, and their average.

| Sample pruning | | | | Token pruning | | | | LLaMA2-7B | | | | | |
|---|---|---|---|---|---|---|---|---|---|---|---|---|---|
| Q1 | Q2 | Q3 | Q4 | Q1 | Q2 | Q3 | Q4 | ARC-E | ARC-C | GSM8K | SQuAD | TriviaQA | Avg. |
| | | | | | | | | 12.5% *Samples,* 50% *Tokens* | | | | | |
| | ✓ | | | | ✓ | | | 60.14 | 41.02 | 8.04 | 13.11 | 50.23 | 34.51 |
| ✓ | ✓ | | | | ✓ | | ✓ | 64.02 | 41.02 | 9.86 | **19.26** | 52.00 | 37.23 |
| ✓ | ✓ | | | | ✓ | | | **64.20** | **42.03** | **10.54** | 18.79 | **53.12** | **37.74** |
| | | | | | | | | 25% *Samples,* 50% *Tokens* | | | | | |
| | ✓ | | | | ✓ | | | 59.61 | 41.02 | 7.51 | 13.74 | 50.25 | 34.43 |
| ✓ | ✓ | | | | ✓ | | ✓ | 61.02 | 40.68 | **9.10** | **18.66** | **51.68** | 36.23 |
| ✓ | ✓ | | | | ✓ | | | **63.14** | **42.03** | 8.87 | 16.76 | 51.52 | **36.47** |

Table 13: Ablation on four quadrants on different datasets and models under sample ratios (12.5%, 25%) and token ratio (50%). We report performance under four quadrants (Q1–Q4) for both sample pruning and token pruning, where ✓ indicates that the corresponding quadrant is pruned. Experiments were conducted on Wizard using Mistral-7B. Metrics include ARC-E, ARC-C, GSM8K, SQuAD, TriviaQA, and their average.

| Sample pruning | | | | Token pruning | | | | Mistral-7B | | | | | |
|---|---|---|---|---|---|---|---|---|---|---|---|---|---|
| Q1 | Q2 | Q3 | Q4 | Q1 | Q2 | Q3 | Q4 | ARC-E | ARC-C | GSM8K | SQuAD | TriviaQA | Avg. |
| | | | | | | | | 12.5% *Samples,* 50% *Tokens* | | | | | |
| | ✓ | | | | ✓ | | | 71.43 | 47.80 | 28.13 | **31.09** | 54.27 | 46.54 |
| ✓ | ✓ | | | | ✓ | | ✓ | 71.60 | **48.47** | 28.89 | 26.46 | **57.88** | 46.66 |
| ✓ | ✓ | | | | ✓ | | | **71.60** | 48.14 | **29.34** | 27.75 | 57.78 | **46.92** |
| | | | | | | | | 25% *Samples,* 50% *Tokens* | | | | | |
| | ✓ | | | | ✓ | | | 71.25 | 46.10 | **29.04** | 31.55 | 54.84 | 46.56 |
| ✓ | ✓ | | | | ✓ | | ✓ | 70.72 | 47.12 | 28.58 | 31.85 | **56.33** | **46.92** |
| ✓ | ✓ | | | | ✓ | | | **71.78** | 47.12 | 26.08 | **32.79** | 56.17 | 46.79 |

Table 14: Ablation on four quadrants on different datasets and models under sample ratios (12.5%, 25%) and token ratio (50%). We report performance under four quadrants (Q1–Q4) for both sample pruning and token pruning, where ✓ indicates that the corresponding quadrant is pruned. Experiments were conducted on OpenHermes using Qwen3-8B. Metrics include ARC-E, ARC-C, GSM8K, SQuAD, TriviaQA, and their average.

| Sample pruning | | | | Token pruning | | | | Qwen3-8B | | | | | |
|---|---|---|---|---|---|---|---|---|---|---|---|---|---|
| Q1 | Q2 | Q3 | Q4 | Q1 | Q2 | Q3 | Q4 | ARC-E | ARC-C | GSM8K | SQuAD | TriviaQA | Avg. |
| | | | | | | | | 12.5% *Samples,* 50% *Tokens* | | | | | |
| | ✓ | | | | ✓ | | | 79.01 | 52.20 | **74.68** | 36.35 | 48.20 | 58.09 |
| ✓ | ✓ | | | | ✓ | | ✓ | 79.37 | 52.20 | 71.80 | 35.55 | 47.23 | 57.23 |
| ✓ | ✓ | | | | | | | 78.84 | **53.90** | 68.99 | 44.32 | 49.15 | 59.04 |
| ✓ | ✓ | | | | ✓ | | | **80.95** | 53.56 | 70.51 | **45.76** | **50.16** | **60.19** |
| | | | | | | | | 25% *Samples,* 50% *Tokens* | | | | | |
| | ✓ | | | | ✓ | | | 75.31 | 50.17 | 68.08 | 29.47 | 45.90 | 53.79 |
| ✓ | ✓ | | | | ✓ | | ✓ | 76.37 | 50.17 | 65.28 | 31.63 | 45.24 | 53.74 |
| ✓ | ✓ | | | | ✓ | | | **79.89** | **52.88** | **69.37** | **43.54** | **48.33** | **58.80** |

Table 12, Table 13 and Table 14 investigate how pruning different quadrants (Q1–Q4) at both the sample and token levels affects downstream performance across multiple datasets (Wizard, OpenHermes) and models (LLaMA2-7B, Mistral-7B, Qwen3-8B). Across all settings, three consistent trends emerge, shedding light on the functional roles of the four quadrants.

**(1) Retaining Q1 is harmful and should not be combined with Q2 or Q4.** In every model and dataset, the configurations that *retain Q1* (*i.e.*, the first row of each block where Q1 has no ✓ under sample pruning) produce the lowest average performance. This degradation is particularly visible for LLaMA2-7B and Qwen3-8B: keeping Q1 while pruning Q3 and applying token pruning to Q2 leads to significant drops in GSM8K and SQuAD, and consistently lower averages. Since Q1 corresponds to high-entropy, high-PPL regions, which represents noisy or unstable training signals, these results indicate that Q1 is intrinsically harmful, and pruning it is essential.

**(2) Q4 contains confident tokens that should not be pruned.** Q4 consists of low-PPL yet high-entropy tokens: these are diverse continuations that the model still predicts with high confidence. Such regions often encode useful semantic variation. The ablations confirm that pruning Q4 at the token level (second row of each block, where Q4 has a ✓) consistently leads to worse performance. For LLaMA2-7B at both 12.5% and 25% sample ratios, Q4 pruning reduces ARC-E, weakens ARC-C, and lowers the overall average. For Mistral-7B, removing Q4 lowers SQuAD or destabilizes performance. The trend is strongest on OpenHermes with Qwen3-8B, where Q4-pruned configurations degrade GSM8K and the final average. These results confirm that Q4 provides valuable and diverse training signal and should be preserved during token pruning, as done in Q-Tuning.

**(3) High-PPL Q2 tokens are harmful and should be removed.** The final rows (Q-tuning) consistently perform best because they prune Q1 and Q3 at the sample level while also pruning Q2 at the token level. Q2 contains high-PPL tokens that contribute disproportionately to loss spikes, and pruning them yields improved stability and reasoning accuracy. The clearest evidence comes from Qwen3-8B under 12.5% samples and 50% tokens, where we include an additional configuration that prunes no quadrants at the token level (third row). This setting, which retains Q2's high-PPL tokens, performs notably worse on GSM8K, SQuAD, and the final average compared to the Q2-pruned configuration. Across most experiments, the best-performing rows are always those that remove Q2 tokens, confirming that high-PPL tokens are indeed harmful and that pruning them is beneficial.

# I    FURTHER CASE STUDY

## I.1    SEMANTIC ANALYSIS OF PRUNED TOKENS

To better understand the semantic nature of the pruned content and verify that we primarily remove "noise", we conduct a microscopic analysis on the Q2 quadrant. Concretely, we randomly sample 300 instances from Q2 and use GPT-5 to annotate the functional roles of over 44K tokens. We then compare the semantic distribution between the tokens that are *pruned* and those that are *kept* by our method.

Table 15: Semantic distribution of pruned vs. kept tokens within the Q2 quadrant. "Δ" denotes the difference in proportion between the kept and pruned sets.

| Semantic Category | Total Count | % Pruned | % Kept | Δ (Kept – Pruned) |
|---|---|---|---|---|
| Boilerplate / Templates | 323 | 0.8% | 0.3% | **-0.5%** (Preferentially Pruned) |
| Functional Words | 13,832 | 31.3% | 32.1% | +0.8% |
| Content (Non-Answer) | 18,582 | 42.7% | 38.6% | -4.1% |
| Gold Answer Spans | 1,245 | 2.7% | **3.5%** | **+0.8%** (Preferentially Kept) |
| Numbers / Symbols | 8,834 | 19.6% | 23.1% | +3.5% |
| Markdown / Formatting | 1,283 | 3.0% | 2.4% | -0.6% |
| **Total** | **44,099** | **100.0%** | **100.0%** | |

The results in Table 15 highlight three behaviors of our pruning mechanism:

- **Targeting redundancy.** Boilerplate and template tokens are more likely to be pruned (0.8% vs. 0.3%), indicating that the method effectively removes non-informative "fluff".

- **Preserving ground truth.** Gold answer spans are preferentially kept (3.5% vs. 2.7%), suggesting that the neighbor-aware smoothing helps separate "hallucinated" high-PPL tokens from factual high-PPL entities.

- **Nuanced filtering.** The shift in the proportion of numbers and symbols (19.6% pruned vs. 23.1% kept) shows that the method can distinguish useful operands and key quantities from spurious or intermediate calculations.

## I.2    PRUNED CASE STUDY

We present representative token-pruning cases from MATHINSTRUCT using the MISTRAL-7B model. For each example, we show the model's original `output` and highlight in red the fine-grained segments that are *absent* in the corresponding `pruned_output`. This visualization makes the pruning behavior transparent at the character/substring level and helps diagnose which semantic components are preferentially removed (e.g., boilerplate, redundant explanations, or unstable numeric derivations).

Unless otherwise stated, we keep the original formatting and line breaks of the model outputs to preserve exact alignment. The highlighting is produced by an LCS-based diff between `output` and `pruned_output`, ensuring that only truly deleted spans are marked while retained text remains uncolored. We do *not* introduce any additional edits beyond necessary LaTeX escaping.

These qualitative cases complement the aggregate statistics reported in the main text by revealing typical pruning patterns on mathematical reasoning tasks (definitions and restatements vs. computational steps), and by surfacing edge cases where pruning removes numerically relevant content. Together, they provide an interpretable view of how token-level pruning affects the fidelity and brevity of generated solutions.

### SAMPLE 0

**Instruction** sandy is younger than molly by 20 years . if the ratio of their ages is 7 : 9 , how old is sandy ? Let's write a program.

**Output**

```
n0 = 20.0
n1 = 7.0
n2 = 9.0
t0 = n2 - n1|
t1 = n1 / t0
answer =n0 * t1
print(answer)
```

### SAMPLE 1

**Instruction** A manufacturing process produces 5000 units of a particular product. The probability of a certain type of defect occurring in a unit is 0.02. What is the probability that exactly 100 units will have this defect? **Output (deletions highlighted)**

```
To solve this problem, we can use the binomial probability
formula, which is:
P(X = k) = (nCk) * (p^k) * (1-p)^(n-k)
where:
- P(X = k) is the probability of exactly k successes (defective
units in this case) in n trials (total units produced)
- nCk (read as "n choose k") is the number of combinations of
choosing k successes from n trials, calculated as n! / (k! *
(n-k)!)
- p is the probability of success (defect occurring) in a single
trial
- 1-p is the probability of failure (no defect occurring) in a
single trial
- n is the total number of trials (units produced)
- k is the number of successes (defective units) we want to find
the probability for
In this problem, we have:
- n =5000 (total units produced)
- k = 100 (defective units)
- p = 0.02 (probability of a defect occurring in a unit)
Now, we can plug these values into the binomial probability
formula:
P(X = 100) = (5

First, let's calculate 5000C100:
5000C100 = 5000! / (100! * 4900!)
= 2.031759e+134 / (9.332622e+157 * 1.929713e+126)
= 2.031759e+134 / (1.801713e+284)
= 1.127271e-150
Now, let's calculate 0.02^100 and 0.98^4900:
0.02^100 = 1.267651e-70
0.98^4900 = 2.372472e-44
Finally, let's plug these values back into the formula:
The probability that exactly 100 units will have this defect is
approximately 3.394e-264, which is an extremely small probability.
```

## J EFFICIENCY ANALYSIS: THEORETICAL FLOPS AND EMPIRICAL LATENCY

### J.1 THEORETICAL ANALYSIS OF EFFICIENCY

To assess the computational efficiency of Q-Tuning, we report both the theoretical floating-point operation counts (FLOPs) and the empirical end-to-end training latency.

**Theoretical FLOPs.** Let $\mathcal{C}_{\text{fwd}}$ denote the FLOPs required for one forward pass over the full training data, and $\mathcal{C}_{\text{bwd}}$ the FLOPs for the corresponding backward pass. For Transformer-based LLMs it is standard that $\mathcal{C}_{\text{bwd}} \approx 2\mathcal{C}_{\text{fwd}}$.

**Full-data SFT.** The model performs both a forward and backward pass on all samples and tokens:

$$\text{Cost}_{\text{Full}} = \mathcal{C}_{\text{fwd}} + \mathcal{C}_{\text{bwd}} \approx 3\mathcal{C}_{\text{fwd}}. \tag{5}$$

**Q-Tuning.** Our method consists of two stages.

*Selection phase.* We run a gradient-free forward pass on the full data to compute PPL and entropy for sample- and token-level scoring, with cost $\mathcal{C}_{\text{sel}} = 1 \cdot \mathcal{C}_{\text{fwd}}$.

*Training phase.* We then perform SFT only on the pruned subset. Let $r \in (0, 1]$ be the effective data retention ratio (combining sample- and token-level pruning). The cost of this phase is

$$\mathcal{C}_{\text{train}} = r\big(\mathcal{C}_{\text{fwd}} + \mathcal{C}_{\text{bwd}}\big) \approx 3r\mathcal{C}_{\text{fwd}}. \tag{6}$$

The total theoretical cost of Q-Tuning is therefore

$$\text{Cost}_{\text{Q-Tuning}} \approx (1 + 3r)\mathcal{C}_{\text{fwd}}. \tag{7}$$

Q-Tuning is more FLOP-efficient than full-data SFT whenever $1 + 3r < 3$, *i.e.*,

$$r < \tfrac{2}{3}. \tag{8}$$

In our typical configurations (e.g., keeping 12.5% of samples and 50% of tokens), the effective retention ratio $r$ is far below this threshold, so Q-Tuning yields a substantial theoretical FLOPs reduction.

## J.2 EMPIRICAL WALL-CLOCK LATENCY AND MEMORY FOOTPRINT.

We further quantify the practical cost–benefit trade-off by measuring end-to-end training efficiency under our standard three-epoch setup on LLaMA2-7B with $8\times$ A100 GPUs. While full-data SFT requires about 90 minutes, Q-Tuning (including the selection phase) completes training in approximately 65 minutes, corresponding to a $\frac{90-65}{90} \approx 28\%$ reduction in wall-clock time. This improvement arises because the cost of per-sample screening is more than offset by the savings from pruning: screening requires only an inference-mode forward pass to compute selection metrics, whereas the backward pass and optimizer updates dominate the total FLOPs. Since these expensive steps are skipped for most of the data that Q-Tuning prunes, the pruning gains amortize the screening overhead.

Table 16 compares relative FLOPs, wall-clock time, and peak memory usage between full-data SFT and Q-Tuning under several representative pruning configurations. Table 17 further decomposes the screening overhead, showing that selection time remains a modest fraction of the overall training budget.

Table 16: Efficiency comparison of full-data SFT and Q-Tuning on LLaMA2-7B. "Rel. FLOPs" reports theoretical training FLOPs normalized by full-data SFT.

| Sample ratio | Token ratio | Rel. FLOPs (vs full) | Time (min) | Peak mem (%) |
|---|---|---|---|---|
| 12.5% | 50% | 0.0625 | 65 | 27% |
| 25% | 50% | 0.1250 | 70 | 35% |
| 50% | 50% | 0.2500 | 80 | 50% |
| 100% (Full) | 100% | 1.0000 | 90 | 78% |

Table 17: Breakdown of screening overhead and total training time on LLaMA2-7B under different pruning settings.

| Sample ratio | Token ratio | Sample selection (min) | Token selection (min) | Total time (min) | Peak mem (%) |
|---|---|---|---|---|---|
| 12.5% | 50% | 35.88 | 6.96 | 65 | 27% |
| 25% | 50% | 38.64 | 7.49 | 70 | 35% |
| 50% | 50% | 44.16 | 8.56 | 80 | 50% |
| 100% | 100% | 0.00 | 0.00 | 90 | 78% |

These results confirm that Q-Tuning reduces both computation and memory consumption roughly in proportion to the effective data retained. Importantly, this efficiency gain is achieved without any degradation in model performance and, in many cases, even leads to improved performance.

