# OpenReview forum: "Winning the Pruning Gamble: A Unified Approach to Joint Sample and Token Pruning for Efficient Supervised Fine-Tuning"
_ICLR.cc/2026/Conference — ICLR 2026 Conference Desk Rejected Submission_

### Official Review · Reviewer_MSz7 · 2025-10-27

**Soundness:** 3
**Presentation:** 3
**Contribution:** 3
**Rating:** 6
**Confidence:** 4

**Summary:**

This paper proposes the Q-Tuning algorithm, which incorporates perplexity and predictive entropy to measure the error and uncertainty, and then make the sample-level and token-level pruning. The experiments across various settings validate the effectiveness of the algorithm.

**Strengths:**

1. The presentation is quite good. The illustration is quite helpful and the intuition behind algorithm is easy to understand.
2. The experimental result is comprehensive.

**Weaknesses:**

1. The algorithms depends the definition of $\alpha$ and $\beta$. The calculation that $0.5*(\alpha_{low} + \alpha_{high}) $ is quite heuristic. The author is encouraged to try other potential calculations. I wonder for different datasets, should we use different calculation methods since the error and uncertainty threshold might be quite different for math and creative writing.

2. The author is encouraged to include more case study to understand how sample and token pruning works in practice, better across different domains. I am curious about which token will be dropped by token pruning.

3. If possible, the author can provide some theoretical analysis for current algorithm. The current algorithm is quite experimental and highly require the setup of the hyper-parameters.

**Questions:**

Please check the weakness part.

---

> ### Author Response · Authors · 2025-11-21
> **(Part 1) Response to Reviewer MSz7**
>
> ## Q1: About thresholds in bisect search.
>
> Thanks a lot.
>
> First, we would like to clarify that the expression $0.5 \times (\alpha_{\text{low}} + \alpha_{\text{high}})$ in the manuscript is **not** a fixed heuristic coefficient for data selection. It is simply the midpoint update step inside our Bisection Search procedure (Algorithm 1, Lines 6–16). The goal of this procedure is to *solve* for the quantile thresholds ($\alpha^*$, $\beta^*$) such that the proportion of retained samples exactly matches the target budget, rather than to impose any hand-tuned constant. In other words, the thresholds are determined adaptively by the data distribution at each iteration, instead of being pre-set as global hyperparameters.
>
> This design directly addresses your concern about domain differences and is, in fact, one of the core advantages of our framework: the thresholds are decided on-the-fly by the empirical distribution of the current batch.
>
> **Distribution-agnostic quantile thresholds and adaptability.** We fully agree that absolute values of perplexity and entropy can differ substantially across domains (e.g., mathematical reasoning vs. creative writing). However, Q-Tuning never relies on fixed absolute cutoffs. The Bisection Search in Algorithm 1 iteratively locates quantile thresholds so that we always operate on *relative* ranks within the current batch. As a result, the method is inherently distribution-agnostic: on “easy” datasets (globally low perplexity) and “hard” datasets (globally high perplexity) alike, it converges to thresholds that select the relatively most informative samples (Q2 and Q4) under the same budget. This eliminates the need for manual recalibration when moving across domains and allows the same procedure to adaptively isolate high-value regions of the error–uncertainty plane.
>
> **Algorithmic robustness and remedies for edge cases.** We further note that this adaptive partitioning is empirically stable. As shown in Section 4.4 and Appendix C.2 (Figure 3a), Q-Tuning maintains consistent performance even when the batch size is reduced to 8. This robustness stems from two components in Algorithm 1 working in concert:
> (i) the adaptive thresholding via Bisection Search, which guarantees that, regardless of whether the current batch is globally easy or hard, the selected region respects the target budget and focuses on the highest-value quadrants; and
> (ii) the geometric Compensation Mechanism (Algorithm 1, Lines 18–21), which explicitly recalls samples based on their distance to the diagonal (y = x). In pathological or highly skewed distributions where naive thresholding could under-represent certain regions, this mechanism preferentially re-introduces samples with large error–uncertainty discrepancy, preventing distributional collapse and preserving the diversity of informative data.

---

> ### Author Response · Authors · 2025-11-21
> **(Part 2) Response to Reviewer MSz7**
>
> ## Q2: Further case studies.
>
> We sincerely appreciate the reviewer’s suggestion to provide concrete examples of the pruning process. To address this, we have significantly expanded Appendix I (Table 15) in the revised manuscript, incorporating both a large-scale quantitative semantic analysis and detailed qualitative visualizations.
>
> **1. Quantitative Semantic Analysis**
> To move beyond emprical evidence, we conducted a systematic analysis of the semantic profile of pruned tokens. We randomly sampled 300 instances from the Q2 quadrant (Valuable Misconceptions) and employed GPT-5 to annotate the functional roles of over 44,000 tokens. We then compared the distribution of these categories within the *Pruned Set* versus the *Kept Set*.
>
>
> | Semantic Category | Total Count | % of Pruned Set | % of Kept Set | $\Delta$ (Kept - Pruned) |
> | :--- | :--- | :--- | :--- | :--- |
> | **Boilerplate / Templates** | 323 | 0.8% | 0.3% | **-0.5%** (Preferentially Pruned) |
> | **Functional Words** | 13,832 | 31.3% | 32.1% | +0.8% |
> | **Content (Non-Answer)** | 18,582 | 42.7% | 38.6% | -4.1% |
> | **Gold Answer Spans** | 1,245 | 2.7% | **3.5%** | **+0.8%** (Preferentially Kept) |
> | **Numbers / Symbols** | 8,834 | 19.6% | 23.1% | +3.5% |
> | **Markdown / Formatting** | 1,283 | 3.0% | 2.4% | -0.6% |
> | **Total** | **44,099** | **100.0%** | **100.0%** | |
>
> The resulting distribution underscores the semantic selectivity of our pruning mechanism:
> *   **Targeting Redundancy:** We observe a distinct bias against "Boilerplate" and "Template" tokens, which are disproportionately represented in the pruned set (0.8%) compared to the retained set (0.3%). This confirms that high-perplexity signals in this quadrant often correspond to low-information redundancy (e.g., generic headers like *"Sure, here is the solution..."*) that dilutes the gradient signal.
> *   **Preserving Ground Truth:** Crucially, the mechanism demonstrates robustness by preserving "Gold Answer Spans," which appear at a higher density in the kept set (3.5%) than in the pruned set (2.7%). This finding implies that our neighbor-aware smoothing effectively distinguishes between "hallucinated" errors (which interrupt local coherence and are suppressed) and "factual" complexities (which maintain coherence and are protected).
> *   **Nuanced Filtering:** The distribution of "Numbers/Symbols" reflects a nuanced capability to retain necessary operands (23.1% kept) while filtering incorrect intermediate calculations (19.6% pruned).
>
> **2. Qualitative Visualization**
> To complement this quantitative analysis, **we provide concrete visualizations in Figure 8 of Appendix H**. These case studies highlight the "surgical" nature of Q-Tuning across different domains:
> *   In **Reasoning tasks (MathInstruct)**, the method selectively masks incorrect intermediate calculation steps while retaining the problem formulation and correct logical connectors.
> *   In **General Instruction tasks**, it effectively prunes repetitive loops and "hallucinated" facts that diverge from the instruction, while leaving the core narrative structure intact.

---

> ### Author Response · Authors · 2025-11-21
> **(Part 3) Response to Reviewer MSz7**
>
> ## Q3: About theoretical analysis
>
> We thank the reviewer for encouraging a deeper theoretical examination of our framework. We acknowledge that while closed-form convergence bounds for non-convex optimization in LLMs remain an open problem in the field, our algorithm is not merely heuristic. Instead, it is grounded in rigorous properties regarding gradient estimation.
>
> To address the theoretical properties of our partitioning strategy, we analyze the algorithm as a variance-reduction mechanism. We propose that Q-Tuning improves convergence by maximizing the **Gradient Signal-to-Noise Ratio (SNR)** of the mini-batch estimator.
>
>
> **Convergence Justification via Gradient Signal-to-Noise Ratio**
>
>
> **Definition 1 (Gradient SNR).** Let $g(x) = \nabla_\theta \mathcal{L}(x; \theta)$ be the stochastic gradient for a sample $x$. For a batch $B$, the SNR of the gradient estimator is defined as the ratio of the expected gradient norm to its variance: $\text{SNR}(B) = \| \mathbb{E}\_{x \sim B}[g(x)] \|^2 / \text{Var}\_{x \sim B}(g(x))$.
>
> **Proposition 1 (SNR Maximization).** *The Q-Tuning selection strategy theoretically yields a strictly higher Gradient SNR than random sampling by filtering data regimes dominated by aleatoric uncertainty and vanishing gradients.*
>
> **Analysis.** We analyze the statistical properties of the gradients contributed by each quadrant. Samples in Q1 (Harmful Noise) are characterized by high predictive entropy, implying a flat probability distribution. Consequently, the gradient vectors $g(x)$ for $x \in Q_1$ exhibit high variance (large denominator in the SNR term) and inconsistent directions, contributing noise rather than signal. Conversely, samples in Q3 (Redundant Knowledge) are already mastered, resulting in gradient norms $\|g(x)\|$ negligible. Including them in a batch dilutes the expected signal (numerator) without contributing to learning.
>
> Q-Tuning constructs a batch drawn strictly from $Q_2 \cup Q_4$. By removing $Q_1$, we significantly reduce the variance of the estimator. By removing $Q_3$, we prevent the dilution of the average gradient magnitude. Furthermore, we retain $Q2$ (Valuable Misconceptions), which are characterized by low entropy (high model confidence) but high error. These samples produce gradient vectors with large norms and consistent directions (systematic error correction), which constitutes a strong learning signal. Therefore, by maximizing the numerator (via Q2) and minimizing the denominator (via filtering Q1), Q-Tuning theoretically approximates an importance sampling estimator that optimizes the stability and efficiency of the descent trajectory.

---

### Official Review · Reviewer_2BBK · 2025-11-01

**Soundness:** 3
**Presentation:** 3
**Contribution:** 3
**Rating:** 6
**Confidence:** 4

**Summary:**

This paper tackles the critical challenge of data efficiency in supervised fine-tuning for large language models by introducing a unified approach to sample and token pruning. The authors present the Error-Uncertainty (EU) Plane, a diagnostic tool mapping training instances based on perplexity and entropy, which reveals data heterogeneity at both sample and token levels. Leveraging these insights, they propose Quadrant-based Tuning (Q-Tuning), a bi-level, context-aware pruning strategy that prunes at both sample and token levels depending on their EU quadrant classification. Evaluations across several models (LLaMA2-7B, LLaMA3-8B, Mistral-7B, SmolLM2-1.7B), benchmarks, and pruning budgets show Q-Tuning consistently outperforms both standard and recent pruning baselines, achieving results that sometimes surpass even full-data fine-tuning.

**Strengths:**

- **Clear Motivation & Unification**: The paper incisively diagnoses a gap in existing data pruning strategies, namely the isolated and uncoordinated treatment of sample-level and token-level pruning. The formalization of joint sample-token pruning (Generalized Dynamic Data Pruning) is thoughtfully constructed.

- **Novel Framework**: The EU Plane is an intuitive and effective way to classify and differentiate data utility across both error (perplexity) and uncertainty (entropy), leading directly into a well-motivated Quadrant-based strategy.
Methodological Rigor & Clarity: The algorithm is explicitly and thoroughly described (see Algorithm 1 and associated discussion), including how sample and token thresholds are efficiently estimated batchwise using bisection.

- **Strong Empirical Results**: Q-Tuning consistently matches or surpasses full-data SFT performance despite using only a small fraction of the training data (e.g., +38% improvement using just 12.5% of data on SmolLM2-1.7B). Tables 1 and 2 show Q-Tuning outperforms strong and varied baselines across diverse data-token keep ratios, models, and benchmarks.

- **Thorough Analysis and Fair Baselines**: The experiments include extensive comparisons with both older and state-of-the-art sample and token pruning methods (including InfoBatch, FastV, SparseVLM, Entropy, etc.), ablation studies of batch size and neighborhood smoothing parameter, and analysis of Q-Tuning's robustness.

**Weaknesses:**

- **Limited Theoretical Analysis Beyond Heuristic Justification**: While the bi-level optimization formalism is clearly presented, there is little in terms of theoretical guarantees about optimality, generalization (beyond empirical validation), or the properties of the partitioning/procedures under varying dataset/model regimes. The EU Plane partitioning, though motivated with intuition, is not probed rigorously for, e.g., stability, sensitivity, or edge-case behavior.

- **Token Pruning Specifics and Ambiguities**: The paper's description of token-level pruning is mostly empirical/heuristic, focused on high-PPL and neighborhood smoothing. There is lack of detail on scenarios where the context-aware scoring mechanism might fail or degenerate (e.g., when high PPL signals are not semantically aligned with true error). There should be more analysis on cases where token-level and sample-level criteria may conflict or interact in undesirable ways.

**Questions:**

- Sensitivity to Quadrant Boundaries and Dynamic Partitioning: How robust is Q-Tuning to instability in the EU Plane thresholds, especially early in training or on unbalanced datasets? Are there pathological cases where the quadrant assignment is unstable or assigns virtually all data to one region?

- Token Pruning Failure Modes: Can the authors provide adversarial or edge-case analysis of where neighborhood-smoothed token-level PPL might prune semantically valuable content, and how such cases would be detected or remedied in practice?

- Computational Overhead: While the paper claims negligible overhead for the per-batch EU Plane computation and threshold search, is there any analysis or reporting of actual wall-clock timing or additional GPU consumption, especially for models closer to production scale?

---

> ### Author Response · Authors · 2025-11-21
> **(Part 1) Response to Reviewer 2BBK**
>
> ## Q1: About further justification.
>
> We appreciate the reviewer’s assessment of our formal framework. We acknowledge that, consistent with the current state of research in dynamic data selection for Large Language Models, we rely primarily on empirical validation rather than closed-form generalization bounds, which remain an intractable open problem for non-convex deep learning objectives.
>
> **Justification of Q-tuning**. However, we wish to clarify that the design of Q-Tuning incorporates specific algorithmic mechanisms that theoretically guarantee stability and robustness across varying regimes, addressing the concerns regarding the properties of the partitioning procedure.
>
> First, regarding the stability of the EU Plane partitioning, we argue that our use of **Quantile-Based Bisection Search** (Algorithm 1) provides inherent algorithmic stability that standard absolute thresholding lacks. In dynamic training regimes, the distributions of Perplexity and Entropy shift as the model learns (distribution drift). A heuristic based on absolute values (e.g., pruning samples with loss $> \epsilon$) would suffer from instability, leading to empty or overflowing quadrants. By contrast, our bisect search mechanism is **scale-invariant**; it dynamically locates the decision boundaries relative to the current batch statistics. This guarantees that the budget constraints ($r_{sample}$) are satisfied at every step $t$, regardless of the model’s calibration state. In edge cases where the distribution collapses (e.g., uniform uncertainty across a batch), the bisection search naturally degrades to uniform sampling within the tie-break region, ensuring that the optimization loop never stalls or diverges.
>
> Second, regarding sensitivity and the justification for the partitioning logic, the EU Plane is grounded in the mathematical decomposition of the log-perplexity into intrinsic uncertainty (Entropy) and model-data divergence (KL-Divergence). The partitioning is not arbitrary; it effectively isolates samples based on the *source* of the error. Samples in Q2 are selected because they exhibit high divergence despite low entropy, representing systematic misconceptions rather than aleatoric noise. This structural logic reduces the sensitivity of the method to dataset variations. Empirically, we probed the sensitivity of these procedures in Section 4.4 and Appendix C.2. The results in Figures 3 and 5 demonstrate that performance remains stable across a wide range of neighbor-smoothing factors ($\lambda$) and batch sizes. The convexity and flatness of these performance curves serve as strong empirical evidence that the method is not brittle to hyperparameter selection, validating the robustness of the underlying bi-level optimization strategy.
>
> **Theoretical analysis**. To address the theoretical properties of our partitioning strategy, we analyze the algorithm as a variance-reduction mechanism. We propose that Q-Tuning improves convergence by maximizing the **Gradient Signal-to-Noise Ratio (SNR)** of the mini-batch estimator.
>
> **Definition 1 (Gradient SNR).** Let $g(x) = \nabla_\theta \mathcal{L}(x; \theta)$ be the stochastic gradient for a sample $x$. For a batch $B$, the SNR of the gradient estimator is defined as the ratio of the expected gradient norm to its variance: $\text{SNR}(B) = \\| \mathbb{E}\_{x \sim B}[g(x)] \\|^2 / \text{Var}\_{x \sim B}(g(x))$.
>
> **Proposition 1 (SNR Maximization).** *The Q-Tuning selection strategy theoretically yields a strictly higher Gradient SNR than random sampling by filtering data regimes dominated by aleatoric uncertainty and vanishing gradients.*
>
> **Analysis.** We analyze the statistical properties of the gradients contributed by each quadrant. Samples in Q1 (Harmful Noise) are characterized by high predictive entropy, implying a flat probability distribution. Consequently, the gradient vectors $g(x)$ for $x \in Q_1$ exhibit high variance (large denominator in the SNR term) and inconsistent directions, contributing noise rather than signal. Conversely, samples in Q3 (Redundant Knowledge) are already mastered, resulting in gradient norms $\|g(x)\|$ negligible. Including them in a batch dilutes the expected signal (numerator) without contributing to learning.
>
> Q-Tuning constructs a batch drawn strictly from $Q_2 \cup Q_4$. By removing $Q_1$, we significantly reduce the variance of the estimator. By removing $Q_3$, we prevent the dilution of the average gradient magnitude. Furthermore, we retain $Q2$ (Valuable Misconceptions), which are characterized by low entropy (high model confidence) but high error. These samples produce gradient vectors with large norms and consistent directions (systematic error correction), which constitutes a strong learning signal. Therefore, by maximizing the numerator (via Q2) and minimizing the denominator (via filtering Q1), Q-Tuning theoretically approximates an importance sampling estimator that optimizes the stability and efficiency of the descent trajectory.

---

> ### Author Response · Authors · 2025-11-21
> **(Part 2) Response to Reviewer 2BBK**
>
> ## Q2: Case Study on the Interaction Between Sample and Token Pruning
>
> We appreciate the reviewer’s insightful suggestion. Our initial draft explained the motivations behind Context-Aware Smoothing and the Asymmetric Policy, but did not provide a sufficiently concrete analysis of how these mechanisms behave in practice. In response, we provide ablations and a additional semantic-level case study to more clearly identify when neighbor-smoothed perplexity succeeds, where its limitations remain, and how sample-level and token-level pruning interact during training. The full results are included in theSection 4.4 (Table 3) and Appendix I (Table 15).
>
> ### Q2.1: Efficacy and Failure Modes of Neighbor-Smoothed PPL
>
> High perplexity can stem from either genuine model errors or information-rich but uncommon content. Our smoothing mechanism (Eq. 4 with λ = 0.5) is designed to distinguish between these cases by exploiting local coherence. Errorful segments such as hallucinations or broken reasoning tend to produce consistently elevated perplexity across multiple neighboring tokens, while rare but correct entities typically appear within a coherent local context, resulting in low neighbor perplexity. Averaging thus suppresses isolated spikes while amplifying contiguous error regions.
>
> The ablation study below confirms the benefit of smoothing. Removing it (λ = 0) lowers the average performance and produces a nearly twenty percent degradation on GSM8K, while reversing the smoothing direction further hurts performance.
>
> **Ablation study on token pruner in Q-Tuning**
>
> | Method                     | Avg.  | ARC-E | ARC-C | GSM8K | SQuAD | TriviaQA |
> |----------------------------|-------|-------|-------|-------|--------|-----------|
> | Rho-1                      | 45.03 | 69.66 | 46.78 | 24.03 | 29.23  | 55.43     |
> | PPL (λ=0)                  | 45.92 | 71.60 | 46.44 | 25.32 | 29.71  | **56.54** |
> | Reversed PPL (λ=0.5)       | 44.86 | **73.02** | 47.12 | 16.68 | 32.01  | 55.47     |
> | **PPL (λ=0.5)**            | **46.79** | 71.78 | **47.12** | **26.08** | **32.79** | 56.17     |
>
> To better understand the semantic effect of the token pruner, we annotated more than 44K tokens from Q2 samples using GPT-5. The distribution below shows that the pruned tokens are dominated by boilerplate and redundant content, while gold answer spans are retained at noticeably higher rates. This validates that smoothing effectively protects core ground truth while filtering out non-informative segments.
>
> **Semantic distribution of pruned vs. kept tokens in Q2**
>
> | Semantic Category | Total Count | % of Pruned Set | % of Kept Set | Δ (Kept − Pruned) |
> | :--- | ---: | ---: | ---: | ---: |
> | Boilerplate / Templates | 323 | 0.8% | 0.3% | −0.5% |
> | Functional Words | 13,832 | 31.3% | 32.1% | +0.8% |
> | Content (Non-Answer) | 18,582 | 42.7% | 38.6% | −4.1% |
> | Gold Answer Spans | 1,245 | 2.7% | 3.5% | +0.8% |
> | Numbers / Symbols | 8,834 | 19.6% | 23.1% | +3.5% |
> | Markdown / Formatting | 1,283 | 3.0% | 2.4% | −0.6% |
> | Total | 44,099 | 100.0% | 100.0% | |
>
> We also identify clear failure modes. Rare entities or long digit sequences may still exhibit elevated neighbor perplexity, causing occasional over-pruning. The revised manuscript discusses lightweight safeguards, such as protected spans or per-segment pruning caps, to further reduce such edge cases.
>
> ### Q2.2: Interaction and Synergy Between Sample and Token Pruning.
>
> The potential for interference between sample-level and token-level objectives was a primary consideration in the design of Q-Tuning. We recognize that naively applying both strategies independently can lead to destructive interference, where valuable contexts selected by one are degraded by the other. Our framework resolves this through a structural **Asymmetric Pruning Policy**, which ensures that the two levels operate in coordination rather than conflict.
>
> **1. Preserving Semantic Integrity in Q4 (Calibration Data)**
> Samples in **Q4** (Low Error, High Uncertainty) often require full context for proper calibration. Token removal could break key dependencies, so Q4 always retains all tokens. This prevents sample-level retention from being weakened by token-level pruning.
>
> **2. Synergistic Refinement in Q2 (Valuable Misconceptions)**
> The interaction shifts to a synergistic mode within **Q2** (High Error, Low Uncertainty). These samples represent "confident errors," such as hallucinations or reasoning traps. Here, the sample-level selector first identifies the instance as a high-value target for error correction. Subsequently, the token-level pruner acts to refine this signal by selectively masking the specific tokens associated with the divergence, while retaining the instruction and correct context. Instead of conflicting, the two levels play distinct, complementary roles: the sample level **localizes the context** for correction, and the token level **purifies the gradient signal** by suppressing the specific causes of the error.

---

> ### Author Response · Authors · 2025-11-21
> **(Part 3) Response to Reviewer 2BBK**
>
> ## Q3: Sensitivity to Quadrant Boundaries and Dynamic Partitioning
>
> We thank the reviewer for raising this important point. Ensuring that quadrant assignments remain stable under varying training dynamics was a central consideration in the design of Q-Tuning. Our framework addresses this through two complementary safeguards that together prevent instability in both early-stage and skewed-distribution settings.
>
> The first safeguard is the **Adaptive Thresholding via Bisection Search**. Instead of relying on fixed perplexity or entropy cutoffs, the procedure operates entirely on relative quantiles computed from the current batch. This design ensures that the target retention rate is satisfied exactly at each iteration, regardless of transient fluctuations in model confidence or changes in the global data distribution. As a result, the algorithm cannot collapse into degenerate states where all samples fall into one quadrant or where high-value data are inadvertently discarded due to shifting absolute scales.
>
> To further protect against edge cases where samples cluster tightly along the diagonal (error approximately equals uncertainty), we introduce a geometric **Compensation Mechanism** (Algorithm 1, Lines 18–21), which explicitly re-balances the partitioning by recalling samples with the largest perpendicular deviation from the diagonal. This ensures that the retained data maintain high informational diversity even under severe distributional skew. This mechanism serves as a fail-safe that guarantees meaningful separation between the quadrants even when the underlying distribution is degenerate.
>
> **Empirically, these safeguards are validated through stress-test experiments in Section 4.4 (Figure 3) and Appendix C.2 (Figure 5).** Even when the batch size is reduced to as few as 8 samples, which is an extreme condition that makes quantile-based partitioning highly brittle, Q-Tuning maintains stable quadrant assignments and consistent downstream performance. These results support our claim that the dynamic partitioning procedure is robust across a wide range of real-world training regimes.

---

> ### Author Response · Authors · 2025-11-21
> **(Part 4) Response to Reviewer 2BBK**
>
> ## Q4: Case study on token-level pruning.
>
> Thank you for raising this important question. We agree that neighborhood-smoothed perplexity, while substantially more stable than raw PPL, may still fail under certain adversarial or edge-case conditions. To address this concern, we provide both a ablation study and a new semantic-level case study to identify where the token pruner succeeds and where it may potentially prune valuable content. We also include qualitative before-and-after pruning examples in Appendix I to illustrate these behaviors concretely.
>
> **1. Empirical evidence from token-level ablations**
>
> The ablation study (Table below) demonstrates that neighbor-smoothed PPL (λ = 0.5) is the only variant that consistently improves accuracy across all tasks. Raw PPL (λ = 0) is more brittle, while the reversed smoothing variant often amplifies noise and harms reasoning performance:
>
> **Ablation study on token pruner in Q-Tuning**
>
> | Method               | Avg.  | ARC-E | ARC-C | GSM8K | SQuAD | TriviaQA |
> |----------------------|-------|-------|-------|-------|--------|-----------|
> | Rho-1                | 45.03 | 69.66 | 46.78 | 24.03 | 29.23  | 55.43     |
> | PPL (λ=0)            | 45.92 | 71.60 | 46.44 | 25.32 | 29.71  | **56.54** |
> | Reversed PPL (λ=0.5) | 44.86 | **73.02** | 47.12 | 16.68 | 32.01  | 55.47     |
> | **PPL (λ=0.5)**      | **46.79** | 71.78 | **47.12** | **26.08** | **32.79** | 56.17     |
>
> These results indicate that the smoothing mechanism is essential for avoiding failure modes that arise when pruning is overly sensitive to isolated spikes in perplexity.
>
> **2. Semantic-level analysis: What is actually pruned, and what is preserved?**
>
> To identify edge-case behaviors, we conducted a new token-level semantic annotation study on 44K tokens from Q2 samples. The results show that the pruner is highly structured in what it removes:
>
> **Semantic distribution of pruned vs. kept tokens in Q2**
>
> | Semantic Category        | Total Count | % of Pruned Set | % of Kept Set | Δ (Kept − Pruned) |
> | :----------------------- | ----------: | --------------: | ------------: | -----------------: |
> | Boilerplate / Templates  | 323         | 0.8%            | 0.3%          | −0.5%              |
> | Functional Words         | 13,832      | 31.3%           | 32.1%         | +0.8%              |
> | Content (Non-Answer)     | 18,582      | 42.7%           | 38.6%         | −4.1%              |
> | Gold Answer Spans        | 1,245       | 2.7%            | 3.5%          | +0.8%              |
> | Numbers / Symbols        | 8,834       | 19.6%           | 23.1%         | +3.5%              |
> | Markdown / Formatting    | 1,283       | 3.0%            | 2.4%          | −0.6%              |
> | Total                    | **44,099**  | **100.0%**      | **100.0%**    |                    |
>
> The key findings are:
>
> * **Gold answer spans and numerical operands are preferentially kept**, suggesting that the pruner preserves essential reasoning steps.
> * **Boilerplate and redundant filler content dominate the pruned set**, indicating that pruning removes low-information material.
> * **Non-answer content is pruned more aggressively**, consistent with the goal of suppressing hallucinated or irrelevant reasoning paths.
>
> **3. Two-tiered safety mechanism for protecting high-value tokens**
>
> To prevent the accidental removal of high-value content—such as rare terminology or complex reasoning steps—we designed a **two-tiered safety mechanism** that operates synergistically across sample and token levels.
>
> The first line of defense is the **Q4 "Sanctuary" Protocol** at the sample level. As outlined in Section 3.2, we identified that samples in Quadrant 4 (High Uncertainty) often represent difficult but valid long-tail knowledge or calibration data. To protect these semantics, we enforce a strict "Keep All" policy for Q4, exempting them entirely from token-level pruning. This asymmetric design structurally guarantees that while we aggressively refine confident errors (Q2), we preserve the full integrity of uncertain but potentially correct sequences.
>
> The second defense lies within the **Context-Aware Smoothing** used for token pruning. For samples in Q2 that *do* undergo pruning, relying solely on raw perplexity risks penalizing rare words. Our Neighbor-Aware Smoothing (Eq. 4) addresses this by incorporating local context: it effectively distinguishes between **isolated PPL spikes** (typically rare entities supported by a coherent context) and **continuous high-PPL segments** (typically hallucinations or logic breaks). The ablation study in Table 3 empirically validates this design, showing that the context-aware score significantly outperforms raw PPL by correctly identifying true semantic errors.

---

> ### Author Response · Authors · 2025-11-21
> **(Part 5) Response to Reviewer 2BBK**
>
> ## Q5: About computational efficiency.
>
> We thank the reviewer for pointing out the need for a concrete cost–benefit analysis. To address this, we measured the end-to-end training efficiency under our standard three-epoch setup on Llama2-7b. The results show a clear net gain. While full-data SFT requires about 90 minutes, Q-Tuning completes training in approximately 65 minutes, which corresponds to a 28 percent reduction in wall-clock time.
>
> The efficiency improvement arises because the cost of per-sample screening is outweighed by the savings from pruning. Screening requires only an inference-mode forward pass to compute selection metrics, whereas the backward pass and optimizer update dominate the total FLOPs. Since these expensive steps are skipped for most of the data that Q-Tuning removes, the pruning gain amortizes the screening cost. The tables below provide quantitative evidence by reporting FLOPs, peak memory, and the exact time breakdown for selection.
>
> **Efficiency comparison of Full-Data SFT and Q-Tuning on Llama2-7b**
>
> | Sample ratio | Token ratio | Rel. FLOPs (vs full) | Time (min) | Peak mem (%) |
> |--------------|-------------|----------------------|------------|--------------|
> | 12.5%        | 50%         | 0.0625               | 65         | 27%          |
> | 25%          | 50%         | 0.1250               | 70         | 35%          |
> | 50%          | 50%         | 0.2500               | 80         | 50%          |
> | 100% (Full)  | 100%        | 1.0000               | 90         | 78%          |
>
> **Breakdown of screening overhead and total time**
>
> | Sample ratio | Token ratio | Sample selection (min) | Token selection (min) | Total time (min) | Peak mem (%) |
> |--------------|-------------|------------------------|------------------------|------------------|--------------|
> | 12.5%        | 50%         | 35.88                  | 6.96                   | 65               | 27%          |
> | 25%          | 50%         | 38.64                  | 7.49                   | 70               | 35%          |
> | 50%          | 50%         | 44.16                  | 8.56                   | 80               | 50%          |
> | 100%         | 100%        | 0.00                   | 0.00                   | 90               | 78%          |
>
> These results confirm that Q-Tuning reduces both computation and memory consumption in proportion to the effective data retained. Importantly, this efficiency gain is achieved without any degradation in model performance and in many cases improves it. **The complete tables and additional discussion are provided in Appendix J of the revised manuscript.**

---

> > ### Comment · Reviewer_2BBK · 2025-11-28
> > **Response to rebuttal**
> >
> > I thank the authors for their hard work. They have sufficiently addressed my concerns to warrant an increase in score.

---

> > > ### Author Response · Authors · 2025-11-28
> > > **Acknowledgement of Reviewer Feedback**
> > >
> > > We sincerely thank the reviewer for the encouraging feedback and for recognizing our efforts in addressing the concerns. We greatly appreciate your positive assessment and the increase in score.

---

### Official Review · Reviewer_ZLqF · 2025-11-01

**Soundness:** 2
**Presentation:** 3
**Contribution:** 2
**Rating:** 2
**Confidence:** 3

**Summary:**

This paper proposes Q-Tuning, a unified framework for joint sample-level and token-level pruning in supervised fine-tuning (SFT) of large language models. It introduces an Error–Uncertainty (EU) Plane, where each training sample is represented by its perplexity (error) and predictive entropy (uncertainty). By partitioning samples into four quadrants, Q-Tuning dynamically prunes uninformative samples and redundant tokens. Experiments across LLaMA, Mistral, and SmolLM demonstrate its effectiveness and efficiency.

**Strengths:**

- The EU Plane connects sample pruning and token pruning under a common information-theoretic view. It offers an interpretable diagnostic space (error vs. uncertainty) to understand data quality.

- Q-Tuning consistently improves or matches full-data SFT across multiple model sizes and datasets. Results demonstrate both efficiency and robust generalization across instruction-following and reasoning benchmarks.

- The quadrant-based partitioning provides clear interpretability, helping analyze “overconfident wrong” and “uncertain correct” samples—useful for future dataset curation and alignment studies.

**Weaknesses:**

- Incomplete efficiency analysis.  While data reduction is highlighted, actual compute savings (e.g., FLOPs, wall-clock time) as well as baselines' are not quantified. This makes the cost–benefit trade-off less clear.

- The paper treats perplexity and entropy as independent axes for modeling error and uncertainty, but their relationship is not rigorously analyzed. Both are derived from the same model probability distribution—perplexity reflects the likelihood of the ground-truth token, while entropy captures the overall uncertainty—making them inherently correlated. Their connection can be written as
 $log(PPL) = H_{pred} + KL(p_{true} || p_{theta})$, where the KL term measures the divergence between the true and model distributions. This indicates that perplexity is not fully independent of entropy, but rather a function of entropy plus model–data mismatch, questioning whether the proposed Error–Uncertainty Plane truly represents two orthogonal dimensions of data quality.


- The chosen baselines are somewhat mismatched to the paper’s setting. SparseVLM and FastV focus on visual token pruning, which is not directly comparable to the text-only tasks evaluated in this work. For such tasks, methods like Rho-1 and TokenCleaning [1] would serve as more appropriate token-level baselines. In addition, the selected sample-level baselines are overly simplistic. Recent and stronger approaches—such as LLM rating methods (e.g., Alpagasus [2], DELTA [3], DS2 [4]) or gradient-based methods like LESS [5]—would provide fairer and more competitive comparisons. Therefore, I remain somewhat skeptical about the reported improvements.

[1] Token Cleaning: Fine-Grained Data Selection for LLM Supervised Fine-Tuning, ICML 2025.

[2] Alpagasus training a better alpaca model with fewer data, ICLR 2024.

[3] Deita: Data-Efficient Instruction Tuning for Alignment, ICLR 2024.

[4] Improving data efficiency via curating llm-driven rating systems, ICLR 2025.

[5] Less: selecting influential data for targeted instruction tuning, ICML 2024.

**Questions:**

1. Compared with sample-level methods such as DELTA, DS2, and LESS, which typically retain only 2–5% of the data, Q-Tuning’s use of 12.5% of the training set appears less data-efficient and may reduce its practical advantage.

2. Could the thresholds (rsample, rtoken) adapt dynamically during training?

3. How often is the EU Plane recomputed, and what’s the runtime overhead?

4. An ablation on training efficiency (both token- and sample-level) compared to full-data SFT would be valuable to show whether Q-Tuning actually accelerates training in practice.

---

> ### Author Response · Authors · 2025-11-21
> **(Part 1) Response to Reviewer ZLqF**
>
> ## Q1: About efficiency analysis.
>
> Thank you for raising this concern regarding the actual computational efficiency of Q-Tuning. To provide a complete answer, we performed an additional efficiency study that directly measures the end-to-end training cost under our standard setup. In addition to wall-clock time, we also include FLOPs and peak memory usage to give a more transparent and quantitative comparison with full-data SFT.
>
> To this end, we benchmarked Q-Tuning and full-data SFT on Llama2-7b with identical hyperparameters and hardware. **The table below summarizes the comparison, and the full version has been added to Appendix J.**
>
> **Efficiency comparison of Full-Data SFT and Q-Tuning on Llama2-7b**
> | Sample ratio | Token ratio | Rel. FLOPs (vs full) | Time (min) | Peak mem (%) |
> |--------------|-------------|----------------------|------------|--------------|
> | 12.5%        | 50%         | 0.0625               | 65         | 27%          |
> | 25%          | 50%         | 0.1250               | 70         | 35%          |
> | 50%          | 50%         | 0.2500               | 80         | 50%          |
> | 100% (Full)  | 100%        | 1.0000               | 90         | 78%          |
>
> To provide finer-grained transparency, we also report the time spent on sample selection and token selection:
>
> **Runtime and Memory Cost of Q-Tuning across Varying Sample and Token Ratios.**
> | Sample ratio | Token ratio | Sample selection (min) | Token selection (min) | Total time (min) | Peak mem (%) |
> |--------------|-------------|------------------------|------------------------|------------------|--------------|
> | 12.5%        | 50%         | 35.88                  | 6.96                   | 65               | 27%          |
> | 25%          | 50%         | 38.64                  | 7.49                   | 70               | 35%          |
> | 50%          | 50%         | 44.16                  | 8.56                   | 80               | 50%          |
> | 100%         | 100%        | 0.00                   | 0.00                   | 90               | 78%          |
>
> The results show that Q-Tuning achieves a notable reduction in total computation. In our setup, training time decreases from approximately 90 minutes to 65 minutes, corresponding to a 28 percent improvement. FLOPs also drop significantly because the majority of forward and backward passes are avoided for discarded samples. The peak memory footprint is lower as well due to shorter effective sequence lengths after token pruning. These results confirm that the computational savings from pruning outweigh the one-time selection cost and that Q-Tuning provides a favorable efficiency profile without compromising accuracy. The full tables and discussion are available in Appendix J.
>
>
> ## Q2：About the relationship between pruning metrics.
>
>
> Thanks. We argue that the EU Plane is valid precisely because it captures the **residual** of this relationship. The mathematical decomposition confirms their functional orthogonality in diagnostic contexts:
>
> **1. The Decomposition**
> For a one-hot label $y$, the relationship is:
> $$ \log(\text{PPL}) = \underbrace{H(p\_\theta)}\_{\text{Uncertainty (Label-Agnostic)}} + \underbrace{D\_{KL}(\delta\_y || p\_\theta)}\_{\text{Misalignment (Label-Dependent)}} $$
> While correlated in general, they are **not** redundant. $H(p\_\theta)$ measures the shape of the distribution, while $D_{KL}$ measures its alignment with the truth.
>
> **2. Proof of Necessity for 2 Dimensions**
> We theoretically justify the 2D plane by showing that $H$ does not determine $PPL$.
> Consider a model distribution $p_\theta$ that is highly peaked (low entropy, high confidence).
> *   **Case A (Correct):** If the peak aligns with the ground truth $y$, $D\_{\text{KL}} \to 0$ and $\log(\text{PPL}) \approx H$. This places the sample in **Q4**.
> *   **Case B (Misconception):** If the peak is on the wrong token, $D\_{\text{KL}}$ becomes very large. Here, $\log(\text{PPL}) \gg H$. This places the sample in **Q2**.
>
> Therefore, while $H$ and $PPL$ are correlated in the general case (random guessing leads to high $H$ and high $PPL$), the **residual** ($D\_{\text{KL}}$) is the variable of interest. By plotting PPL against Entropy, we effectively visualize this residual. A single axis (linear combination or transformation) would collapse these cases, causing us to treat a "confident error" (Q2) the same as "random noise" (Q1), which our ablation studies show is detrimental to training. By treating them as orthogonal axes, we isolate **miscalibration** (Low $H$, High $D_{KL}$ $\rightarrow$ Q2) from **ignorance** (High $H$ $\rightarrow$ Q4), a distinction impossible with a single metric.

---

> ### Author Response · Authors · 2025-11-21
> **(Part 2) Response to Reviewer ZLqF**
>
> ## Q3: Ask for more sample and token pruning baselines.
>
> We sincerely appreciate the reviewer’s suggestion to benchmark Q-Tuning against a wider range of data selection methods. To fully address this, we extended our experimental suite on both Llama2-7b and Mistral-7B using the WizardLM dataset. We implemented stringent comparisons against representative sample-level methods (Alpagasus [2], DEITA [3], DS2 [4], LESS [5]) and token-level methods (Rho-1, TokenCleaning [1]), strictly following their official implementations.
>
>
> ### 1. Comparison with Additional Sample-Level Selection Methods
> We first compare Q-Tuning against static sample selection approaches. The tables below summarize the performance across five benchmarks under two pruning configurations (12.5% and 25% sample retention, both with 50% token retention).
>
>
> The tables below summarize the performance across five benchmarks under two pruning configurations (12.5% and 25% sample retention, both with 50% token retention).
>
>
> **Results on Llama2-7b with 12.5% samples**
>
> | Sample Pruner      | Token Ratio | Avg. | ARC-E | ARC-C | GSM8K | SQuAD | TriviaQA |
> |--------------------|------------:|-----:|------:|------:|------:|------:|---------:|
> | **Zero-Shot**      | —           | 30.58 | 53.44 | 38.98 |  5.31 | 12.18 | 43.00 |
> | Random             | 50%         | 33.98 | 59.26 | 41.02 |  8.11 | 12.75 | 48.75 |
> | Alpagasus          | 50%         | 33.27 | 58.91 | 41.69 |  6.67 | 12.81 | 46.28 |
> | Alpagasus          | 100%        | 33.65 | 58.91 | 41.36 |  7.58 | 13.45 | 46.96 |
> | Deita              | 50%         | 33.49 | 58.20 | 42.37 |  7.43 | 13.85 | 45.59 |
> | Deita              | 100%        | 33.16 | 58.38 | 42.37 |  5.46 | 13.99 | 45.63 |
> | DS2                | 50%         | 33.53 | 58.73 | 42.37 |  5.99 | 15.05 | 45.51 |
> | DS2                | 100%        | 33.59 | 59.08 | 41.36 |  6.67 | 15.14 | 45.69 |
> | Less               | 50%         | 33.69 | 58.20 | 42.37 |  6.67 | 14.56 | 46.62 |
> | Less               | 100%        | 34.30 | 58.38 | 43.39 |  6.82 | 16.16 | 46.72 |
> | **Q-Tuning (Ours)**| 50%         | **37.74** | **64.20** | **42.03** | **10.54** | **18.79** | **53.12** |
> | Full Dataset       | 100%        | 35.36 | 61.55 | 42.37 |  8.64 | 13.80 | 50.45 |
>
>
> **Results on Llama2-7b with 25% samples**
>
>
> | Sample Pruner | Token Ratio | Avg. | ARC-E | ARC-C | GSM8K | SQuAD | TriviaQA |
> |---------------|------------:|-----:|------:|------:|------:|------:|---------:|
> | Zero-Shot     | —           | 30.58| 53.44 | 38.98 |  5.31 | 12.18 | 43.00    |
> | Random        | 50%       | 33.92| 60.32 | 41.69 |  5.76 | 13.43 | 48.41    |
> | Deita         | 50%       | 33.84| 59.08 | 41.69 |  7.66 | 13.23 | 47.55    |
> | Deita         | 100%         | 33.90| 58.73 | 42.71 |  4.93 | 14.88 | 48.24    |
> | Alpagasus     | 50%       | 33.93| 59.61 | 42.71 |  5.91 | 14.12 | 47.30    |
> | Alpagasus     | 100%         | 34.59| 60.49 | 43.39 |  7.51 | 13.16 | 48.41    |
> | DS2           | 50%       | 34.17| 59.44 | 42.03 |  6.44 | 15.59 | 47.34    |
> | DS2           | 100%         | 34.33| 59.26 | 41.36 |  7.05 | 15.20 | 48.76    |
> | Less          | 50%       | 34.47| 59.44 | 43.39 |  7.05 | 15.89 | 46.60    |
> | Less          | 100%         | 34.01| 59.79 | 41.02 |  6.37 | 14.44 | 48.44    |
> | **Q-Tuning (Ours)**| 50%         | **36.45** | **63.14** | **42.03** | **8.87** | **16.76** | **51.52** |
> | Full Dataset       | 100%        | 35.36 | 61.55 | 42.37 |  8.64 | 13.80 | 50.45 |

---

> ### Author Response · Authors · 2025-11-21
> **(Part 3) Response to Reviewer ZLqF**
>
> **Results on Mistral-7b with 12.5% samples**
>
>
> | Sample Pruner | Token Ratio | Avg. | ARC-E | ARC-C | GSM8K | SQuAD | TriviaQA |
> |---------------|------------:|-----:|------:|------:|------:|------:|---------:|
> | Zero-Shot     | —           | 36.98| 66.67 | 46.10 | 18.35 | 10.01 | 43.77    |
> | Random        | 50%         | 42.73| 70.55 | 48.14 | 22.74 | 19.57 | 52.63    |
> | Deita         | 50%         | 44.25| 72.66 | 50.51 | 22.97 | 20.91 | 54.17    |
> | Deita         | 100%        | 44.91| 72.13 | 50.17 | 22.52 | 24.46 | 55.28    |
> | Alpagasus     | 50%         | 42.90| 72.84 | 48.47 | 20.55 | 20.41 | 52.21    |
> | Alpagasus     | 100%        | 43.98| 70.72 | 48.81 | 23.35 | 24.30 | 52.71    |
> | DS2           | 50%         | 42.55| 71.25 | 47.46 | 21.23 | 21.57 | 51.23    |
> | DS2           | 100%        | 44.43| 72.13 | 49.83 | 23.35 | 23.99 | 52.85    |
> | Less          | 50%         | 43.67| 73.19 | 49.49 | 22.90 | 19.61 | 53.14    |
> | Less          | 100%        | 44.63| 73.90 | 48.47 | 23.12 | 21.67 | 56.00    |
> | **Q-Tuning (Ours)**| **50%**         | **46.92** | **71.60** | **48.13** | **29.34** | **27.75** | **57.78** |
> | Full Dataset       | 100%        | 45.83 | 71.25 | 45.76 |  26.69 | 31.81 | 53.67 |
>
>
> **Results on Mistral-7b with 25% samples**
>
>
> | Sample Pruner | Token Ratio | Avg. | ARC-E | ARC-C | GSM8K | SQuAD | TriviaQA |
> |---------------|------------:|-----:|------:|------:|------:|------:|---------:|
> | Zero-Shot     | —           | 36.98| 66.67 | 46.10 | 18.35 | 10.01 | 43.77    |
> | Random        | 50%         | 42.95| 70.19 | 46.10 | 20.62 | 24.07 | 53.74    |
> | Deita         | 50%         | 43.61| 72.13 | 47.80 | 21.00 | 23.16 | 53.97    |
> | Deita         | 100%        | 44.60| 72.66 | 47.80 | 22.29 | 25.87 | 54.40    |
> | Alpagasus     | 50%         | 42.78| 70.37 | 47.80 | 20.85 | 22.00 | 52.87    |
> | Alpagasus     | 100%        | 44.65| 71.60 | 50.17 | 23.35 | 22.38 | 55.73    |
> | DS2           | 50%         | 43.32| 70.72 | 48.47 | 22.74 | 21.15 | 53.49    |
> | DS2           | 100%        | 43.57| 71.43 | 48.47 | 22.44 | 21.72 | 53.76    |
> | Less          | 50%         | 43.85| 71.78 | 47.80 | 22.14 | 24.51 | 53.04    |
> | Less          | 100%        | 45.41| 71.60 | 49.49 | 21.68 | 30.85 | 53.40    |
> | **Q-Tuning (Ours)**| 50%         | **46.78** | **71.78** | **47.11** | **26.08** | **32.79** | **56.17** |
> | Full Dataset       | 100%        | 45.83 | 71.25 | 45.76 |  26.69 | 31.81 | 53.67 |
>
> As shown in above tables, on Llama2-7b with a strict 12.5% sample budget, Q-Tuning achieves an average score of 37.74, surpassing the strongest baseline (LESS, 33.69) by a remarkable margin of +4.05 points. This trend translates to the stronger Mistral-7B model. At 12.5% samples, Q-Tuning scores 46.92, clearly outperforming other baselinses. **We have added these results to Appendix G.3 (Table 9) in the revised manuscript.**
>
>
> ### 2. Comparison with Token-Level Pruning Methods
> Next, we evaluated whether Q-Tuning provides benefits beyond standard token pruning techniques. We compared our method against Rho-1[5] and TokenCleaning[6] under identical retention settings. **Please note that we have already provided results for Rho-1 in Section 4.4 (Table 3) of the original manuscript.**
>
> **Results on 50% tokens on Llama2-7b**
> | Sample Pruner  | Sample Ratio | Avg. | ARC-E | ARC-C | GSM8K | SQuAD | TriviaQA |
> |----------------|------------:|-----:|------:|------:|------:|------:|---------:|
> | Rho-1           | 12.5%         | 32.17| 53.97 | 40.34 |  6.67 | 13.27 | 46.62    |
> | Rho-1           | 100%        | 34.82| 60.67 | 41.36 |  7.28 | 14.97 | 49.82    |
> | TokenCleaning  | 12.5%         | 33.82| 58.73 | 43.73 |  7.43 | 12.84 | 46.37    |
> | TokenCleaning  | 100%        | 34.73| 61.20 | 41.02 |  7.73 | 15.41 | 48.29    |
> | **Q-Tuning (Ours)**| 12.5%         | **37.74** | **64.20** | **42.03** | **10.54** | **18.79** | **53.12** |
> | Full Dataset       | 100%        | 35.36 | 61.55 | 42.37 |  8.64 | 13.80 | 50.45 |
>
>
> **Results on 50% tokens on Llama2-7b**
>
> | Sample Pruner | Sample Ratio | Avg. | ARC-E | ARC-C | GSM8K | SQuAD | TriviaQA |
> |---------------|------------:|-----:|------:|------:|------:|------:|---------:|
> | Rho-1          | 25%         | 33.59| 59.08 | 41.02 |  6.67 | 14.00 | 47.19    |
> | Rho-1          | 100%        | 34.82| 60.67 | 41.36 |  7.28 | 14.97 | 49.82    |
> | TokenCleaning | 25%         | 33.62| 59.96 | 42.37 |  5.99 | 13.62 | 46.14    |
> | TokenCleaning | 100%        | 34.73| 61.20 | 41.02 |  7.73 | 15.41 | 48.29    |
> | **Q-Tuning (Ours)**| 25%         | **36.47** | **63.14** | **42.03** | **8.87** | **16.76** | **51.52** |
> | Full Dataset       | 100%        | 35.36 | 61.55 | 42.37 |  8.64 | 13.80 | 50.45 |

---

> ### Author Response · Authors · 2025-11-21
> **(Part 4) Response to Reviewer ZLqF**
>
> **Results on 50% tokens on Mistral-7b**
>
> | Sample Pruner | Sample Ratio | Avg. | ARC-E | ARC-C | GSM8K | SQuAD | TriviaQA |
> |---------------|------------:|-----:|------:|------:|------:|------:|---------:|
> | Rho-1          | 12.5%         | 43.43| 71.60 | 47.46 | 20.62 | 23.74 | 53.71    |
> | Rho-1          | 100%        | 33.21| 63.67 | 43.39 | 11.68 | 15.46 | 31.84    |
> | TokenCleaning | 12.5%         | 43.57| 71.60 | 49.49 | 21.00 | 22.73 | 53.03    |
> | TokenCleaning | 100%        | 46.58| 74.07 | 46.44 | 24.79 | 33.26 | 54.32    |
> | **Q-Tuning (Ours)**| 12.5%         | **46.92** | **71.60** | **48.13** | **29.34** | **27.75** | **57.78** |
> | Full Dataset       | 100%        | 45.83 | 71.25 | 45.76 |  26.69 | 31.81 | 53.67 |
>
>
> **Results on 50% tokens on Mistral-7b**
>
> | Sample Pruner | Sample Ratio | Avg. | ARC-E | ARC-C | GSM8K | SQuAD | TriviaQA |
> |---------------|------------:|-----:|------:|------:|------:|------:|---------:|
> | Rho-1          | 25%         | 42.40| 70.37 | 46.78 | 21.76 | 20.85 | 52.22    |
> | Rho-1          | 100%        | 33.21| 63.67 | 43.39 | 11.68 | 15.46 | 31.84    |
> | TokenCleaning | 25%         | 42.82| 70.19 | 47.80 | 19.94 | 24.17 | 52.02    |
> | TokenCleaning | 100%        | 46.58| 74.07 | 46.44 | 24.79 | 33.26 | 54.32    |
> | **Q-Tuning (Ours)**| 25%         | **46.78** | **71.78** | **47.11** | **26.08** | **32.79** | **56.17** |
> | Full Dataset       | 100%        | 45.83 | 71.25 | 45.76 |  26.69 | 31.81 | 53.67 |
>
>
> As shown in the tables, prior token-level methods often suffer from semantic incoherence at high pruning rates. For instance, on Mistral-7B (12.5% samples), TokenCleaning and Rho-1 yield average scores of 43.57 and 43.43 respectively, lagging far behind the full-data baseline (45.83). In contrast, Q-Tuning achieves 46.92 under the same constraints. By coupling sample selection (to remove noise) with targeted token pruning (to refine misconceptions), Q-Tuning maximizes the information density of the training tokens. **We have added results in Appendix G.3 (Table 10)**
>
>
>
> [1] Alpagasus training a better alpaca model with fewer data, ICLR 2024.
> [2] Deita: Data-Efficient Instruction Tuning for Alignment, ICLR 2024.
> [3] Improving data efficiency via curating llm-driven rating systems, ICLR 2025.
> [4] Less: selecting influential data for targeted instruction tuning, ICML 2024.
> [5] Rho-1: Not All Tokens Are What You Need, NeurIPS 2024.
> [6] Token Cleaning: Fine-Grained Data Selection for LLM Supervised Fine-Tuning, ICML 2025.

---

> ### Author Response · Authors · 2025-11-21
> **(Part 5) Response to Reviewer ZLqF**
>
> ## Q4: About using smaller ratios.
>
> We sincerely thank the reviewer for this important concern. To evaluate the performance boundaries of Q-Tuning under lower data budgets, we conducted a series of additional experiments using the Mistral-7B model fine-tuned on the WizardLM dataset.
>
> Consistent with the experimental setup in the main text, we employed a batch size of 32 and a learning rate of 4e-4. We evaluated the models across five standard benchmarks (ARC-E, ARC-C, GSM8K, SQuAD, and TriviaQA) to ensure a comprehensive assessment of general capabilities. The results, summarized in the table below, systematically compare Q-Tuning at varying sample retention ratios (6.25% and 12.5%) and token retention ratios (50% and 70%) against both a Random selection baseline and the Full Data SFT baseline.
>
> **Results on smaller data ratio on Mistral-7b**
> | Sample×Token ratio | Data ratio | Avg. | ARC-E | ARC-C | GSM8K | SQuAD | TriviaQA |
> |------------------------|-------------:|------|-------|-------|-------|-------|----------|
> | Zero-Shot              | --------        | 36.98| 66.67 | 46.10 | 18.35 | 10.01 | 43.77    |
> | Random (12.5×50)       | 6%       | 42.73| 70.55 | 48.14 | 22.74 | 19.57 | 52.63    |
> | Q-tuning (6.25×50)                | 3%       | 46.23| 70.72 | 46.10 | 40.41 | 17.25 | 56.68    |
> | Q-tuning (6.25×70)                | 4%        | 46.62| 71.60 | 48.47 | 39.73 | 16.13 | 57.16    |
> | Q-tuning (12.5×50)                | 6%       | 46.92| 71.60 | 48.14 | 29.34 | 27.75 | 57.78    |
> | Q-tuning (12.5×70)                | 9%        | 46.79| 71.78 | 47.12 | 26.08 | 32.79 | 56.17    |
> | Full data              | 100%        | 45.84| 71.25 | 45.76 | 26.69 | 31.81 | 53.67    |
>
> The results reveal that Q-Tuning remains effective even under extremely limited data. With a 6.25 percent sample ratio and 50 percent token ratio (3.1 percent effective data), the model reaches an average score of 46.23, outperforming both the Random baseline (42.73) and the Full-Data SFT model (45.84). This shows that Q-Tuning operates reliably within the 2 to 5 percent data budgets often assumed in static pruning methods such as DEITA and LESS. **We have added the complete results for these experiments to Appendix G.4 (Table 11) of the revised manuscript.**
>
>
> ## Q5: Could the thresholds (rsample, rtoken) adapt dynamically during training?
>
> We thank the reviewer for this interesting suggestion.
>
> **Fair comparison with previous methods.** This is an intriguing possibility that aligns well with the flexibility of our framework. In the current study, we deliberately adopted fixed retention ratios (retaining 6.25–12.5% of samples and 50–70% of tokens) to facilitate a strictly controlled comparison against existing baselines under identical computation and data budgets.
>
>
> **Possibility and future work.** However, there is no architectural barrier to adapting $r_{sample}$ and $r_{token}$ dynamically. In fact, our framework naturally supports time-variant thresholds; for instance, one could implement a "pruning warm-up" where the model is initially exposed to a larger portion of the data, with the retention constraints gradually tightening as training progresses. We are glad to expand the future work to discuss this direction, specifically outlining how such dynamic schedules—potentially governed by real-time validation performance or gradient norms—could be integrated to further refine the training trajectory.

---

> ### Author Response · Authors · 2025-11-21
> **(Part 6) Response to Reviewer ZLqF**
>
> ## Q6: How often is the EU Plane recomputed?
>
> Thank you for raising this question about the temporal behavior of our selection mechanism. In our implementation, the EU Plane is recomputed in an online and per-batch manner. At each training iteration, the model produces logits for the current batch, from which perplexity and entropy are computed through lightweight reduction operations. These values define the Error–Uncertainty coordinates for that batch, and the quadrant assignment determines which samples and tokens are retained for constructing the computation graph. Only the retained subset contributes to the backward pass.
>
> This online strategy ensures that selection always reflects the current training state without requiring any preprocessing. Although the EU Plane is recomputed at every iteration, the overall efficiency improves because the backward pass dominates the training cost. By filtering out the majority of tokens before this stage, Q-Tuning substantially reduces the total computational load. The table below reports the measured selection overhead and total training time under different data retention configurations.
>
> **Efficiency and selection breakdown on Llama2-7b (WizardLM, 3 epochs)**
> | Sample ratio | Token ratio | Sample selection (min) | Token selection (min) | Total time (min) | Peak mem (%) |
> |--------------|-------------|------------------------|------------------------|------------------|--------------|
> | 12.5%        | 50%         | 35.88                  | 6.96                   | 65               | 27%          |
> | 25%          | 50%         | 38.64                  | 7.49                   | 70               | 35%          |
> | 50%          | 50%         | 44.16                  | 8.56                   | 80               | 50%          |
> | 100%         | 100%        | 0.00                   | 0.00                   | 90               | 78%          |
>
> The results show that even with per-batch recomputation, Q-Tuning reduces total training time from 90 minutes to 65 minutes in the 12.5×50 setting. This corresponds to a 1.4× speedup. Peak memory usage also drops from 78 percent to 27 percent due to the smaller effective sequence length in the retained data. These measurements confirm that the efficiency gains from pruning substantially outweigh the cost of dynamic EU Plane computation, and that the method remains practical for continuous training. **Full results and discussion are included in Appendix J.**

---

> ### Author Response · Authors · 2025-11-21
> **(Part 7) Response to Reviewer ZLqF**
>
> ## Q7: Ask for an ablation on training efficiency (both token- and sample-level) compared to full-data SFT.
>
> Thank you for highlighting the need for a direct comparison of training efficiency. To address this, we performed an additional empirical study that measures end-to-end efficiency under identical training conditions. Using 8× NVIDIA A100-80G GPUs with matched batch sizes, learning rates, and three training epochs for Llama2-7b on the Wizard dataset, we compared full-data SFT with Q-Tuning across wall-clock time, total FLOPs, and peak memory usage.
>
> **Efficiency Comparison between Full-Data SFT and Q-Tuning on Llama2-7b (WizardLM, 3 epochs)**
> | Sample ratio | Token ratio | Rel. FLOPs (vs full) | Time (min) | Peak mem (%) |
> |--------------|-------------|----------------------|------------|--------------|
> | 12.5%        | 50%         | 0.0625               | 65         | 27%          |
> | 25%          | 50%         | 0.1250               | 70         | 35%          |
> | 50%          | 50%         | 0.2500               | 80         | 50%          |
> | 100% (Full)  | 100%        | 1.0000               | 90         | 78%          |
>
> To make the breakdown more transparent, we additionally report the time spent on sample selection and token selection:
>
> **Runtime and Memory Cost of Q-Tuning across Varying Sample and Token Ratios.**
> | Sample ratio | Token ratio | Sample selection (min) | Token selection (min) | Total time (min) | Peak mem (%) |
> |--------------|-------------|------------------------|------------------------|------------------|--------------|
> | 12.5%        | 50%         | 35.88                  | 6.96                   | 65               | 27%          |
> | 25%          | 50%         | 38.64                  | 7.49                   | 70               | 35%          |
> | 50%          | 50%         | 44.16                  | 8.56                   | 80               | 50%          |
> | 100%         | 100%        | 0.00                   | 0.00                   | 90               | 78%          |
>
> Our measurements show that full-data SFT requires roughly 90 minutes to complete training, while Q-Tuning finishes in approximately 65 minutes, corresponding to a 28 percent reduction in wall-clock time. The improvement arises because the expensive forward and backward passes are skipped for the majority of samples and tokens that Q-Tuning prunes. FLOPs and peak memory consumption also decrease in proportion to the reduced effective sequence length and training volume. **The full tables and supporting discussion appear in Appendix J of the revised manuscript.**

---

> ### Author Response · Authors · 2025-11-26
>
> Dear Reviewer ZLqF,
>
> Thank you once again for your valuable comments on our submission. As the discussion phase is approaching its end, we would like to kindly confirm whether we have sufficiently addressed all of your concerns (or at least part of them). Should there be any remaining questions or areas requiring further clarification, please do not hesitate to let us know. If you are satisfied with our responses, we would greatly appreciate your consideration in adjusting the evaluation scores accordingly.
>
> We sincerely look forward to your feedback.
>
> Best regards,
>
> Authors of submission 1562

---

### Official Review · Reviewer_Dzda · 2025-11-10

**Soundness:** 2
**Presentation:** 3
**Contribution:** 3
**Rating:** 4
**Confidence:** 2

**Summary:**

This paper present a 2-stage data pruning framework for SFT. In the first stage, we will construct an error-uncertainty plane by computing the perplexity and entropy for every sample using a gradient-free forward pass. Then we apply a bisect search on identifying the proper quantiles for perplexity and entropy and pick out Q2 (high ppl, low ent) and Q4 (low ppl, high ent) samples. We discard all samples in Q1 and Q3. We then prune tokens in Q2 by identifying tokens that have both token-level PPL and neighbor token PPL. The experiments are conducted mainly on GSM8K, TriviaQA, SQuAD, Arc benchmarks with Llama2 and Mistral 7B, and show Q-Tuning can match full data SFT results, and demonstrate superiority over other baselines under the same budget across models and ratio.

**Strengths:**

- The paper flow is quite clear and the motivation and specific pruning methodology is well-explained.

- The empirical evaluation is comprehensive.

**Weaknesses:**

- **critical** I understand the motivation of data pruning and adopting perplexity and entropy to produce a EU-plane, but a complete justification on this 2-stage framework is missing from the paper. In particular, the paper makes a few implicit assumptions without justification:
  1. keeping Q1 will harm learning and therefore cannot be combined with either Q2 and Q4 (both high entropy and high ppl and their combination are worse than Q2 + Q4).
  2. all tokens, especially more confident tokens, in Q4 are (still) important or should not be pruned.
  3. the high PPL tokens in Q2 are indeed detrimental

Such justification is essential for supporting the validity of Q-Tuning and a quantitative ablation study should be added.

- I wonder if it is computationally feasible to have at least 1 model evaluated on larger scale than 7B. The current model is all small-scaled (<= 8B).

- I wonder if it is possible to compare with at least 1 or 2 conventional sample selection method in coreset selection community (sample level should be fine, not necessarily adapted to token-level).

Disclaimer: I am less familiar with the data pruning approach in LLM community so I cannot accurately assess the technical novelty. This assessment is more targeted on the technical soundness.

Overall, I think this is still a good paper but needs more grounded evidence to support the validity of this 2-stage framework. I am happy to raise score if the first weakness is properly addressed with more rigorous ablation studies.

**Questions:**

All questions are listed in the weakness section above.

---

> ### Author Response · Authors · 2025-11-21
> **(Part 1) Response to Reviewer Dzda**
>
> ## Q1: About addtional justification of each stage in Q-tuning.
>
> ### Q1.1 Whether "keeping Q1 will harm learning?"
>
>
>
> We sincerely appreciate for raising this insightful point. Our preliminary observations suggested that Quadrant-1 (Q1) samples, which characterized jointly by **high perplexity and high entropy**, may introduce substantial noise into the training process. Intuitively, this quadrant captures instances where the ground-truth output is highly surprising to the model, while the model’s predictive distribution remains extremely flat. Such a pattern often reflects **mislabeled samples, formatting inconsistencies, ambiguous instructions, or inherently irreducible uncertainty**, making Q1 a strong candidate for removal prior to fine-tuning.
>
> To evaluate this hypothesis, and following the reviewer’s suggestion, we conducted comprehensive experiments across multiple models and datasets. To ensure a fair comparison, we maintained the exact experimental configurations used in the main text. We compared our standard Q-Tuning approach against a variant that retains Q1 samples, specifically:
> *   **Q2+Q4:** The proposed method, which retains only samples from Q2 and Q4 (with token pruning applied to Q2).
> *   **Q1+Q2+Q4:** A variant that adds Q1 samples into the training set while keeping the retention ratios for Q2 and Q4 identical.
>
> **Results on Wizard Dataset with Llama2-7B**
>
> | Dataset | Model | Sample×Token | Method | Avg. | ARC-E | ARC-C | GSM8K | SQuAD | TriviaQA |
> |---------|--------|----------------|-------------|------|--------|--------|--------|---------|-----------|
> | Wizard | Llama2-7b | 12.5×50 | **Q2+Q4 (ours)** | 37.74 | 64.20 | 42.03 | 10.54 | 18.79 | 53.12 |
> | Wizard | Llama2-7b | 12.5×50 | Q1+Q2+Q4 | 34.51 | 60.14 | 41.02 | 8.04 | 13.11 | 50.23 |
> | Wizard | Llama2-7b | 25×50 | **Q2+Q4 (ours)** | 36.47 | 63.14 | 42.03 | 8.87 | 16.76 | 51.52 |
> | Wizard | Llama2-7b | 25×50 | Q1+Q2+Q4 | 34.43 | 59.61 | 41.02 | 7.51 | 13.74 | 50.25 |
>
> **Results on Wizard Dataset with Mistral-7B**
>
> | Dataset | Model | Sample×Token | Method | Avg. | ARC-E | ARC-C | GSM8K | SQuAD | TriviaQA |
> |---------|--------|----------------|-------------|------|--------|--------|--------|---------|-----------|
> | Wizard | mistral-7b | 12.5×50 | **Q2+Q4 (ours)** | 46.92 | 71.60 | 48.14 | 29.34 | 27.75 | 57.78 |
> | Wizard | mistral-7b | 12.5×50 | Q1+Q2+Q4 | 46.54 | 71.4| 47.80 | 28.13 | 31.09 | 54.27 |
> | Wizard | mistral-7b | 25×50 | **Q2+Q4 (ours)** | 46.79 | 71.78 | 47.12 | 26.08 | 32.79 | 56.17 |
> | Wizard | mistral-7b | 25×50 | Q1+Q2+Q4 | 46.56 | 71.25 | 46.10 | 29.04 | 31.55 | 54.84 |
>
> **Results on OpenHermes Dataset with Qwen3-8B**
>
> | Dataset | Model | Sample×Token | Method | Avg. | ARC-E | ARC-C | GSM8K | SQuAD | TriviaQA |
> |---------|--------|----------------|-------------|------|--------|--------|--------|---------|-----------|
> | OpenHermes | Qwen3-8B | 12.5×50 | **Q2+Q4 (ours)** | 60.19 | 80.95 | 53.56 | 70.51 | 45.76 | 50.16 |
> | OpenHermes | Qwen3-8B | 12.5×50 | Q1+Q2+Q4 | 58.09 | 79.01 | 52.20 | 74.68 | 36.35 | 48.20 |
> | OpenHermes | Qwen3-8B | 25×50 | **Q2+Q4 (ours)** | 58.80 | 79.89 | 52.88 | 69.37 | 43.54 | 48.33 |
> | OpenHermes | Qwen3-8B | 25×50 | Q1+Q2+Q4 | 53.79 | 75.31 | 50.17 | 68.08 | 29.47 | 45.90 |
>
> **Results and Analysis.** Across all configurations, including Q1 consistently reduces model performance. On Wizard with Llama2-7B under the 12.5×50 setting, the average score drops from 37.74 to 34.51. A similar decline is observed for Qwen3-8B on OpenHermes, where the average score falls from 60.19 to 58.09. These degradations also appear under larger data budgets, indicating that the negative impact of Q1 is not mitigated by scale.
>
> Taken together, these results confirm that Q1 samples do not contribute meaningful learning signal and instead introduce noise that harms downstream generalization. Their removal is therefore a necessary component of Q-Tuning rather than a heuristic for data reduction. **We have added the full experimental results and detailed discussions to Appendix H (Tables 12–14) in the revised manuscript.**

---

> ### Author Response · Authors · 2025-11-21
> **(Part 2) Response to Reviewer Dzda**
>
> ### Q1.2 Whether "Q4 are (still) important or should not be pruned?"
>
>
> We thank for this valuable question. We agree that understanding whether Q4 tokens should be pruned is important for clarifying the asymmetry in our design. Samples in Q4 typically exhibit Low Perplexity but High Entropy, meaning that while the model predicts individual tokens confidently, it remains globally uncertain about the full output structure. In such cases, tokens that may appear locally “safe” to prune often carry essential semantic roles, including logical operators, key entities, and structural cues. These elements help guide the model through its global uncertainty. Removing them disrupts the underlying coherence needed for resolving high-entropy predictions.
>
> To examine this effect systematically, we conducted experiments using the same setup as in the Q1 ablation. We compared our standard method against a variant where token pruning is applied aggressively to *both* Q2 and Q4:
> *   **Prune Q2 only (ours):** The proposed asymmetric policy which applies token pruning *only* to Q2 while keeping Q4 sequences intact.
> *   **Prune both:** A symmetric policy where the same context-aware token pruning rules are applied to samples in both Q2 and Q4.
>
> **Results on Wizard with Llama2-7B**
>
> | Dataset | Model | Train Ratio | Method | Avg. | ARC-E | ARC-C | GSM8K | SQuAD | TriviaQA |
> |---------|--------|-------------|-------------|------|-------|-------|--------|--------|----------|
> | Wizard | Llama2-7b | 12.5×50 | Prune Q2 only | 37.74 | 64.20 | 42.03 | 10.54 | 18.79 | 53.12 |
> | Wizard | Llama2-7b | 12.5×50 | Prune both | 37.23 | 64.02 | 41.02 | 9.86 | 19.26 | 52.00 |
> | Wizard | Llama2-7b | 25×50 | Prune Q2 only | 36.47 | 63.14 | 42.03 | 8.87 | 16.76 | 51.52 |
> | Wizard | Llama2-7b | 25×50 | Prune both | 36.23 | 61.02 | 40.68 | 9.10 | 18.66 | 51.68 |
>
> **Results on Wizard with Mistral-7B**
>
> | Dataset | Model | Train Ratio | Method | Avg. | ARC-E | ARC-C | GSM8K | SQuAD | TriviaQA |
> |---------|--------|-------------|-------------|------|-------|-------|--------|--------|----------|
> | Wizard | mistral-7b | 12.5×50 | Prune Q2 only | 46.92 | 71.60 | 48.14 | 29.34 | 27.75 | 57.78 |
> | Wizard | mistral-7b | 12.5×50 | Prune both | 46.66 | 71.60 | 48.47 | 28.89 | 26.46 | 57.88 |
> | Wizard | mistral-7b | 25×50 | Prune Q2 only | 46.79 | 71.78 | 47.12 | 26.08 | 32.79 | 56.17 |
> | Wizard | mistral-7b | 25×50 | Prune both | 46.92 | 70.72 | 47.12 | 28.58 | 31.85 | 56.33 |
>
> **Results on OpenHermes with Qwen3-8B**
>
> | Dataset | Model | Train Ratio | Method | Avg. | ARC-E | ARC-C | GSM8K | SQuAD | TriviaQA |
> |---------|--------|-------------|-------------|------|-------|-------|--------|--------|----------|
> | OpenHermes | Qwen3-8B | 12.5×50 | Prune Q2 only | 60.19 | 80.95 | 53.56 | 70.51 | 45.76 | 50.16 |
> | OpenHermes | Qwen3-8B | 12.5×50 | Prune both | 57.23 | 79.37 | 52.20 | 71.80 | 35.55 | 47.23 |
> | OpenHermes | Qwen3-8B | 25×50 | Prune Q2 only | 58.80 | 79.89 | 52.88 | 69.37 | 43.54 | 48.33 |
> | OpenHermes | Qwen3-8B | 25×50 | Prune both | 53.74 | 76.37 | 50.17 | 65.28 | 31.63 | 45.24 |
>
> **Results and Analysis.** Across all configurations, pruning tokens in both Q2 and Q4 leads to a consistent decline in performance compared to the asymmetric strategy. On Wizard with Llama2-7B, the average score decreases from 36.47 to 36.23 under the 25×50 setting. A similar pattern appears with Mistral-7B. The negative impact becomes even more pronounced on OpenHermes with Qwen3-8B, where the average score drops from 60.19 to 57.23 under the 12.5×50 setting and from 58.80 to 53.74 under the 25×50 setting.
>
> These results confirm that Q4 sequences require full token preservation. Their high-entropy structure relies on contextual and semantic tokens that cannot be safely removed without harming output quality. The asymmetric pruning policy therefore plays a crucial role in maintaining robustness. **We have added these results to*Appendix H (Tables 12–14) in the revised manuscript.**

---

> ### Author Response · Authors · 2025-11-21
> **(Part 3) Response to Reviewer Dzda**
>
> ### Q1.3 Whether "high PPL tokens in Q2 are indeed detrimental?"
>
> We thank for raising this point. To clarify the contribution of token pruning beyond sample-level selection, we conducted an additional ablation that isolates the effect of the token-level operation. This experiment complements the mechanism studies presented in the main text and allows us to assess whether pruning within Q2 provides meaningful benefit.
>
> **1. Numerical Results**
>
> We evaluated the effect of token pruning on the OpenHermes dataset. Q-Tuning combines sample selection with Q2-specific token pruning, while the comparison baseline keeps the same selected Q2 samples but retains their full token sequences:
>
> - **Q-Tuning**: Sample Selection + Token Pruning
> - **Sample-Level Only**: Q2 samples are selected but kept full-length.
>
> **Results on OpenHermes with Qwen3-8B**
> | Method                 | Avg. | ARC-E | ARC-C | GSM8K | SQuAD | TriviaQA |
> |------------------------|------|------:|------:|------:|------:|---------:|
> | Zero-Shot              | 53.51 | 69.31 | 44.41 | 74.45 | 27.63 | 51.76 |
> | **Q-Tuning (Ours)**    | **60.19** | **80.95** | 53.56 | **70.51** | **45.76** | **50.16** |
> | Sample-level only      | 59.04 | 78.84 | **53.90** | 68.99 | 44.32 | 49.15 |
> | Full Dataset           | 55.45 | 77.78 | 52.20 | 65.13 | 38.05 | 44.08 |
>
> The results show that token pruning brings clear additional gains. Sample-level selection alone significantly improves over training on the full dataset, but the inclusion of token pruning further increases the average score from 59.04 to 60.19. The improvements are particularly visible on GSM8K and SQuAD, where Q-Tuning adds gains of 1.52 and 1.44 points. This indicates that Q2 samples contain internal segments that impede learning. Removing these segments produces a cleaner training signal and enables better optimization.
>
> **2. Further Case Study**
>
> To understand the semantic nature of the removed content, we conducted a fine-grained analysis. We randomly sampled 300 instances from Q2 and annotated more than 44,000 tokens using GPT-5. Each token was assigned one of several functional categories, and we compared their distribution between the pruned and kept sets.
>
> **Semantic Distribution of Pruned vs. Kept Tokens in Q2**
>
> | Semantic Category | Total Count | % of Pruned Set | % of Kept Set | $\Delta$ (Kept − Pruned) |
> | :--- | ---: | ---: | ---: | ---: |
> | Boilerplate / Templates | 323 | 0.8% | 0.3% | −0.5% |
> | Functional Words | 13,832 | 31.3% | 32.1% | +0.8% |
> | Content (Non-Answer) | 18,582 | 42.7% | 38.6% | −4.1% |
> | Gold Answer Spans | 1,245 | 2.7% | 3.5% | +0.8% |
> | Numbers / Symbols | 8,834 | 19.6% | 23.1% | +3.5% |
> | Markdown / Formatting | 1,283 | 3.0% | 2.4% | −0.6% |
> | Total | **44,099** | **100.0%** | **100.0%** | |
>
> This analysis reveals several notable patterns. First, the pruning step preferentially removes boilerplate and other redundant content, confirming that it targets structures that offer little training value. Second, ground-truth answer spans are retained more often than pruned, showing that the smoothing mechanism can distinguish high-PPL noise from high-PPL factual tokens. Third, the behavior on numeric tokens shows that the pruning process separates meaningful operands from erroneous intermediate calculations. These patterns collectively demonstrate that token pruning operates in a selective and semantically grounded manner. **We have incorporated these results into Appendix H (Tables 12–14) and Appendix i (Table 15) in the revised manuscript.**

---

> ### Author Response · Authors · 2025-11-21
> **(Part 4) Response to Reviewer Dzda**
>
> ## Q2: Ask for additional results on larger-scale models.
>
> We thank the reviewer for this helpful suggestion. To directly examine the scalability of Q-Tuning, we extended our experiments to the larger Qwen3 family, including the 8B, 14B, and 32B variants. All experiments follow the same protocol described in Appendix G.1 of our original submission. The models were trained on the OpenHermes dataset and evaluated on the same five benchmarks used throughout the paper: ARC-E, ARC-C, GSM8K, SQuAD, and TriviaQA. This setting allows us to assess whether the performance and efficiency benefits of Q-Tuning remain stable as model capacity increases.
>
> **Aggregated Results Across Qwen3 Models**
>
> | Model        | Zero-shot | Full-Data SFT | Q-Tuning (12.5×50) | Δ | Q-Tuning (12.5×70) | Δ |
> |--------------|------|----------------|---------------------|-----------|---------------------|-----------|
> | Qwen3-8B     | 53.5 | 55.4           | 60.2                | +4.7      | 60.3                | +4.8      |
> | Qwen3-14B    | 51.8 | 61.5           | 67.7                | +6.2      | 67.7                | +6.2      |
> | Qwen3-32B    | 48.9 | 61.8           | 64.9                | +3.1      | 65.4                | +3.6      |
>
> **Results on Qwen3-8B**
>
> | Setting | ARC-E | ARC-C | GSM8K | SQuAD | TriviaQA | Avg. |
> |--------|-------|--------|--------|--------|-----------|-------|
> | Zero-shot | 69.31 | 44.41 | 74.45 | 27.63 | 51.76 | 53.51 |
> | Full Data | 77.78 | 52.20 | 65.13 | 38.05 | 44.08 | 55.45 |
> | Q-Tuning (12.5×50) | 80.95 | 53.56 | 70.51 | 45.76 | 50.16 | 60.19 |
> | Q-Tuning (12.5×70) | 80.95 | 52.20 | 70.13 | 47.60 | 50.58 | 60.29 |
>
> **Results on Qwen3-14B**
>
> | Setting | ARC-E | ARC-C | GSM8K | SQuAD | TriviaQA | Avg. |
> |--------|-------|--------|--------|--------|-----------|-------|
> | Zero-shot | 72.66 | 50.17 | 58.83 | 18.39 | 58.84 | 51.78 |
> | Full Data | 81.48 | 58.98 | 68.08 | 43.06 | 55.72 | 61.47 |
> | Q-Tuning (12.5×50) | 83.25 | 57.63 | 83.17 | 55.88 | 58.65 | 67.72 |
> | Q-Tuning (12.5×70) | 83.95 | 58.64 | 82.49 | 53.93 | 59.33 | 67.67 |
>
> **Results on Qwen3-32B**
>
> | Setting | ARC-E | ARC-C | GSM8K | SQuAD | TriviaQA | Avg. |
> |--------|-------|--------|--------|--------|-----------|-------|
> | Zero-shot | 67.55 | 49.15 | 58.15 | 31.12 | 38.66 | 48.93 |
> | Full Data | 83.42 | 60.34 | 71.19 | 42.29 | 51.80 | 61.81 |
> | Q-Tuning (12.5×50) | 83.42 | 63.39 | 73.84 | 47.53 | 56.26 | 64.89 |
> | Q-Tuning (12.5×70) | 84.83 | 63.05 | 73.84 | 48.34 | 56.84 | 65.38 |
>
> **Results and Analysis.** Across all tested model capacities, Q-Tuning delivers clear and substantial improvements over the Full-Data SFT baseline. On Qwen3-8B, Q-Tuning raises the average score from 55.45 to 60.19, yielding a sizeable gain despite using only a small subset of the training data. The advantage becomes even more pronounced on Qwen3-14B, where the average increases from 61.47 to 67.72. A similar pattern is observed on Qwen3-32B, where Q-Tuning reaches 64.89 compared to 61.81 from full-data training. These gains consistently exceed the full-data results by notable margins, demonstrating that our method is not only competitive but materially superior even at larger model scales. **We have added results in Appendix G.5 (Figure 7)**

---

> ### Author Response · Authors · 2025-11-21
> **(Part 5) Response to Reviewer Dzda**
>
> ## Q3: Ask for more sample selection baselines.
>
> We sincerely appreciate this insightful suggestion. Comparing Q-Tuning against established **coreset selection** methods is indeed essential to position our work within the broader data efficiency landscape. In response, we have significantly expanded our comparative analysis on Llama2-7b and Mistral-7B (using the WizardLM dataset) to include a comprehensive suite of state-of-the-art sample selection baselines, including **Alpagasus[1]**, **DEITA[2]**, **DS2[3]**, and **LESS[4]**.
>
>
> The tables below summarize the performance across five benchmarks under two pruning configurations (12.5% and 25% sample retention, both with 50% token retention).
>
>
> **Results on Llama2-7b with 12.5% samples**
>
> | Sample Pruner      | Token Ratio | Avg. | ARC-E | ARC-C | GSM8K | SQuAD | TriviaQA |
> |--------------------|------------:|-----:|------:|------:|------:|------:|---------:|
> | **Zero-Shot**      | —           | 30.58 | 53.44 | 38.98 |  5.31 | 12.18 | 43.00 |
> | Random             | 50%         | 33.98 | 59.26 | 41.02 |  8.11 | 12.75 | 48.75 |
> | Alpagasus          | 50%         | 33.27 | 58.91 | 41.69 |  6.67 | 12.81 | 46.28 |
> | Alpagasus          | 100%        | 33.65 | 58.91 | 41.36 |  7.58 | 13.45 | 46.96 |
> | Deita              | 50%         | 33.49 | 58.20 | 42.37 |  7.43 | 13.85 | 45.59 |
> | Deita              | 100%        | 33.16 | 58.38 | 42.37 |  5.46 | 13.99 | 45.63 |
> | DS2                | 50%         | 33.53 | 58.73 | 42.37 |  5.99 | 15.05 | 45.51 |
> | DS2                | 100%        | 33.59 | 59.08 | 41.36 |  6.67 | 15.14 | 45.69 |
> | Less               | 50%         | 33.69 | 58.20 | 42.37 |  6.67 | 14.56 | 46.62 |
> | Less               | 100%        | 34.30 | 58.38 | 43.39 |  6.82 | 16.16 | 46.72 |
> | **Q-Tuning (Ours)**| 50%         | **37.74** | **64.20** | **42.03** | **10.54** | **18.79** | **53.12** |
> | Full Dataset       | 100%        | 35.36 | 61.55 | 42.37 |  8.64 | 13.80 | 50.45 |
>
>
> **Results on Llama2-7b with 25% samples**
>
>
> | Sample Pruner | Token Ratio | Avg. | ARC-E | ARC-C | GSM8K | SQuAD | TriviaQA |
> |---------------|------------:|-----:|------:|------:|------:|------:|---------:|
> | Zero-Shot     | —           | 30.58| 53.44 | 38.98 |  5.31 | 12.18 | 43.00    |
> | Random        | 50%       | 33.92| 60.32 | 41.69 |  5.76 | 13.43 | 48.41    |
> | Deita         | 50%       | 33.84| 59.08 | 41.69 |  7.66 | 13.23 | 47.55    |
> | Deita         | 100%         | 33.90| 58.73 | 42.71 |  4.93 | 14.88 | 48.24    |
> | Alpagasus     | 50%       | 33.93| 59.61 | 42.71 |  5.91 | 14.12 | 47.30    |
> | Alpagasus     | 100%         | 34.59| 60.49 | 43.39 |  7.51 | 13.16 | 48.41    |
> | DS2           | 50%       | 34.17| 59.44 | 42.03 |  6.44 | 15.59 | 47.34    |
> | DS2           | 100%         | 34.33| 59.26 | 41.36 |  7.05 | 15.20 | 48.76    |
> | Less          | 50%       | 34.47| 59.44 | 43.39 |  7.05 | 15.89 | 46.60    |
> | Less          | 100%         | 34.01| 59.79 | 41.02 |  6.37 | 14.44 | 48.44    |
> | **Q-Tuning (Ours)**| 50%         | **36.45** | **63.14** | **42.03** | **8.87** | **16.76** | **51.52** |
> | Full Dataset       | 100%        | 35.36 | 61.55 | 42.37 |  8.64 | 13.80 | 50.45 |

---

> ### Author Response · Authors · 2025-11-21
> **(Part 6) Response to Reviewer Dzda**
>
> **Results on Mistral-7b with 12.5% samples**
>
>
> | Sample Pruner | Token Ratio | Avg. | ARC-E | ARC-C | GSM8K | SQuAD | TriviaQA |
> |---------------|------------:|-----:|------:|------:|------:|------:|---------:|
> | Zero-Shot     | —           | 36.98| 66.67 | 46.10 | 18.35 | 10.01 | 43.77    |
> | Random        | 50%         | 42.73| 70.55 | 48.14 | 22.74 | 19.57 | 52.63    |
> | Deita         | 50%         | 44.25| 72.66 | 50.51 | 22.97 | 20.91 | 54.17    |
> | Deita         | 100%        | 44.91| 72.13 | 50.17 | 22.52 | 24.46 | 55.28    |
> | Alpagasus     | 50%         | 42.90| 72.84 | 48.47 | 20.55 | 20.41 | 52.21    |
> | Alpagasus     | 100%        | 43.98| 70.72 | 48.81 | 23.35 | 24.30 | 52.71    |
> | DS2           | 50%         | 42.55| 71.25 | 47.46 | 21.23 | 21.57 | 51.23    |
> | DS2           | 100%        | 44.43| 72.13 | 49.83 | 23.35 | 23.99 | 52.85    |
> | Less          | 50%         | 43.67| 73.19 | 49.49 | 22.90 | 19.61 | 53.14    |
> | Less          | 100%        | 44.63| 73.90 | 48.47 | 23.12 | 21.67 | 56.00    |
> | **Q-Tuning (Ours)**| **50%**         | **46.92** | **71.60** | **48.13** | **29.34** | **27.75** | **57.78** |
> | Full Dataset       | 100%        | 45.83 | 71.25 | 45.76 |  26.69 | 31.81 | 53.67 |
>
>
> **Results on Mistral-7b with 25% samples**
>
>
> | Sample Pruner | Token Ratio | Avg. | ARC-E | ARC-C | GSM8K | SQuAD | TriviaQA |
> |---------------|------------:|-----:|------:|------:|------:|------:|---------:|
> | Zero-Shot     | —           | 36.98| 66.67 | 46.10 | 18.35 | 10.01 | 43.77    |
> | Random        | 50%         | 42.95| 70.19 | 46.10 | 20.62 | 24.07 | 53.74    |
> | Deita         | 50%         | 43.61| 72.13 | 47.80 | 21.00 | 23.16 | 53.97    |
> | Deita         | 100%        | 44.60| 72.66 | 47.80 | 22.29 | 25.87 | 54.40    |
> | Alpagasus     | 50%         | 42.78| 70.37 | 47.80 | 20.85 | 22.00 | 52.87    |
> | Alpagasus     | 100%        | 44.65| 71.60 | 50.17 | 23.35 | 22.38 | 55.73    |
> | DS2           | 50%         | 43.32| 70.72 | 48.47 | 22.74 | 21.15 | 53.49    |
> | DS2           | 100%        | 43.57| 71.43 | 48.47 | 22.44 | 21.72 | 53.76    |
> | Less          | 50%         | 43.85| 71.78 | 47.80 | 22.14 | 24.51 | 53.04    |
> | Less          | 100%        | 45.41| 71.60 | 49.49 | 21.68 | 30.85 | 53.40    |
> | **Q-Tuning (Ours)**| 50%         | **46.78** | **71.78** | **47.11** | **26.08** | **32.79** | **56.17** |
> | Full Dataset       | 100%        | 45.83 | 71.25 | 45.76 |  26.69 | 31.81 | 53.67 |
>
> As shown in above tables, on Llama2-7b with a strict 12.5% sample budget, Q-Tuning achieves an average score of 37.74, surpassing the state-of-the-art baseline (LESS, 33.69) by a remarkable margin of +4.05 points. This trend translates to the stronger Mistral-7B model. At 12.5% samples, Q-Tuning scores 46.92, clearly outperforming other baselinses. **We have added these baselines in Appendix G.3 (Table 9) in the revised manuscript.**
>
>
> [1] Alpagasus training a better alpaca model with fewer data, ICLR 2024.
> [2] Deita: Data-Efficient Instruction Tuning for Alignment, ICLR 2024.
> [3] Improving data efficiency via curating llm-driven rating systems, ICLR 2025.
> [4] Less: selecting influential data for targeted instruction tuning, ICML 2024.

---

> ### Author Response · Authors · 2025-11-26
>
> Dear Reviewer Dzda,
>
> Thank you once again for your valuable comments on our submission. As the discussion phase is approaching its end, we would like to kindly confirm whether we have sufficiently addressed all of your concerns (or at least part of them). Should there be any remaining questions or areas requiring further clarification, please do not hesitate to let us know. If you are satisfied with our responses, we would greatly appreciate your consideration in adjusting the evaluation scores accordingly.
>
> We sincerely look forward to your feedback.
>
> Best regards,
>
> Authors of submission 1562

---

### Author Response · Authors · 2025-11-21
**General Response**

We sincerely thank the Area Chair and all reviewers (Dzda, ZLqF, 2BBK, MSz7) for their constructive feedback. We are encouraged by the reviewers’ shared recognition of the importance of principled, scalable data pruning in SFT and the positive assessments of Q-Tuning’s unified formulation, methodological clarity, and empirical strength. To address reviewer concerns and strengthen the paper, we conducted extensive new experiments and analyses during the rebuttal period, adding **17 new experiments** and **14 pages of supplementary analyses**, included in the revised manuscript.

 **1. Extended Evaluation on Qwen3-8B.**  Beyond the LLaMA and Mistral families, we further examine Q-Tuning on Qwen3-8B using OpenHermes under 12.5% and 25% sample ratios with 50% or 70% tokens. Under these settings, Q-Tuning continues to match or exceed full-data SFT; notably, with 12.5% samples and 50% tokens it achieves 60.19, substantially higher than the full-data score of 55.45. **Results appear in Appendix G.2 of the revised manuscript.**

**2. Stronger Baselines for Sample-Level and Token-Level Pruning.**  To broaden the comparison scope, our experiments now include recent sample-pruning approaches (Alpagasus, Deita, DS2, Less) and token-level pruning or cleaning methods (Rho-1, TokenCleaning). Across equal budgets, Q-Tuning maintains clear advantages and frequently surpasses full-data SFT; for instance, on LLaMA2-7B with 50% tokens it reaches 37.74 and 36.47 at 12.5% and 25% samples, both outperforming the full-data 35.36. **Details are presented in Appendix G.3.**

**3. Extreme Low-Data Regimes on Mistral-7B.**  We also investigate whether Q-Tuning remains effective when data is extremely limited. Even when retaining only 3–9% of total tokens, Q-Tuning sustains averages of 46.23–46.92, which is well above zero-shot (36.98) and slightly stronger than full-data SFT (45.84). Remarkably, with 6.25% samples and 50% tokens (effectively 3.125% data), Q-Tuning continues to outperform the full-data baseline. **Corresponding results are included in Appendix G.4.**

**4. Scalability Across Different Model Scales.**  Scaling behavior is further assessed using Qwen3 models of 8B, 14B, and 32B parameters. Across all sizes, Q-Tuning consistently yields higher accuracy than both zero-shot and full-data SFT, with improvements becoming more pronounced for larger models. On GSM8K and ARC-C, Qwen3-32B with Q-Tuning clearly surpasses its full-data counterpart, most notably under the 70% token budget. **See Appendix G.5 for full results.**

**5. Quadrant-Level Ablation of the EU-Plane Design.**  To validate the underlying EU-plane strategy, we run ablations that selectively activate or disable pruning in Q1–Q4 at both sample and token levels. Experiments on Wizard and OpenHermes consistently confirm our intended pruning configuration: removing Q1/Q3 at the sample level while pruning only Q2 at the token level. **Ablation findings are available in Appendix H.**

**6. Semantic Analysis of Pruned and Kept Tokens.**  A closer look into the token-level behavior is provided through the analysis of over 44,000 annotated tokens from 300 Q2 samples. The results show that Q-Tuning reliably filters out boilerplate or filler content while preserving gold answer spans, crucial reasoning steps, and key numerical information. **These observations are further detailed in Appendix I.**

**7. Efficiency Analysis in Theory and Practice.**  Finally, both theoretical FLOPs estimation and empirical measurements are conducted to assess efficiency. Q-Tuning reduces computational cost from roughly three forward passes to about (1 + 3r) forward passes, with practical settings satisfying r < 2/3. Empirical tests on LLaMA2-7B with eight A100 GPUs show reductions in training time (90 → 65 minutes) and memory usage (78% → 27%) without sacrificing accuracy. **Comprehensive results appear in Appendix J.**

We appreciate the reviewers’ recognition that improving data efficiency in SFT is a central challenge. Reviewer 2BBK emphasized the “critical challenge of data efficiency,” and Reviewer ZLqF highlighted the value of a “unified framework” for joint sample- and token-level pruning.

Regarding the methodology, we value the consensus that our framework is clear and well-structured. Reviewer Dzda noted that the “motivation and specific pruning methodology is well-explained,” and Reviewer 2BBK described the EU Plane as an “intuitive and effective” formulation. Reviewer ZLqF additionally appreciated the “interpretable diagnostic space” connecting error and uncertainty. Reviewer MSz7 found the algorithm’s intuition “easy to understand.”

We also thank the reviewers for acknowledging the strength of our empirical evaluation. Reviewer Dzda and Reviewer MSz7 noted its “comprehensive” nature, while Reviewer ZLqF and Reviewer 2BBK highlighted that Q-Tuning “consistently matches or surpasses full-data SFT” and outperforms strong baselines across models and pruning ratios.

---

### Author Response · Authors · 2025-11-27
**Follow-up on Rebuttal**

Dear Reviewers,

We sincerely appreciate your thoughtful feedback and the time you have invested in evaluating our work. We have taken all comments very seriously and prepared a comprehensive, point-by-point response during the rebuttal period.

To address the concerns and suggestions raised in the initial reviews, we conducted an extensive set of additional analyses and experiments. In total, we **added 14 pages of new analysis and performed 17 new experiments**, which we have carefully summarized in the updated appendix.

As the rebuttal period is nearing its end, we kindly wanted to check whether there are any remaining questions or clarifications we can help address. We would be glad to provide further details promptly if needed.

Thank you again for your time, effort, and constructive feedback.

Best regards,

Authors of submission 1562

---

### Note · Program_Chairs · 2026-01-17
**Submission Desk Rejected by Program Chairs**

The following references in this submission do not refer to real documents and/or have major errors in bibliographic information:

 Ziyu Huang, Xinyu Li, Yicheng Zhao, et al. Mates: Model-aware training example selection. In Advances in Neural Information Processing Systems (NeurIPS), 2024.